

# Grand Challenges in the Design, Manufacture, and Operation of Future Wind Turbine Systems

Paul Veers[1], Carlo L. Bottasso[2], Lance Manuel[3], Jonathan Naughton[4], Lucy Pao[5], Joshua Paquette[6], Amy Robertson[1], Michael Robinson[1], Shreyas Ananthan[7], Athanasios Barlas[8], Alessandro Bianchini[9], Henrik Bredmose[8], Sergio González Horcas[8], Jonathan Keller[1], Helge Aagaard Madsen[8], James Manwell[10], Patrick Moriarty[1], Stephen Nolet[11], Jennifer Rinker[8]

[1] National Renewable Energy Laboratory, Golden, Colorado, 80401, USA
[2] Technical University of Munich, Garching b. München, 85748, Germany
[3] University of Texas, Austin, Texas, 78712, USA
[4] University of Wyoming, Laramie, Wyoming, 82071-2000, USA
[5] University of Colorado Boulder, Boulder, Colorado, 80309-0425, USA
[6] Sandia National Laboratories, Albuquerque, New Mexico, 87185, USA
[7] Siemens Gamesa Renewable Energy, Inc., Orlando, Florida, 32826, USA
[8] Technical University of Denmark, Roskilde, 4000, Denmark
[9] Università degli Studi di Firenze, Firenze, 50139, Italy
[10] University of Massachusetts, Amherst, Massachusetts, 01003, USA
[11] TPI Composites, Warren, Rhode Island, 02885, USA

*Correspondence to*: Paul Veers (paul.veers@nrel.gov)





**Abstract**. Wind energy is foundational for achieving 100% renewable electricity production and significant innovation is required as the grid expands and accommodates hybrid plant systems, energy-intensive products such as fuels, and a transitioning transportation sector. The sizable investments required for wind power plant development and integration make the financial and operational risks of change very high in all applications, but especially offshore. Dependence on a high level of modeling and simulation accuracy to mitigate risk and ensure operational performance is essential. Therefore, the modeling chain from the large-scale inflow down to the material microstructure, and all the steps in between, needs to predict how the wind turbine system will respond and perform to allow innovative solutions to enter commercial application. Critical unknowns in the design, manufacturing, and operability of future turbine and plant systems are articulated and recommendations for research action are laid out.

This article focuses on the many unknowns that affect the ability to push the frontiers in the design of turbine and plant systems. Modern turbine rotors operate through the entire atmospheric boundary layer, outside the bounds of historic design assumptions, which requires reassessing design processes and approaches. Traditional aerodynamics and aeroelastic modeling approaches are pressing against the boundaries of applicability for the size and flexibility of future architectures and flow physics fundamentals. Offshore turbines have additional motion and hydrodynamic load drivers that are formidable modeling challenges requiring innovation. Uncertainty in turbine wakes complicates both structural loading and energy production estimates and requires advances in plant operations and flow control to achieve full energy capture and load alleviation potential. Opportunities in co-design can bring controls upstream into design optimization if captured in design-level models of the physical phenomena. It is a research challenge to integrate improved materials into the manufacture of ever-larger components while maintaining quality and reducing cost. High-performance computing used in high-fidelity, physics-resolving simulations offer opportunities to improve design tools through artificial intelligence and machine learning. Finally, key recommended actions needed to continue the progress of wind energy technology toward even lower cost and greater functionality are summarized.



# 1 Introduction

Wind energy has by many measures been a great success. The world now depends on over 5% of its electricity supply from wind and the installed capacity is about three-quarters of a terawatt (Global Wind Energy Council [GWEC], 2021). At certain times and places the contribution is much larger; Denmark has exceeded 50% of its electricity from wind on average and the much larger grid demands of Germany topped 30% from wind in the first half of 2020. The expectation is that the energy system needs to move away from fossil fuels to carbon-free sources quickly to respond to international climate

change agreements. The only major carbon-free energy supply sources already ramped up and capable of providing the needed 10 TW or more of new generation within the next two decades are wind and solar. GWEC notes that the current fast pace of wind energy deployment, 93 GW in 2021, needs to double to meet established net-zero carbon goals. The question is whether wind energy is already a fully developed technology that can rely on current understanding to simply reproduce current products in larger numbers, or does it need a more solid scientific basis to feed innovation. This article

suggests that critical research on wind energy yet to be done will be needed and lays out the gaps in understanding that must be addressed to move wind turbine technology forward to meet global demands. Figure 1 illustrates the scope of the issues that need to be addressed in a wind turbine-focused research agenda that enables wind energy to supply its expected share of the carbon-free energy system of the future.

## 1.1 Background

Between the 1980s and today, the size of utility-scale wind turbines increased by well over an order of magnitude, as shown in Fig. 2. Advanced designs, new materials, and the evolution of higher-fidelity modeling capabilities have led to less expensive and more operationally efficient wind turbines. Modern blades are 90% lighter than blades from the 1980s if they were simply scaled up proportionately to the same length (Veers, 2019a). This dramatic reduction is a prime example of how slender blades–with high-lift airfoils operating at higher tip speeds, stiffer materials, and optimized structural

design–have revolutionized modern wind technology. Similar improvements in other components, such as variable-speed drives, coupled with advanced manufacturing processes, and smart operational controls have delivered higher-quality machines with better performance, greater resiliency, lower operation and maintenance (O&M) costs, and extended service life expectancy.

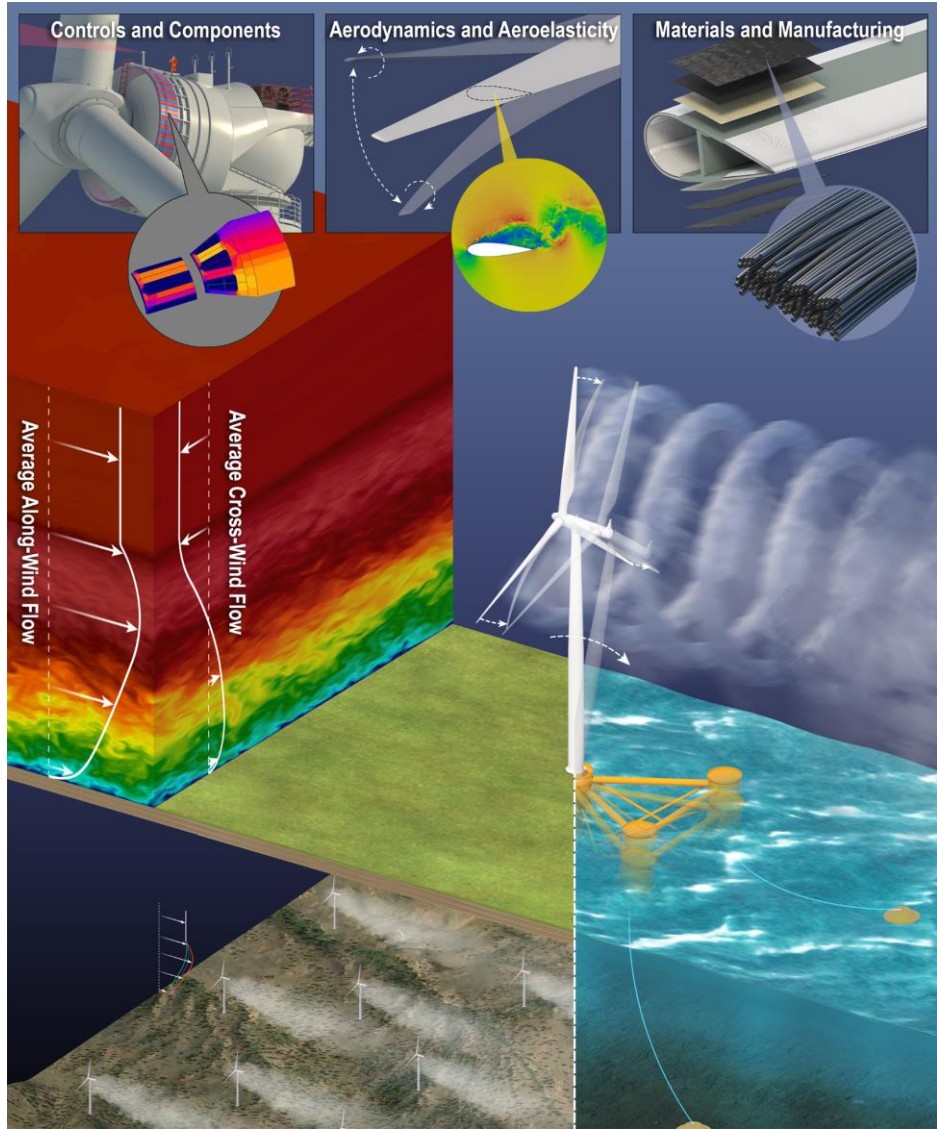

**Figure 1: Current wind turbines are so large their blades extend above the well-behaved atmospheric surface layer. Their long, flexible blades interact dynamically with the complex inflow field. The complexity is enhanced in offshore applications when the turbine is installed on a floating support structure that is itself moving. Within a wind power plant, the flow is also disturbed by the wakes of other turbines. Advanced controls in combination with new sensors and innovative drivetrains (top left), will be**

**integrated into a co-design capable process. Aeroelastic simulation tools (top center) need to capture the larger scale, increased flexibility, and inflow complexity, often using high-fidelity simulations. Manufacturing processes will be expected to use new materials with higher strength and lower weight, with high quality at lengths over 100 m (top right). (Image credit Besiki Kazaishvili, NREL)**



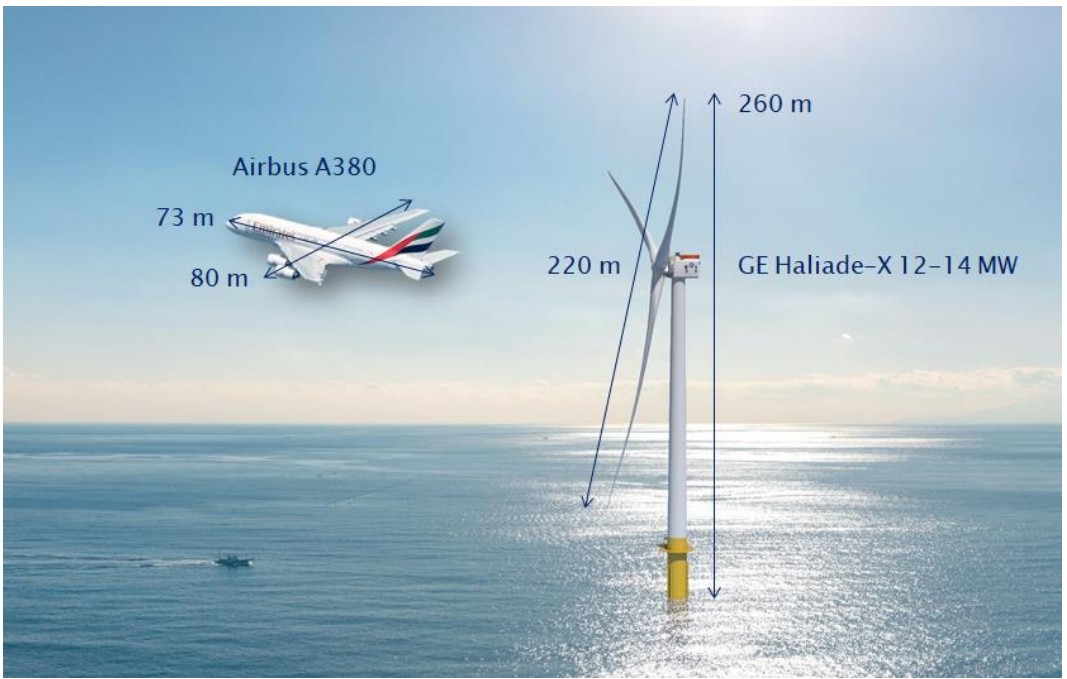

**Figure 2: Modern wind turbines are the largest rotating machines on Earth. It takes three A380s, the largest civil transport airplane in the world, to span the rotor diameter of a large offshore wind turbine. Different from airplanes, however, these machines are designed to operate in a fully autonomous manner in harsh environments, with minimum maintenance while ensuring high availability, accumulating about 100 million fatigue cycles throughout their 20+ year lifetimes. (Haliade-X rendering from www.ge.com; A380 rendering from www.airbus.com)**

Modern wind power plants must operate at sites across a broad range of topographies and environments. Site-specific terrain and atmospheric conditions produce a diverse mix of inflow characteristics that make the design of lightweight wind turbines more complex. As turbines have drastically increased in size, the dynamics of the inflow relative to the size of the rotor, the key design driver, have gradually shifted. The characteristics of the atmospheric boundary layer (ABL), varying both temporally and spatially with weather-driven events, require predictive turbine modeling at a higher fidelity and

resolution than currently available. Modern rotors span a significant portion of the ABL and experience extreme variability through every revolution. The inflow changes dramatically across the rotor diameter and varies in magnitude and characteristic (e.g. velocity, veer, sheer, turbulence structure), depending on local atmospheric conditions and the rotor location within the ABL. Perturbations from turbulence, veer, and sheer produce aeroelastic interactions with the increasingly flexible structures that surpass our experience base with current design tools. Wake interactions from multiple

turbines within wind power plants further complicate turbulence experienced by the blades, as well as reduce overall power production. Moving offshore, and especially into deep waters, raises the bar even higher adding inputs from waves and



currents and coupling the turbine response with that of its massive, possibly floating, support structure. High-fidelity modeling (HFM) and high-performance computing (HPC) resources offer the opportunity to expand our current understanding and enable design tools to effectively explore innovative concepts computationally. But software capable of representing fundamental physics at a full system level depends on code able to run on changing supercomputer architectures and validation of that software in full-scale field experiments.

The continuous market push for lower levelized cost of energy (LCOE) requires all components of the wind turbine system to be as light and cost-effective as possible while maintaining safety, performance, and reliability requirements. While blades and towers have increased structural capacity with both material and design advances, both have become lighter and dynamically softer, which brings them closer to coupling interactions and instability boundaries. Similarly, drivetrain technologies have become more torque-dense and reliable even as their size has increased, yet many design and manufacturing challenges remain. The tools and design assumptions used to create the successful, but costly, stiff, and bulky systems of the past need to be validated with respect to these new design realities. Improved materials and manufacturing methods offer options that need to be proven at scale to get to market. New modeling and analysis approaches are needed to bring higher-fidelity capabilities that incorporate more physics, capture coupled integrated systems, and support both deterministic and probabilistic aspects of optimization into the design process. Experimental campaigns are required to validate these evolving capabilities before they can be trusted for design purposes. Finally, the design process itself needs to be examined to best leverage the improved understanding of the inflow, better characterize as-manufactured properties, and increase sophistication of computational simulation and machine-learning capabilities.

## 1.2 A guide to this article

This article is organized as follows. Section 2 is meant to serve as an effective executive summary, listing the critical challenges related to enhancing the cost-effectiveness and productivity of wind turbine systems. Sections 3–9 explain the scientific gaps in knowledge in each topic area and how those uncertainties are holding back progress. The topic areas are:

- Section 3. Inflow and the design process
- Section 4. Control and co-design
- Section 5. Aerodynamics and aeroelasticity
- Section 6. Wakes
- Section 7. Offshore wind technology
- Section 8. Manufacturing and materials
- Section 9. High-fidelity modeling, high-performance computing, and validation.



Section 10 provides a list of high-level, major research initiatives that address critical gaps and provide the basis for the continued advance of wind turbine science and technology. Interested readers may focus on just those sections that most interest them to understand individual research needs more fully. The recommendations should be read by all because they represent suggested investment in research that will help unlock the full potential of wind energy to meet the demands of
the future energy system.

### 1.3    Comment on scope

There has been a great deal of interest in breaking out of the currently dominant architecture of wind turbines (e.g. horizontal axis of rotation, three blades, upwind, pitch controlled, variable speed) and exploring the opportunities provided by innovative systems, such as airborne wind, vertical axis, or even modestly adjusting the paradigm to operate downwind
or with two blades. There is good reason to expect that substantial advantages may be available, as articulated by Watson et al. (2019). Exploring these innovative options will certainly bring their own challenges. This article is built on a synthesis of the literature to date and aims to build a clear picture of the critical research for the advancement of wind energy systems in general. The literature is dominated by what has been learned in attempting to engineer the current architecture and therefore is a rich source of what needs to be done. Therefore, this article has a natural emphasis on the difficulties
uncovered in this long path to turbines that now dominate the market. However, both innovations in the current architecture and exploration of novel approaches will benefit from the closing of these critical research gaps.

### 2 Summary of outstanding challenges

There are at least four key goals that must be a part of future wind turbine technology development. The first goal is economic optimization, which requires focused attention on driving down LCOE including decisions related to reuse,
repurposing, decommissioning, etc.—all stages in holistic life cycle assessment. Second, design must consider the impact that wind energy development will have on the environment and maximize its value to the community. This requires that we minimize the negative impacts of wind energy technology and in fact seek "good neighbor" status for wind interactions with the natural and built environments. A third goal is achieving seamless and economic integration with other energy generation sources and storage systems. Such integration requires optimized use of the wind resource with tailored specific-
power considerations in low versus high wind-resource regions. The fourth goal is shortening the design cycle, which reduces costs and speeds the adoption of new products to meet changing demands. Accurate, user-friendly simulation capabilities enable rapid design-option evaluation and could also lead to reduced reliance on overly conservative safety factors, thus increasing safety while reducing costs associated with insurance, liability, and failures.

Each increment in wind turbine growth has carefully avoided straying too far from known territory and relied on safety
margins that were sufficient up to that point. However, the large and flexible turbines entering the market today have





gradually pushed into territory where there are significant unknowns. Because of the size, complexity, and cost pressures, many aspects of wind turbine design innovation are stifled by uncertainties, as articulated in Sections 3–9 of this article and summarized here. Substantial opportunities for system improvement depend on resolving these issues.

### 2.1 Inflow and the design process

The design process, as understood and practiced by the industry, is defined by a consensus standard intended to ensure a high level of product reliability. The rapid evolution of wind turbines to unprecedented size and scale and new applications, such as the use of floating platforms offshore or oversized rotors on land, stretches the standard design process. Because the current approach has resulted in reliable products, there may be understandable reluctance to change. However, continued market acceleration to create a carbon-free electricity system requires custom, rapid design optimization for

continuous cost reduction. The design process must be able to accommodate these additional dimensions and bespoke applications.

Wind turbines now extend beyond the top of the atmospheric surface layer and experience inflow characteristics never encountered by smaller machines. Atmospheric flow phenomena at these scales are as diverse and complex as the local driving weather conditions. Extreme and previously unheeded conditions including hurricanes, tornadoes, microbursts,

nocturnal and marine jet flows, and gravity waves must also be considered. Accurate modeling of the ABL state, including detailed inflow characteristics and extreme atmospheric events, is critical for designing the individual wind turbine, as well as the operational control of both turbines and integrated wind plant systems. The exact nature of atmospheric turbulence is not well-characterized at the length scales and resonant frequencies of the modern wind turbine structure. Turbulence at these scales (ranging from hundreds of meters down to just a few meters) is known to be non-Gaussian, nonstationary, and

nonhomogeneous, whereas design criteria are based on conditions that are Gaussian, stationary, and homogeneous. The ABL stable/neutral/unstable transition cycle and corresponding change in the veer, shear, and turbulence (frequency content and coherence) all have fundamental impacts on design loading that need to be incorporated in modeling tools and design standards.

The current design standards often fail to capture the emerging understanding of the underlying physics driving modern

designs. For example, inflow turbulence models incorporate simplified three-dimensional turbulence characteristics that are not intended to match the actual conditions, but to serve as an envelope above the worst cases and thus provide a natural level of conservatism. Often, the design requirements simplify the critical physics that represent particular operating environments (e.g., marine boundary layer, complex terrain). Meanwhile, the ability of modern computational tools to simulate specific atmospheric dynamics and configuration changes has improved to where many such simplifications may

not be necessary. As a result, design standards must evolve to include more of the relevant physics driving modern architectures and anticipate future requirements posed by unique operating environments.



In parallel with seeking a better understanding of the inflow environment, it is also important that the design process intrinsically accounts for and integrates the role of manufacturing, materials, and implied capacity of components and structures. Such considerations must reflect probabilistic consideration in the quantification of resistance as in ultimate and
fatigue limit states, in aero-structural loading assessments, and in anticipated large deflections especially for long and flexible turbine structural components. Life cycle reliability of future wind turbines can also adapt successful practices from other industries, such as aerospace and automobiles, where reliability-centered maintenance programs seek to limit costs and downtime while maximizing safety and other benefits. A future wind turbine probabilistic design paradigm would expand current design standards to include inflow parameters beyond mean wind speed and turbulence. Probabilistic
considerations for capacity, even including operational data, might even allow for a reliability-centered maintenance program.

## 2.2 Control and co-design

Historically, wind turbine control has focused on the performance and survivability of an individual turbine. A fundamental change in design philosophy from individual turbines to multi-turbine wind plants operating as an integrated system has
expanded the role and emphasis of control. As a result, highly flexible wind turbine control strategies are rapidly evolving. Advanced individual control systems are moving toward and/or adopting feedforward paradigms that monitor and "fly" rotors in response to changing inflow conditions (wind and/or waves in the case of floating offshore wind). The objective is to observe and respond to incoming flow to optimize the turbine control for in situ operating conditions. Challenges in control include probabilistic load control, where controllers are designed to account for uncertainties in the inflow as well
as manufacturing variabilities. Another area requiring more development and analysis is that of fault-tolerant operational control that can detect and accommodate faults in a graceful manner through physical and algorithmic redundancies. The ultimate goal is for the control basis and objectives to be able to be programmed and altered in real time, thereby remaining responsive to changes in resource, market, O&M, and operational longevity, as required. Further, as wind turbines are all digitally controlled, there is a need to ensure cyber-physically secure operation of wind turbines.

Wind plant control is adopting turbine-to-turbine situational awareness to actively adapt yaw and pitch in ways that alter wake interaction and maximize plant performance. By selectively lowering the potential performance of a few turbines upstream, the total plant production, O&M cost, and system longevity can be improved. No longer must a turbine simply respond to the inflow resource. Instead, the wind plant system responds temporally and spatially to the inflow environment, modifies the wind resource, and maximizes performance based on systems performance optimization and control
objectives. All of this, however, depends on accurate models of how wakes impact power performance and structural loading. Further work is also needed to develop wind farm site-specific and even turbine pad-specific power performance optimization-based controls that enable adaptive performance optimization in a volatile energy market.



Grid interconnect control is evolving to facilitate the grid of the future where up to 80% of the electricity generated could be from wind and solar. In contrast to traditional power resources (e.g. coal, gas, nuclear, hydropower) that are connected
to the power grid through large spinning synchronous electromechanical machines, renewable generation resources such as wind and solar are generally connected to the power system through power electronic inverters. Major power grid interconnections in many regions around the world are becoming hybrid power systems, comprising both traditional synchronous electromechanical-based resources as well as a noticeable and increasing fraction of inverter-based resources. While inverter controls today are predominantly grid-following, future power systems will involve a mix of inverter-based
resources with both grid-following and grid-forming control capabilities. Grid-following inverters have limited capabilities and must receive voltage and frequency signals from external resources, whereas grid-forming inverters provide functionalities that are traditionally provided by generating resources with synchronous machines (Lin et al., 2020). Grid-forming inverters will be necessary for the stable operation of the bulk power grid with large amounts of wind and solar generation, and ongoing assessments of system performance along with the development of grid-forming controllers will
also be needed (Lin et al., 2020). How both grid-following and grid-forming inverter control performance affect structural responses in wind turbines will also need to be carefully evaluated. As the grid interconnection demands expand, integrated control strategies from the individual turbine through the integrated wind plant are needed to accommodate ancillary services, curtailment, and reactive power responsiveness. The communication strategy, control paradigms, and integrated control architectures are rapidly developing to meet the future needs of integrating large amounts of wind energy into the
grid system. Further, the much higher bandwidths (kHz vs Hz) of the electromechanical and power electronic subsystems needed for integration and hardware control must evolve to guarantee high reliability and resilience.

## 2.3 Aerodynamics and aeroelasticity

The significant blade flexibility of larger modern turbines calls into question many of the design assumptions and well-established performance characteristics of smaller and stiffer platforms. Blade dynamic motion and coupled aerodynamic
response require comprehensive aeroelastic system design and assessment. Stability and damping criteria, if exceeded during nominal or extreme event operation, can drive nonlinear conditions that put the machine at risk and limit operability. Softer and more efficient designs are moving closer to stability boundaries, wherein aeroelastic couplings can induce catastrophic structural failure or reduce life expectancy. The integrated dynamics of such structural design strategies require more holistic systems analysis and modeling methods, capturing both the flow physics and the nonlinear dynamic response.

Classical aerodynamic design approaches assume "linear" deflections and small perturbations, thereby permitting two-dimensional (2D) simplifications of complex three-dimensional (3D) rotor flow and blade motion. Wind turbine blade performance is gleaned from simplified blade element momentum and lifting line theory. Unfortunately, the flexibility of modern blades and the use of 3D flow control devices (e.g. vortex generators) call into question the assumptions of linear deformation, small perturbation, and 2D flow behavior. Further, the operating Reynolds numbers far exceed conventional



wind tunnel test capabilities available worldwide. Hence, among other complexities, the airfoil properties and performance sensitivity to blade surface erosion, deterioration, and fouling cannot be validated. Greater margins of safety must be used in the integrated design of a wind turbine to account for a lack of understanding and an oversimplification in assumptions on dynamics, blade performance, and loads, but there is great uncertainty in exactly what these margins need to be.

Blade and tower flexibility introduce aerodynamic instability considerations that may now be considered as principal
design constraints in flexible architectures. Dynamically soft systems and 3D flow separation phenomena occurring at high Reynolds numbers make existing low- and mid-fidelity modeling and analysis methods incapable of capturing the critical physics and integrated system dynamics. Some examples include:

- Bend/twist blade coupling, flutter from lower torsional stiffness, instability modes involving combined edgewise and flapwise mode shapes, and blade dynamic stall, which introduce additional design model uncertainty.
- Stall-induced vibration (SIV) is an instability occurring when the motion of a blade, oscillating at a structural frequency, drives the incident flow angle beyond static stall, resulting in negative damping.
- Vortex-induced vibration (VIV) is a coupled phenomenon that may appear during separated flow. Sustained vibrations are expected when the frequencies of the shed vortices are in the vicinity of one of the structural frequencies. At different shedding frequencies, VIV can be experienced by wind turbine blades, towers, or
nacelles.

The continuous operation of a wind turbine blade within the lower part of the ABL causes an unsteady inflow to a blade section and a complicated, unsteady aerodynamic response. In particular, the fact that the outer blade continuously traverses heights from 25 m to 250 m of the ABL can cause significant and even extreme variations of the incoming turbulence characteristics. Three-dimensional flow effects diminish the accuracy of 2D polars. Even over one single rotor revolution,
the local turbulent inflow to a blade section can undergo significant changes (Schaffarczyk et al., 2017; Özçakmak et al., 2020; Madsen et al., 2019a). The resulting large disturbances can cause abrupt transitions in otherwise smoothly developing boundary layers, with not yet fully understood consequences (Morkovin, 1985).

Aeroacoustic noise, which depends on blade tip speed raised to the fifth power, can restrict siting options because of the need to set back turbines from habitations. Attempting to lower acoustic emissions can put constraints on the operating
envelope that limit the options for low-wind-speed rotors on tall towers. The design consequence of restricting tip speed for acoustic reasons is a lower generator speed, higher shaft torque, and overall increase in drivetrain mass. Aerodynamic mechanisms that lower the acoustic emissions and facilitate higher tip velocities associated with faster rotor rotation are highly desirable. In fact, they would significantly lower tower-top mass and correspondingly reduce supporting tower and foundation stiffness requirements, thereby significantly lowering the total mass and system cost.





## 2.4 Offshore wind technology

Offshore wind turbine systems need to operate in extremely harsh conditions. These massive, yet dynamically "soft" turbine structures, which are excited by strongly coupled aerodynamic and hydrodynamic forces, push the boundaries of our physical understanding and ability to adequately predict turbine response. Both shallow- and deep-water architectures are subjected to hydrodynamic wave and current loading, wake dynamics, and unique inflow characteristics (e.g., veer, sheer, turbulence, marine jets) of marine boundary layers. Designs are driven by both ultimate and fatigue loading, and require a thorough understanding of the wind/wave characteristics, their interactions, and how to accurately capture their collective influence on these complex, dynamic systems.

Offshore wind architectures are evolving along two pathways determined by water depth. Shallow-water designs may look similar to land-based systems, in which the tower is simply extended and piled into the seafloor. But as the water depth increases, and if the soil conditions make piling difficult, more advanced designs may be needed. Research and development are focused on reducing the cost of the subsurface support structure, installation, and maintenance. In contrast, floating offshore wind turbine architectures are still evolving and innovative designs are being imagined that transcend any existing floating structures in the water today. The full six-degree-of-freedom floating platform motion exacerbates dynamic system coupling and stability concerns. As a result, research and development is focused on integrated system design, floating platform architectures, mooring and anchoring strategies, and advanced control to manage system dynamics. Less traditional wind turbines, including downwind geometries and two-bladed rotors, could well re-emerge as design solutions for floating systems because of the increased benefits of reduced weight and tower-top loads on design costs.

Design simulation capabilities must capture the appropriate physics to compute transient system response at computational speeds that support probabilistic analyses across a range of parametric design requirements. An efficient design process for such coupled dynamic systems, driven by a multidimensional set of stochastic loading inputs, requires a multi-fidelity set of design tools. The accuracy and validation of these tools depends on a new class of high-fidelity computational capabilities based on first principles and limited modeling assumptions.

## 2.5 Wakes

Turbine wakes within large wind power plants impinge on the downwind turbines in complex and poorly understood ways that increase loads, reduce operational life expectancy, and limit energy capture. On a macro scale, the combined wake structure from an entire wind plant can persist over tens of kilometers (especially offshore) and significantly impact the performance of wind plants operating downstream. Wake flow structure and meandering characteristics have a strong synergistic relationship to rotor design and the atmospheric conditions in which they develop and propagate downstream. Depending on spatially and temporally varying local conditions, turbine size, and height-to-diameter ratio, shed wake





vorticity interactions can have strong adverse impacts downwind and represent a significant turbulence enhancement that impacts total load, machine life, production capacity, and even potential changes to the local microclimatology.

Industry models of wind plant-atmosphere interactions are highly uncertain, necessitating field campaigns to gather more detailed observations to achieve a deeper fundamental understanding of wind flows and validate industry and research-grade models. With improved understanding, the wind industry will be able to better design wind farm layouts and operational strategies to reduce overall wake losses and also help improve machine reliability through better management of turbulence within wind farms. A new generation of instrumentation to enable higher-fidelity observations of wind farm flow fields should also be developed.

### 2.6 Manufacturing and materials

Modern wind turbine blades use hybrid composite structures to improve performance, reliability, and longevity, and to reduce total mass, thereby lowering LCOE. Composite materials are manufactured in massive, single-shot molds, requiring consistent high quality over 100 m of continuous material, at less than one-tenth the cost per kilogram of typical aerospace materials. The principal blade load-bearing component is a carbon, glass, or hybrid composite spar cap that transfers the aerodynamic load to the rotor hub. Carbon fiber is often used because of its superior strength and stiffness properties that mitigate blade deflection with reduced mass. Glass fiber is used in the blade skin and leading and trailing edges to both transfer the aerodynamic load to the spar cap and provide buckling strength and toughness. Concern over future constraints that may be imposed by recyclability requirements are encouraging research to examine alternative materials including thermoplastics and more exotic resin formulations that maintain the needed composite material properties.

Although the reliability of major components has improved, challenges remain and are an ongoing concern. Further, as the size and weight of these components increase and turbines are deployed offshore, consequent transportation and installation logistics and maintenance difficulties arise.

Generators often use permanent magnets made with rare-earth elements, which will be in increasing demand across many industries and applications. Therefore, significant investments are needed to overcome the weight and material availability constraints on existing permanent-magnet-generator architectures, such as by developing superconducting generators or other alternatives including magnet reuse and recycling.

Current design adequacy calculations do not incorporate the results of inspections performed either at the factory or in the field. Structural capacity that is based on the smallest detectable flaw and subsequent flaw growth hold the possibility of both enhanced safety and optimal material use. Probabilistic design methods that can directly incorporate inspection and repair of wind turbines during operation have the potential to both decrease component cost and improve reliability.





Increasing numbers of merchant market power plants will require greater knowledge of the current damage state of wind turbine components, and the expected increase in damage during an operational period to be able to evaluate the economics of plant operation. Thus, improvements in inspection and structural health monitoring are needed, as is fundamental research to define a better basis for assessing the progressive damage in components.

### 2.7 High-fidelity modeling, high-performance computing, and validation

Simulations using high-performance computing (HPC) of the fundamental fluid-structure interaction using high-fidelity models (HFM) can capture the relevant turbine and plant physics but come at a substantial computational cost. Advanced computational methods are not readily applicable to design, which requires numerous simulations to represent the stochastic nature of the inflow. On the other hand, design tools using more simplified methods can be inherently conservative and more costly and must be accompanied by safety factors to account for modeling, operational, and physical

uncertainties. As noted in the section on aerodynamics, aeroelastics, and aeroacoustics (Section 5), the increasing uncertainty associated with the accuracy of past/conventional assumptions raises many questions about the appropriateness of historical safety factors. Given the diverse range of uncertainty sources in flow physics characterizations and integrated system dynamics, the extent to which modern, highly flexible designs can be based on simplified physics, and on empirically derived models and approaches, is uncertain.

Future design models will need to couple the fidelity and physics embedded in advanced HFM methods at computational speeds that allow the multiple iterations achieved with today's low- and mid-fidelity models. Artificial intelligence (AI) and machine learning (ML) offer the most promising means by which high-fidelity data from HFM physics-based models are used in training lower-order models for turbine and plant design. Although early in the development stage, these tools have the potential to capture the relevant physics and fidelity needed to simulate integrated complex systems at

computational speeds that are equivalent to lower-fidelity empirical design models.

All models require verification and validation through laboratory and field experiments. Data and the results from analytical models both come with varying levels of uncertainty that must be quantified, such that the range of applicability of the model can be established. The data required to validate high-fidelity models are both expensive to obtain (if even feasible to collect) and represent a significant monetary investment. Most system performance data that exist are proprietary and

held as protected intellectual property by individual companies. The future advancement of modern flexible turbine architectures and advanced wind plant systems will require publicly available, high-fidelity experimental data that are obtained and distributed through public collaboratives and consortia.





### 2.8 How is all this holding back progress?

Great strides have been made over the past decades using simple yet effective models of atmospheric turbulence, advances

in composite manufacturing, and system optimization. The design tools that have been honed over time depend on these simplifications. Yet, wind energy technology has reached the point at which each of the simplifications is pressing against a scientific uncertainty that will limit the ability to innovate. Risk aversion is a fact of life−each wind turbine now manufactured is a multimillion-dollar product and each power plant can be over a billion-dollar investment. The product development timeline for such massive hardware systems can be a decade long and very expensive. The industry is not

going to embrace new technologies, materials, or operating paradigms without the validated ability of computational models to analyze the full-system implications of component-level decisions. The next strides forward in wind energy technology will depend on a revolution in our scientific understanding of these complex and multifaceted systems.

### 3 Inflow and the design process

The multitude of design requirements and coupled effects in a wind turbine system invariably pull the system in

different−often opposing−directions. It is the role of design to bring all of these aspects together, providing the crucial, systemwide point of view wherein all compromises and trade-offs amongst the many internal and external effects are considered. It is the design process that must find an optimal solution between low capital cost, high productivity, long-term reliability, and mitigation of environmental and social impacts. It is also the role of design to ensure that the solution satisfies all necessary constraints. In fact, design constraints extend beyond the domains of aerodynamics, structures, and

controls to include extra requirements linked to production, logistics, operation, maintenance, disposal, recycling, and so on.

Design is an iterative process that leads from the specification of high-level goals and requirements all the way to the detailed definition of a system. In all advanced engineering fields−and wind energy is no exception−this process is fundamentally linked to the ability of simulating the system. In fact, without simulations, the design process is limited to a

build and test (i.e. trial and error), which is lengthy, expensive, potentially unsafe, and in general unable to extensively scan the solution space nor find an optimal solution. Simulation builds on three pillars: 1) science-based mathematical models of all relevant physical effects and their mutual interactions, 2) a holistic approach to couple the numerical methods that solve the models using digital computations to produce predictions, and 3) experimental data that are used for quantifying the accuracy of those predictions with respect to reality and possibly tuning the models. Hence, simulation

technology, with its three pillars, is the key enabler of the design of current and future wind energy systems.

In the wind energy field, design is also a structured process based on standards (International Electrotechnical Commission [IEC], 2019). Standards play the crucial role of distilling the current state of the art into procedures that can be used by





industry, bridging science with engineering practice. Design standards also formulate all the normal, extreme, and emergency conditions that a turbine may encounter over its entire lifetime; typically 20 years (Barone et al., 2012; Manuel
et al., 2013). These conditions ultimately yield the fatigue and extreme loads that the machine should sustain, which may require thousands of transient aeroelastic simulations. Clearly, this has a very large impact on the fidelity of the models that can be used for design. Being at the heart of design and representing a central element of safety, standards are of crucial importance going into the future. First, they should evolve to accommodate the new conditions that extremely large turbines will encounter. Second, standards should evolve as knowledge and analysis methods improve. This evolution is already
happening to some extent, as researchers examine replacing deterministic "equivalent" conditions intended to capture extreme events with more probabilistic-based methods (Moriarty et al., 2003; Sørensen et al., 2010). Further progress in this realm is important to ensure safety while avoiding unnecessary conservatism.

### 3.1 The goals of design

What are the goals of future wind energy technology? Answering this question is key to understanding the scientific
advancements that are necessary and the challenges that lay ahead and need to be overcome.

A first and primary goal is the economic optimization of the process of generating electricity from wind. The most adopted economic figure of merit is LCOE, although other "value" functions are receiving increased attention in recent times (Veers et al., 2019a). Wind LCOE has been making impressive strides in the last 20 years (Veers et al., 2019a), and ensuring low competitive electricity prices remains a key objective in the design and operation of wind turbines and wind power plants.
In fact, wind energy is very different from−for example−aeronautical or automotive applications, wherein there is always a customer interested in paying a premium price to fly a special mission (e.g. search and rescue, transport, military operations) or to drive a high-performance car. Wind energy "flies" a unique mission: generating low-cost, reliable, renewable energy. The problem is made even more complicated by the fact that value assumes various meanings for different stakeholders; for example, the optimal value for society might be different than the optimal value for an energy
company. More generally, future efforts in economic optimization should consider the overall life cycle of the wind turbine. When this is done, better machines can be designed that include many other considerations beyond LCOE, such as, for example, the controllability of the wind power plant, the carbon-dioxide emissions generated during production, installation and decommissioning, the recyclability of the wind turbine at the end of life, the reuse or repurposing of components, and the impact on the environment and the economy.

This leads directly to a second goal, which is to minimize the negative impacts of wind energy technology. Although some effects on the environment are positive (e.g. the increased biodiversity observed in some offshore installations), it is clear that all energy technologies do generate some negative impacts. Wind is clearly no exception to this general rule (for example, see Figure 3 for the onshore case) and can induce, in some cases, limited acceptance or even outright opposition



from the public. It should be a primary goal of design—as the system-level integrator—to make wind energy a "good
neighbor." This means paying close attention to all of the many interactions and couplings of wind turbines with the
extended environment (in its widest sense, including the natural and built environment) in which they operate.

In addition to better integration with the environment, a third goal of future wind energy technology is to seamlessly
integrate into a new energy mix characterized by a prevalence of renewable sources. This goal is strictly related to the
previously cited economic "value"; for example, with an abundance of wind capacity, electricity prices are inversely
correlated with wind speed, and it pays to design turbines and plants capable of producing in low-wind conditions (Madsen
et al., 2020a) while limiting the maximum power in higher winds. It has been shown (Bolinger et al., 2020; Wiser et al.,
2020; Swisher et al., 2022) that a low specific power (i.e. power divided by rotor swept area) system generally leads to the
lowest LCOE, and has the ancillary benefit of delivering power in times of low resource, therefore delivering a higher
system value for the energy. Additionally, the integration of wind into the energy system of the future requires the ability
to provide various services to the grid (Michalke and Hansen, 2013), to work in synergy with storage systems as well as
renewable and non-renewable energy sources, or to use low-cost or surplus electricity for various production and
reconversion pathways, often referred to as Power-to-X (Erdem et al., 2021).

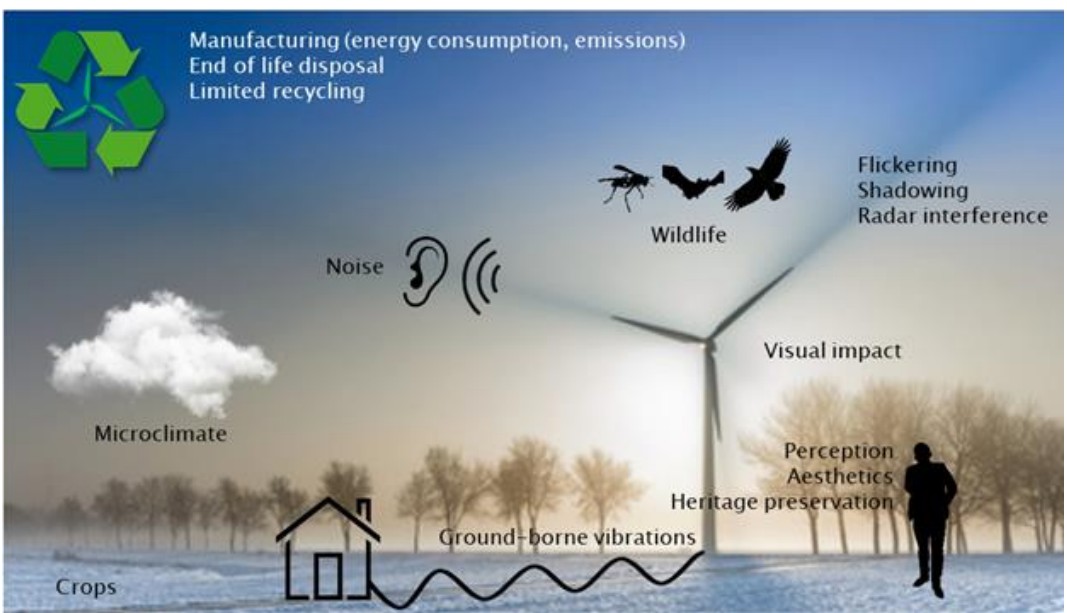

**Figure 3: As all sources of energy, or any human activity, even wind energy produces impacts on the natural and built**
**environment, as shown here. Scientific knowledge and technology can mitigate these impacts, helping wind energy become**
**a "good neighbor." (Figure credits Carlo Bottasso, TUM; background photo Rene Hartmann)**



Finally, a fourth goal is to shorten the design cycle, reducing costs and enabling faster adoption of new products to changing needs and requirements. As witnessed in similar engineering fields, the key enabler of shorter times to market is simulation technology. With general, comprehensive, robust, and reliable validated simulation tools, a closer match between design
predictions and actual field behavior can be achieved, leading to fewer prototypes and posteriori fixes. Simulation also enables customization (i.e. the adaptation of generic design solutions to site-specific conditions) and the evaluation of design innovations that cannot be fielded until it is known that they have a high probability of success. Over time, an improved command of simulation technology also leads to greater confidence, with a reduced need to rely on safety factors to hedge against uncertainties, minimal failures, higher safety, and increased availability. In turn, this spurs a virtuous cycle
that lowers financing and insurance costs and reduces liability, with obvious positive economic implications.

## 3.2 Classical design approach

The classical approach to design uses a process that considers the uncertainties in both the load (L), and the strength or resistance (R), as illustrated schematically in Figure 4. It uses "characteristic" loads and resistances, accompanied with calibrated safety factors, to ensure that the probability of a load exceeding the resistance is at some acceptable (low) level.
The schematic in Figure 4 is easiest to understand in the framework of a maximum load resulting from an extreme event compared against a static strength measure. The strength (capacity) of a manufactured part is never known exactly nor is the extreme structural load a turbine component will experience. Therefore, characteristic values are selected from the upper range of possible loads and from the lower end of the strength distribution to establish a conservative starting point. Then, a safety factor is applied to each characteristic quantity, commensurate with the total uncertainty and calibrated for
a target reliability. The safety factors on the characteristic loads, $\gamma_L$, are sometimes borrowed from older fields (such as wind engineering for buildings or bridges) and an assumption is made that the know-how needed to convert wind speed to loads is similar. Strength safety factors, $\gamma_R$, are derived by applying a cascade of "knock-down factors" to the inherent strength of the material based on the quality of the manufacturing process and underlying material property variability at relevant spatial scales, which are discussed in Section 8. Both these factored terms (on load and resistance) lead to the
design criterion,

$$\gamma_L L < \frac{1}{\gamma_R} R \qquad (1)$$

which, as is evident, requires an inflated load to be less than a knocked-down strength. All the quantities in such design equations should continue to evolve. Advances in the ability to simulate complex loads and represent resistance variables leads to refinement in characteristic values, whereas improved understanding of all uncertainties in loads and resistance
leads to fine-tuning the safety factors. Such progress and refinements are especially needed in new locations such as offshore, where calibration studies informed by simulation and testing are key.

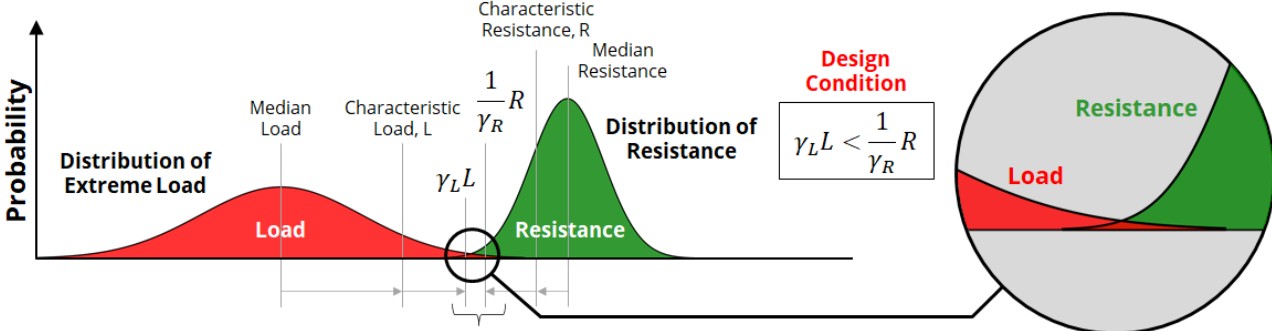

**Figure 4: Schematic of the design process. Characteristic values of loads are taken from the upper tail of the distribution, typically associated with a 50-year return period for an intended 20-year design life. Strength values are taken from the lower tail of the strength distribution at a select level of statistically defined minimum material strength. A margin of safety is defined to ensure that the probability of an uncertain load exceeding the uncertain resistance (strength) is very small. (Figure credit Lance Manuel, UT Austin)**

The idea of a characteristic load is easiest to understand in the context of an extreme or highest wind speed, but there are many other situations in which an operating wind turbine must be evaluated relative to its loading. For example, the characteristic "load" for fatigue is sometimes defined after first deriving a distribution of cyclic load amplitudes driven by some specified inflow over the turbine's design life. A distribution that represents the more severe end of all possibilities is created by assuming an enhanced (extreme) turbulence level; such a conservative distribution effectively envelops the real distribution. The developers of early editions of the IEC 61400-1 design standard back in the 1990s defined a set of design load cases (DLCs) that now includes 23 specific situations or characteristic loads against which the structure must be shown to survive and remain operational. These DLCs include consideration for turbines parked in extreme winds, operating in normal or extreme turbulence at all wind speeds, wind gusts and direction changes during operation, shutdowns during gusts, operating or shutting down with a control fault, and various combinations of these situations. Each of these DLCs produces a "characteristic load" that poses a threat to either the ultimate strength or fatigue durability of the primary structure, and hence to survival of the system. It is assumed that selected aeroelastic models can convert these design situations into loads everywhere in the structure and that these 23 situations collectively represent the key risks to survival.

### 3.3 Inflow modeling

Establishing an appropriate characteristic load for a wind turbine component is a challenge because of the extreme variability in external conditions. During the design phase, the atmospheric conditions must be modeled in a way that drives an aeroelastic calculation of characteristic loads that are robust with respect to a variety of potential site conditions; but this is not an easy task. A wind turbine installed in the mountains experiences a substantially different load profile than one floating offshore. To complicate matters, the loading distribution for each wind turbine has a large spread because of the



stochastic nature of turbulence, which adds uncertainty to the characteristic load calculations. The largest turbines in the market today regularly extend to above the top of the atmospheric surface layer; therefore, the turbulence at the top of the rotor may be significantly different than at the bottom, and wind shear may deviate from standard assumptions as well.
Although the wind industry has enhanced the state of the art such that multi-megawatt turbines can be successfully designed and deployed, gaps remain that hinder the innovation necessary for continuous improvement.

One of the greatest challenges in design inflow definition results from limitations in the turbulence models used in the design standard. For example, the two turbulence models specified in IEC 61400-1 use parameters that were fit to limited data from relatively short meteorological masts (Kaimal, 1990). Moreover, inflow conditions generated using the two
models exhibit substantial differences (Nybø, 2020). These differences in inflow definitions lead to different loads (Eliassen and Andersen, 2016; Doubrawa et al., 2019); sometimes these results are even contradictory and such differences remain to be thoroughly investigated. Other work notes that what designers use as an "extreme turbulence model" (IEC, 2019) might not reflect any real conditions (Hannesdottir et al., 2019; Moon and Sahasakkul, 2014), and the extreme events, in particular wind gust definitions, may be non-conservative at some sites (Rakib et al., 2020). Extreme events, such as
thunderstorm downbursts, can have spatial and temporal characteristics that are quite distinct from IEC design load cases, as noted in analytical models that match recorded events (Nguyen et al., 2011), as well as in LES Weather Research and Forecasting models of realistic downbursts (Hawbecker et al., 2017). Lastly, the shear model assumes a wind turbine operating entirely within the atmospheric surface layer, which is not appropriate for the size of modern-day turbines. Other inaccuracies in the assumed turbulence models exist but cannot be listed because of brevity.

Besides the noted inaccuracies, there are several other phenomena that are missing in current inflow models that likely have a significant impact on design loads. For example, design inflow conditions do not include veer, and yet published results indicate that loads are quite sensitive to it (Robertson et al., 2019). Only simple power-law vertical mean profiles without veer–which are representative of only a neutral boundary layer–are suggested in design standards. Stable boundary layer conditions that occur for longer times over each diurnal cycle can introduce veer, enhanced shear relative to the
power-law profile, and even low-level jets (Park et al., 2014; Rife et al., 2010; Banta et al., 2002). In general, during times of day associated with significant electricity demand, such as during and slightly after the evening transition period, ABL conditions undergo contrasting convective, near-neutral, and stable stratification with associated differences in wind velocity fields not considered in design load cases (Lu et al., 2019), yet important when turbine loads are considered. Non-Gaussian characteristics in wind velocity field structure functions are also not typically considered in inflow design
definitions, although it has been shown that extrapolated design loads can be significantly affected by their inclusion (Gontier et al., 2007). The Kaimal spectral model, in IEC 61400-1, does not offer any guidance on spatial coherence in lateral and vertical turbulence components, which reduces those components to noise. In general, coherence functions as defined in IEC 61400-1 are not consistent with measurements over wind turbine rotor spatial separations, as found from



limited field studies (Saranyasoontorn et al., 2004). Vortex-induced vibrations that arise in parked turbine cases resulting
from extreme wind flow angles cannot be captured without high-fidelity simulations, such as vortex codes or computational
fluid dynamics (CFD) (Zou et al., 2014). Despite all the noted inaccuracies and inexact or missing phenomena in the inflow
models currently in use, the industry experience has not uncovered frequent failures of the primary structure, which
indicates that assumptions of overly energetic inflow turbulence, accompanied by sizable safety factors, are producing
conservative designs. However, the size of modern-day turbines is such that inaccuracies in existing inflow models may
prove to be unsustainable for cutting-edge design and to trim margins in overdesign. This combination of overly energetic
inflow with large safety factors is a rather blunt instrument for design and effectively limits the options that can be
considered within reasonable boundaries of risk. Therefore, reducing the uncertainties and inaccuracies in the inflow
conditions as much as possible would help enable the innovation essential for progress.

In recent work, there have been developments outside of the IEC inflow models that might eventually prove to be essential
in future revisions and improvements of standards for load definitions. Of particular interest is the continual development
of high-fidelity flow-field simulators, most notably using LES, as shown in Figure 1. Although there have been some
comparisons reported between loads generated with LES and those based on stochastic simulation using the two spectral
models in IEC 61400-1 (Doubrawa et al., 2019; Park et al., 2015), there is little published work establishing which of these
methods can best reproduce measurements. LES offers the unique ability to simulate mesoscale events and structures,
which could prove extremely beneficial in simulating a turbine's response in a plant, even for complex and extreme
transient events such as thunderstorm downbursts (Hawbecker et al., 2017; Lu et al., 2019). However, the best method to
couple such mesoscale simulations with microscale physics is currently unidentified, and the potential effect on loads is
also unknown.

Additional work is needed to design turbines that operate in non-standard conditions. Design conditions within the IEC
standard have historically assumed uninterrupted flow and, until recently, relatively simple terrain. However, more and
more turbines are being installed in locations where their inflow conditions do not match that assumption, whether it be
offshore, in complex terrain, or in waked conditions within a plant. Offshore conditions, even considering only wind fields,
are associated with greater uncertainty than for land-based applications, even though it is generally assumed that offshore
winds are accompanied by lower turbulence intensity and higher mean wind speeds. Today, there is no classification system
(i.e. design class definitions) for different meteorological oceanographic (metocean) conditions, thereby implying that
offshore turbines must either be certified using land-based classification, which likely results in a suboptimal design, or
custom designed and certified for each site's local climatology. Finally, many turbine units in wind plants (especially with
deep arrays) will never experience inflow conditions as prescribed in the standard, as a result of the presence of wakes
generated from adjacent upstream turbines. Although much work has been done on the modeling of wakes and the resulting





effects on wind turbine power production and loads, a final standards-based connection to the inflow conditions useful at the design stage has yet to be established.

The wind energy field would benefit from robust techniques to create full-field turbulence data from atmospheric measurements. In aeroelastic evaluation of a wind design, it is essential that turbulence fields used in the simulation studies are as realistic as possible to ensure an accurate prediction of loads. As noted previously, there are considerable deficiencies
in existing turbulence methods; they are often unrealistic and run counter to observations. One possible method to increase the accuracy in turbulent fields is to incorporate wind measurements into the stochastic turbulence generation process, thereby capturing real phenomena and improving the accuracy of the predicted results. There are several methods that have been proposed (Rinker, 2018; Dimitrov and Natarajan, 2017), but their robustness and accuracy have not been demonstrated. Moreover, the techniques generally assume either a Mann or Kaimal spectral model, which could be
inaccurate depending on the flow conditions (as noted earlier). If a technique to reconstruct turbulence fields from measurements could be adequately developed, demonstrated, and regulated, it could not only benefit the wind turbine design process but also be used to reduce loads during operation.

In summary, considering the previous discussion points, it is clear that there are still several challenges regarding modeling of inflow during the turbine design process. An especially significant opportunity for improvement is related to the two
spectral models allowed in the IEC design standard, both of which may be non-conservative depending on the loading conditions (Doubrawa et al., 2019). Several other important phenomena (e.g. non-Gaussian wind velocity field structure functions, wind veer, non-stationary turbulence) are not addressed in current wind turbine design practices, and their omission should be either justified or corrected. The most open fields for research are in high-fidelity modeling, such as in the use of LES mesoscale modeling during the design process, and in reconstructing full-field turbulence from
measurements. Reducing knowledge gaps in these areas is expected to lead to more accurate design loads, which will improve wind turbine designs and advance the wind energy field as a whole.

### 3.4 From inflow to design loads

The process of propagating inflow to simulated turbine loads is by no means straightforward. In the current design process, it is customary to employ a blade element momentum (BEM) model with inflow conditions as specified in the design
standard. However, this method and its simplifying assumptions limit the model's versatility and accuracy in many realistic situations. Longer and more flexible blades, for instance, not only result in large structural deflections but also invalidate key linearity assumptions made in simpler aerodynamic models for wind turbines, as discussed in Section 5. Wind turbine controllers are becoming increasingly adaptable on the fly, in response to varying inflow conditions (Section 4); however, at the same time, differences between the expected and true inflow can lead to large load discrepancies.



The placement of turbines in wind plant arrays complicates load calculations further. Expanding the control strategy to go from considering a single turbine to an entire wind plant necessitates full plant-level simulations. Plant-level control strategies can be implemented to optimize production, respond to grid demands, and so on (see Section 4), but the aero-servo-elastic response of each turbine in the plant is still unique and needs to be considered separately. Furthermore, the wakes within a plant propagate to each turbine differently, and their propagation depends on additional atmospheric

parameters that are not typically considered in design loads calculations (e.g. wind direction or atmospheric stability; see Section 7). The calculation of loads for modern-day turbines, regardless of whether they are standing alone or within a plant, is therefore an area in which useful advances can be made.

Wind turbines are designed so that loads remain within material strength limits in both fatigue and ultimate limit states. These two design constraints are quite different in nature. Fatigue loads account for a cumulative effect that requires

consideration of the entire loading spectrum resulting from exhaustive consideration of operational conditions over the design life of the turbine. While fatigue damage is less sensitive to any single loading event, it requires all loading sources to be aggregated and applied to the selected damage accumulation model for each individual component. The combination of both fatigue and ultimate strength design evaluation requires that aeroelastic models be able to represent extreme events that appear at the corners of the design space, as well as simulate a sufficiently large ensemble of typical operating cases,

to estimate lifetime distributions of fatigue loads. Existing damage models are typically simple, using linear damage accumulation laws such as Miner's Rule to calculate fatigue life (material strength and damage modeling are addressed in greater detail in Section 8). Ultimate loads result either from extreme wind conditions at the site while the turbine is parked or from the largest loads that arise during normal operation over the service lifetime. The extreme values in each case also depend on how the control system interacts with the inflow at the site and include events so rare that they are impossible

to establish empirically during a typical site assessment. Extremes are also difficult to estimate computationally because current simulation capabilities are not validated for unusual atmospheric conditions. The various types of extremes, no matter how they are generated, are stochastic in nature and must also include atmospheric extremes of stability, low-level jets, discrete storms, and other such details now omitted from design specifications.

### 3.5 How models and data dealing with capacity influence reliability-based design

The preceding sections serve to illustrate how external conditions, inflow models, and a probabilistic design philosophy accommodate variability on the "demand" side in a probabilistic design paradigm. The other side of this design framework deals with the "capacity" of wind turbine systems that must rationally, economically, and safely be balanced against imposed demands. Often, the intrinsic (aleatory) variability in coupons, components, assemblages, and sub-systems must be accounted for and, equally important, there is a need to deal with imperfect models, limited test data, etc. that translate

to epistemic uncertainty. Design needs to capture all of the load- or demand-side variabilities with their own complexities





along with capacity-side uncertainties in reliability-based design. The following discussion deals with these capacity-specific challenges in design.

After loads are calculated, the resistance of the structure to those loads must be evaluated. This evaluation includes the
ability to withstand both ultimate and fatigue loads, remain in structural and dynamic stability regimes, and accommodate anticipated deflections. For ultimate loads, material coupon tests are performed to evaluate limit states, typically on a 95/95 basis, which means ensuring that there is a 95% probability of meeting a specified limit state with 95% confidence. Composite materials require more extensive testing than metals, as associated limit states depend on both material orientation and loading direction. Fatigue resistance requires either the cyclic testing of materials at a given load amplitude and mean level until failure or testing at this loading level for a specified number of cycles, and then assessing the remaining
strength. Material properties assessed in this manner are then used as a starting point for resistance calculations, wherein they are further reduced (conservatively) by applying a series of multiplicative safety factors that account for uncertainties in the effects of environmental conditions, temperature effects, the presence of manufacturing flaws, the accuracy of the analysis method, and the fidelity of the load characterization. After inclusion of all these safety factors, "design" values are thus established, which in many cases results in a value of only one-half to two-thirds of the initial resistance obtained from
material testing. Structural and dynamic stability must also be evaluated for wind turbine components. Structural stability issues, one manifestation of which involves buckling, typically affect blades and towers, and can occur at much lower loads than would cause ultimate failure of the materials. Dynamic stability issues, including both system resonance and aeroelastic instabilities, require an evaluation of the interaction of atmospheric forcing and the dynamics of the structural component (e.g. blade), ensuring that vibrations are adequately damped during operation and do not lead to catastrophic
failure of the machine. Finally, for blades, a critical deflection analysis, focused mainly on stiffness considerations, is performed to ensure that they do not strike the tower under extreme loading conditions.

### 3.5.1 Probabilistic considerations for capacity

Wind turbine structures and components have historically been designed using deterministic approaches based on industry-standardized methods of defining structural loads, material resistances, and factors to establish margins of safety. Such
approaches have resulted in hardware that has a high reliability with respect to primary structural survival, whereas individual components have often proved to be insufficiently reliable compared to industry expectations. The goal of existing standards is to ensure that no more than 1% of hardware will fail during a defined 20-year life. On average, major component failure rates are higher than this, particularly with respect to main bearings, gearboxes, and rotor blades (Dao et al., 2019). In some cases, these failure rates reflect uncertainty in defining the structural loads and material characteristics
that fall outside assumptions inherent in the deterministic methods and the accompanying prescribed safety factors that are intended to accommodate that uncertainty. In other cases, the root causes of the failure mode are not well-understood or easily characterized.



As a result, efforts are underway to advance wind turbine and component design to integrate reliability-based approaches
that could result in reducing the LCOE to levels previously considered unrealizable. Such approaches can better account

for the uncertainties in the structural loads resulting from the relevant inflow conditions and operational response, as well
as for the variability in material characteristics arising from differences in materials and manufacturing processes.
Historically, the wind energy industry did not have sufficiently long operational history records that could serve to
empirically validate such probabilistic models. However, the industry has now acquired nearly 40 years of field experience
and 20 years of experience with multi-megawatt turbines that have converged to a relatively consistent architecture (three

blades, pitch control, variable speed, etc.). This, combined with new material and structural test methods, should allow for
a better accounting of the effects of uncertainty in design. Admittedly, operational data–while useful and inherently related
to failure rates with a time-in-use basis to assess reliability–may not be directly compared against other reliability-based
targets and safety factors that are common in reliability-based design codes. Nevertheless, operational data and failure rates
in the field can serve to assess effectiveness of design philosophies that are risk-based and that consider both demand and

capacity uncertainties. Rational use of failure-rate data in reliability-centered maintenance programs is common, for
example, in the aerospace and automobile industry; such practices are not yet codified for use in the wind energy industry
but experiences in the field with turbines can possibly also be adopted and perhaps even be part of life extension strategies.

Adoption of reliability-based approaches to engineering, manufacturing, operation, and maintenance could result in a more
realistic approach to managing the life cycle cost of ownership of wind energy assets. Key to the success of this approach

will be implementing cost-effective methods of manufacturing process quality assurance and quality control that can better
ensure that parts leave the factory with acceptably small nonconformities using, for instance, statistical quality control.
This process is presently particularly challenging with respect to rotor blades, primarily due to their size. Although many
original equipment manufacturers perform ultrasonic inspections or use similar nondestructive testing on some of the
critical load-carrying components (e.g. spar caps), in general much of the blade structure is fabricated without strict quality

control that can guarantee conformities with all critical-to-quality elements of the design stipulated by engineering. Given
the low-cost requirements of the industry, quality assurance associated with many such manufacturing processes does not
necessarily guarantee conformity with statistically quantifiable reliability. Evolving to reliability-based approaches will
require the adoption of such quality assurance and quality control methods that can ensure quantifiable quality.

Rather than establishing unrealistic expectations for decades of low-maintenance operation, new methods must rely upon

cost-effective intervals for inspection and proactive maintenance, establish up-front expectations for major component
replacements, and ultimately produce estimates of remaining useful life as plants age. Many other mature industries,
including aviation and nuclear power generation, rely on reliability-based approaches and reliability-centered operations
and maintenance programs, so the techniques are well-established; however, similar programs will need to be adapted to
the unique characteristics of the wind energy industry. The challenge ahead is to integrate the probabilistic analyses that





quantify loads and demands with manufacturing, quality control, and operational experience and field performance data in
a rational, risk-based design framework.

### 3.6 A new design philosophy

The current wind turbine design practice may have its deficiencies and challenges, but there are many opportunities for
innovation leading to improvement. Class-based design is rather simplistic and reduces the flexibility to customize designs
for local conditions while likely increasing design margins unnecessarily. The classes themselves as defined only consider
reference wind speeds and turbulence categories, which are not the only atmospheric parameters that significantly impact
loads. Therefore, it is left to the site assessment, as part of project certification, to deal with all the external conditions that
fall outside the limited dimension space inherent in class-based design. It is entirely possible that a different prescriptive
code (or perhaps even a performance-based design philosophy) might be more appropriate for modern turbines and
complex sites–one that expands the design parameter dimension space accounting, say, for atmospheric stability and greater
coverage for shear, veer, density, and so on, as well as considers realistic deterministic extreme events, gusts, and site-
based coherence functions and turbulence spectra. Robertson et al. (2019) have shown that the turbulence intensity,
shear/veer, and shape of the coherence function have the largest influence on system loads of the various inflow
assumptions, and yet, of these, only turbulence intensity is explicitly treated as a variable in the current design process.

Any performance-based philosophy or new prescriptive design approach with an expanded design parameter space must
still comply with the underlying design basis. In IEC 61400-1 for land-based wind turbines as well as IEC 61400-3-1 for
fixed-bottom offshore wind turbines, the underlying principle at work is that these standards, when used, seek to attain an
annual probability of failure that is not larger than 0.05% (the associated target reliability index is 3.3). This target
probability is selected to be low enough as to be acceptably safe for land-based and offshore turbines, which fall into the
category "normal safety class," implying that a failure would result in risk of personal injury or some other social or
economic consequence. Partial safety factors used in the standards with design checking equations are calibrated and
applied to characteristic loads and resistances to meet the target reliability. A fully probabilistic design philosophy could
revisit both the nominal loads that could be derived using a higher dimension inflow parameter space and different
calibrated safety factors. This might be a very rational extension to class-based design that diminishes the role of other
wind-field-related variables beyond mean wind speed at hub height and a turbulence category. For instance, atmospheric
stability affects wind shear and wind veer. It is not accounted for in today's codes that assume neutral boundary layer
stability, which is valid only for a very small portion of each diurnal cycle, compared to unstable (convective) and stable
(stratified) boundary layer situations. For today's large rotors, enhanced wind shear and the possibility of including veer
can lead to larger nominal loads, if the parameter space were to be expanded to account for those. A probability-based
design approach that expands the parameter space might seem onerous, but calibration of new partial safety factors is
possible and, more importantly, nominal loads defined in this new framework could possibly help better identify design



drivers that control rotor and tower design and trim margins of conservatism in existing standards. With changes in turbine design paradigms for land-based and offshore use, such refinements are timely, and indeed IEC is embarking on a probabilistic design study that is expected to develop a technical specification ahead of what may soon be a design standard.

## 4 Control and co-design

Control of wind energy systems can be roughly broken down into controllers for wind turbines, wind plants, and grid integration. In each of these areas, control design can lead to significantly improved performance. For instance, at the wind turbine level, an adaptive controller demonstrated 5% to 14% improvement in power capture than the industry-standard controller (Johnson et al., 2004, 2006). As this was only a software algorithm change and did not require any new hardware or sensors, the increase in power capture meant a direct increase in revenue. Similarly, many controllers have been developed and experimentally shown to lead to lower structural loads (Bossanyi, 2003; Stol et al., 2006; Laks et al., 2011; Dunne et al., 2011; Bottasso et al., 2014a; Petrović and Bottasso, 2017; Sinner et al., 2021; Zalkind et al., 2021). Individual pitch controllers that vary the pitch angle of each blade independently (Bossanyi, 2003; Stol et al., 2006) can both improve the regulation of the generator speed as well as reduce structural loading compared to simpler collective blade pitch controllers wherein the pitch angles of all the blades are controlled collectively. The use of preview wind speed information, such as from a lidar sensor mounted in the hub or on the nacelle of a wind turbine, can further reduce structural loads (Laks et al., 2011; Dunne et al., 2011). Mitigating structural loads can mean longer component and turbine lifetimes. Alternatively, the ability of controllers to mitigate blade structural loads could mean that, for instance, lighter-weight blades could be designed and used. Lighter-weight blades could mean that lighter-weight towers are then needed, and lighter-weight components typically mean lower transportation and assembly costs (where it is less expensive to hoist a lighter-weight rotor onto the turbine).

More advanced model-predictive controllers can lead to optimized performance while accounting for various constraints (blade pitch constraints, structural load limits, and/or generator overspeed limits) (Bottasso et al., 2014a; Petrović and Bottasso, 2017; Sinner et al., 2021). Model-predictive controllers are computationally demanding, however, and there have only been very limited experimental demonstrations of such wind turbine controllers to date (Sinner et al., 2021). An alternate method for accounting for generator overspeed constraints while boosting power performance, that is less computationally intensive, builds upon what are considered industry-standard generator torque and blade pitch controllers (Zalkind et al., 2021). As wind turbine rotors become larger and more flexible, higher-order models are needed for controller design and higher-fidelity software (as discussed in Sections 3 and 5) are needed to validate the expected controller performance. With more flexible rotors, predicting the achievable power production performance also becomes more complex and simplified LCOE estimates that are often based on rigid rotor assumptions will need to be expanded in order to provide accurate predictions.



### 4.1 Wind plant control

At the wind power plant level, coordinated control of wind turbines across the plant to optimize plantwide objectives has shown that overall wind plant power can be increased, often while reducing average structural loads. One of the most promising developments in wind plant control is wake steering. With wake steering, the total power generated along a row of turbines in a plant is considered, rather than individual turbine power production. Simulation studies have shown that, by intentionally yawing upstream turbines slightly away from the wind, a radial aerodynamic force is generated that can push (or steer) the wake away from downstream turbines (Jiménez et al., 2010). Then, the downstream turbines experience higher-velocity inflow and can generate more power than in waked conditions. While the intentional yawing results in a gross power loss at the upstream turbines, the recovery of higher wind speeds at the downstream turbines can result in a net gain in power across a row (Fleming et al., 2015). This potential has since been realized both using wind tunnel testing (Campagnolo et al., 2016; Bastankhah and Porté-Agel, 2019) and in field tests (Fleming et al., 2017, 2019, 2020; Howland et al., 2019).

The promise of wake-steering control is mitigated by the realization that controlling yaw misalignment results in imbalanced loads, potentially impacting the rotor, drivetrain, and tower loads; hence, damage. This is a good example of how farm level power production enhancement trades off against individual turbine loads and cost. A significant barrier to addressing this trade-off is the lack of quantitative models of damage leading to component failure or expensive maintenance actions, as discussed further in Section 8. The full connection between energy gains and maintenance burden to date is not well-understood. Progress depends on detailed material damage models, higher-fidelity models, and integrated tools for asset-level and farm-level simulation to further optimize the design of the system and controls. Given certain assumptions about wake and structural loading models, distributed optimization and control methods can be used to coordinate and effectively manage the structural load distributions across a wind farm (Vali et al., 2019, 2021).

### 4.2 Extensions for grid support and offshore

In terms of enabling the integration of increasingly larger amounts of wind power into the grid while maintaining its stability, it has been shown that wind turbines can be actively controlled to produce power that follows a power reference from system or wind plant operators (Aho et al., 2012, 2013, 2013a, 2016; Ela et al., 2014; Wang and Seiler, 2015; Tang et al., 2019; Pao, 2021). Active power control at the wind plant level has also been an active research area (van Wingerden et al., 2017; Bay et al., 2018; Vali et al., 2019, 2021), but is more complex because of uncertainties in how wakes develop and meander. Such active power control studies show that wind turbines and wind plants can effectively provide ancillary services for the power grid (Aho et al., 2016; Hansen et al., 2016; Rebello et al., 2020).



As offshore wind moves farther from the shoreline and shallow waters into deeper waters, there will be a need for active research into floating offshore wind turbines and the control of such structures. Floating wind turbines have more degrees of freedom, and control studies have uncovered instability issues associated with the platform pitch mode and its association with negative aerodynamic damping, which appears to be problematic in a variety of floating foundation types (Larsen and Hanson, 2007; Jonkman, 2008; van der Veen et al., 2012; Fischer, 2013; Fleming et al., 2016; Yu et al., 2018). Additional tower-top velocity and/or platform pitch velocity feedback has been shown to improve performance (van der Veen et al., 2012; Fischer, 2013; Fleming et al., 2016; Yu et al., 2018). Moreover, the challenges of controlling floating offshore wind turbines are expected to become even greater as lighter-weight platforms are designed to reduce capital costs (Damiani and Franchi, 2021). Controllers must be designed to actively mitigate oscillations of large wind turbines atop soft and lightweight support structures in complex environmental conditions that could include typhoons and hurricanes.

### 4.3 Incorporating control development inside the design loop

Wind turbines have traditionally been designed in a sequential process with the aerodynamic design of the rotor completed first, followed by the structural design that includes the detailed layup of materials to be used in manufacturing the blades. Usually, by the time the control systems engineers are requested to design and tune the controllers, the wind turbine has already been completely designed (Garcia-Sanz, 2019). In this sequential design process, conservatism is built in at each step to ensure success at the next step, thereby leading to sub-optimal overall designs. Co-design methods that simultaneously optimize the system and the control design have been shown to yield superior results (Garcia-Sanz, 2019; Patil et al., 2012; Allison et al., 2014; Herber and Allison, 2013). Because of the complexity of the wind turbine design process, with numerous parameters that must be determined, and where it is not clear how certain parameters depend upon others, a fully integrated optimization with consideration for objective functions involving all the physical wind turbine and control design parameters is not currently possible. Even when parameter dependencies become better understood, such simultaneous optimization will remain computationally intensive for wind turbine design. Given existing software tools, an iterative, sequential, co-design process (Fathy et al., 2001; Allison and Herber, 2013) is currently the most practical approach for applying co-design to wind turbines (Pao et al., 2021).

In contrast to simultaneous co-design, wherein both the wind turbine and controller are simultaneously designed to minimize some cost function (such as LCOE), in each iteration of a sequential iterative co-design process (see Figure 5) the wind turbine is first designed to optimize one measure (such as minimizing mass subject to structural stability constraints) and then the controller is designed to optimize another function (such as maximizing annual energy production (AEP) subject to generator overspeed constraints). Each iteration makes use of performance measures (e.g. mass, structural loads, AEP, LCOE) from previous iterations to try to achieve improved performance. Developing automated controller tuning methods can help expedite the tuning of many controller designs (assuming a particular controller structure) across



multiple wind turbine designs being evaluated in the co-design process (Bottasso et al., 2012; Tibaldi et al., 2014; Zalkind et al., 2019, 2020).

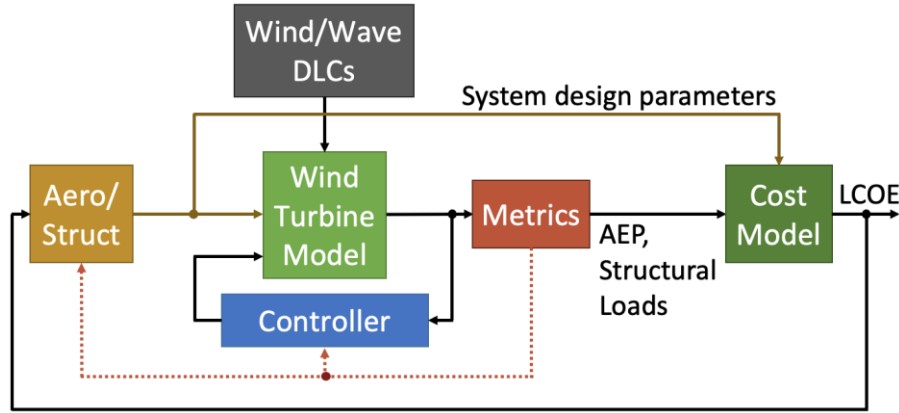


**Figure 5: This block diagram illustrates the integrated wind turbine aero-structural-control co-design process. An initial aerodynamic and structural design model is translated into a wind turbine simulation model in an aeroelastic software tool such as FAST (Jonkman and Buhl, 2005). A controller is required to evaluate simulations defined in design load cases (DLCs) specified by the International Electrotechnical Commission (IEC) (IEC, 2019). Feedback paths indicate that**


**various performance measures (such as annual energy production and structural loads) from the results of the DLC simulations are fed back so that the wind turbine and controller designs can be refined to improve on these metrics. Wind turbine system design parameters and performance metrics are used in cost models that determine the levelized cost of energy (LCOE). Information from the cost analysis is then used in subsequent design cycle iterations until a sufficiently optimized wind turbine design is achieved that meets cost and other objectives. (Figure credit Lucy Pao, CU Boulder)**

At the current time, the software tools available do not easily allow for simultaneous optimization of the many aerodynamic, structural, and control parameters at the level of fidelity typically expected in wind turbine design. The software tools (such as PROPID and PROFOIL (Selig and Tangler, 1995; Selig and Maughmer, 1992) that aerodynamicists use to design wind turbine blades are not easily integrated with the tools (such as NuMAD [Berg and Resor, 2012]) that structural dynamicists use to determine the detailed structural layup of materials in wind turbine blades or the tools (such as MATLAB® and

Simulink®) control experts use to design controllers. By working more closely together and providing regular feedback between the usual disciplinary teams, a novel, 13-MW, two-bladed downwind turbine was designed with a 25% lower LCOE than a conventional, 13-MW, three-bladed upwind turbine (Pao et al., 2021). While using lower-level fidelity models, systems engineering tools such as the Wind-plant Integrated Systems Design and Engineering Model (WISDEM®) (Dykes et al., 2011, 2014) are becoming invaluable for initial co-design iterations to explore larger design spaces and

narrow the potentially optimal options that are then assessed in further detail using higher-fidelity software tools (such as



those mentioned earlier). Enhancing the fidelity of these tools and expanding the necessary tools to fully enable accurate control co-design for floating offshore wind turbines is an active area of research and development (ARPA-E ATLANTIS, 2019), and there are many avenues for future work. Eventually, it may even be possible to carry out more advanced nested and simultaneous (Allison et al., 2014; Herber and Allison, 2019; Fathy et al., 2001) control co-design approaches for wind

turbine design. Control co-design at the wind plant level would incorporate layout optimization in the context of potential wind plant control approaches while aiming to integrate detailed design options at the wind turbine level. Expanding co-design to the power grid level would include the considerations of integration of wind with solar and energy storage options, including those that can take advantage of tall towers (Qin et al., 2017). Further expansion of co-design should also explore new applications of wind energy in conjunction with green hydrogen production, water desalination, and other emerging

applications, which may or may not yield different wind turbine architectures (Mehta et al., forthcoming).

## 4.4 Recommendations and future challenges

As the size of wind turbines increases and new wind turbine concepts (e.g. downwind, highly coned, floating) are explored, control co-design pushes the need for computationally efficient and accurate-enough software. It then becomes possible to create stabilizing controllers that optimize wind turbine designs with lower rotor, tower, and substructure mass. Optimized

designs will continue to yield lower LCOE that will enable accelerated deployments of wind power plants worldwide. As discussed in other sections, larger, more novel wind turbine designs are now exceeding the bounds (today) of our understanding of their dynamics and interactions with the atmospheric and marine boundary layers. As high-fidelity software tools continue to be developed, it is also important to derive reduced-order and computationally efficient models and tools that can enable more rapid iterations and evaluations of novel wind energy systems to expedite the development

of optimized floating and larger-scale offshore wind turbines.

There is a need for computationally efficient models as well as computational architectures that are deployable, e.g. leveraging state-of-the-art computational architectures like multi-core central processing units (CPUs), graphical processing units (GPUs), and field-programmable gate arrays (FPGAs) in deterministic control loops, which, in combination with more demanding and complex control algorithms, allow a turbine to better "sense" the operational

conditions that it is exposed to, and more quickly and adequately "act" in response. This applies at not only the turbine level but also at the farm level, as also noted in Sections 3 and 5 on the nature of the inflow and interactions with wind turbines and wind farms. There is also a need to develop and analyze fault-tolerant operational schemes wherein physical and algorithmic redundancy is traded off in order to allow operating windows to be extended. Moreover, it is necessary to ensure cyber-physical secure operation of closed-loop controlled turbines, wherein detection, localization, and

neutralization of attack vectors are hampered by the high dimensionality of the problem and the stochastic nature of the resource.





Further in line with arguments made in Section 3, there is a need to develop probabilistic load control methods, which would shift the way control algorithms are designed from peak load control to control of load distribution. Accounting for wind direction variability and uncertainty in wind farm wake steering has shown that active wake deflection has a high
sensitivity toward short-term wind directional changes (Rott et al., 2018). Considering wind direction and/or yaw position uncertainty, it has been shown that robust wake-steering controllers can be designed to still yield overall wind farm power production increases in realistic inflow conditions (Rott et al., 2018; Simley et al., 2020). Extensions of these studies, at both the wind farm and wind turbine levels, and to account not only for power production but also structural loads, are areas that require more attention.

Another challenge is to develop wind farm site-specific (and even specific to each turbine location) power-performance-optimization-based controls that enable non-linear life consumption by design, and implementation of life versus performance trade-offs that allow dispatch optimization of wind power in a volatile energy market. A further challenge with adaptive wind farm or wind-turbine location-specific operational control strategies is the ability to measure and quantify the power performance impact of such solutions. This measurement challenge impacts not only the development
stage during which the benefit of such solutions needs to be validated, but also the deployed stage wherein monetization of such solutions hinges on such capability. This is not a trivial problem, considering the relatively small differences that are often realized and the dependency of those on local wind conditions that change diurnally and seasonally.

## 5. Aerodynamics and aeroelasticity

The aerodynamics of the rotor, as it interacts with the inflow, dictates the efficiency of the energy conversion process from
wind. However, the role of aerodynamics is not limited to establishing the performance of the turbine. In fact, the interaction of the airflow with the blades generates loads, which excite the response of the overall structure of the machine. The ensuing deformed shape of the structure and its motion alter the aerodynamic loads that generated them in the first place, creating a feedback mechanism. Additionally, the dynamic motion of the structure generates inertial forces, which couple with the aerodynamic and structural ones. In offshore applications, further couplings are generated by the interaction of the turbine
and its support structure (either fixed bottom or floating) and associated hydrodynamic influences. In turn, the controllers that govern the turbine react to the changed response of the machine. The closure of this multi-way loop combining aerodynamics and hydrodynamics, structural deformation, inertial forces, and actuation is termed "aeroelasticity" (Rasmussen et al., 2003; Hansen et al., 2006).

Aeroelasticity was typically not a major concern with earlier wind turbine designs, except in some stall-regulated machines
that were prone to sustained edgewise vibrations in some operating conditions (Rasmussen et al., 1993; Rasmussen et al., 1999; Chaviaropoulos, 2001; Riziotis et al., 2004; Hansen, 2007). In fact, because of the conservatism intrinsic in any new





technology, early wind turbines featured stiff oversized structures with ample safety factors, implying a reduced role for aeroelastic phenomena. This situation has drastically changed today: wind turbines have been steadily growing to increase capacity factors and exploit scale effects (Veers et al., 2019a) (Figure 2). Unfortunately, however, bigger machines cannot

be designed by simply scaling up existing ones. In fact, weight−which is strongly correlated with cost−grows with the cube of size, whereas energy capture−being proportional to rotor swept area−grows only with the square of size. Technological innovation in design, simulation, materials, controls, and manufacturing has allowed industry to deploy machines that break this vicious square-cube law, resulting in the large, slender, flexible solutions available today whose weight has only grown by an exponent on size of about 2.3 instead of 3 (Griffith and Richards, 2014).

For modern wind turbines, aerodynamics and aeroelasticity dictate, to a large extent, both the internal behavior of the machine, in terms of performance and loading, and its external interaction with the environment, in terms of noise, visual appearance, vibrations transmitted through the foundations, and modifications to the ambient flow. In particular, aerodynamic and control choices (and, to a lesser extent, aeroelastic ones) have a profound effect on the wake released behind a rotor, and hence impact the behavior of the entire wind power plant.

**5.1 Unknowns, and their impact on the ability to make progress**

There are a number of unknowns that stand between the present technology and the future goals outlined earlier. These can be grouped into physics, modeling, numerical methods, and experimental unknowns.

**5.1.1 Physics unknowns**

Wind turbine blade airfoils operate at large Reynolds numbers in highly unsteady conditions (Leishman, 2002), often in a

degraded surface state because of soiling and erosion (Sareen et al., 2014). Even over one single rotor revolution, the local turbulent inflow to a blade section can undergo significant changes (Schaffarczyk et al., 2017; Özçakmak et al., 2020; Madsen et al., 2019a). The resulting large disturbances can cause abrupt transitions in otherwise smoothly developing boundary layers, with not yet fully understood consequences (Morkovin, 1985). Because of a lack of insight into the transition mechanisms on rotor blades, airfoils are typically designed to provide an empirical mix of transitional and fully

turbulent characteristics (Özçakmak et al., 2020), which can be a conservative approach, e.g. used in the recent design of the International Energy Agency Wind (IEA Wind) 10-MW offshore reference wind turbine (Bortolotti et al., 2019a).

Another area in which knowledge is insufficient is aeroelastic stability. In fact, a precise evaluation of aeroelastic damping remains an elusive goal in some operating conditions. Additionally, in parked and idling conditions, highly flexible rotors can be prone to stall-induced vibrations (SIV), whereas both blades and towers may suffer from vortex-induced vibrations

(VIV), which may lead to reduced lifetime (Thirstrup Petersen et al., 1988; Heinz et al., 2016).





These and many other phenomena are not only poorly understood but also affected by a wide range of uncertainties, including the flow conditions, material properties, state of degradation, and damage of the turbine and its components. There is also a lack of understanding of the effects and coupling of these uncertainties. Although safety factors can be used to hedge against uncertainties, they do not "model" uncertainties per se, and can thus result in suboptimal or occasionally 910 even unsafe solutions.

### 5.1.2 Modeling unknowns

Two key aspects of models of physical systems are their generality and fidelity. Generality refers to the model's applicability to a range of possibilities, whereas fidelity refers to the accuracy with which the behavior of the real physical system is represented by the model. Limitations in both the generality and fidelity of models push their use outside of the 915 validity range with unpredictable results, which may hinder performance, safety, and longevity.

For example, blades are routinely modeled as beams (i.e. one-dimensional line elements in 3D space). Because of their computational efficiency, beams will probably remain the backbone of most simulation environments in support of design, load analysis, control synthesis, and certification. Yet, large deflections, complex geometries, and couplings challenge the assumptions of some of the most commonly used beam models (Rasmussen et al., 2003; Larsen et al., 2004). Similarly, 920 most aerodynamic models are challenged by present and future applications. Specifically, lifting lines and 2D airfoil characteristics are used in the full range of aerodynamic rotor models, from blade element momentum (Madsen et al., 2020b) and free vortex wake (Voutsinas et al., 2006) to large-eddy simulation actuator line models (Martinez-Tossas et al., 2015). All these methods rely on the common assumption that local blade loads can be computed from 2D airfoil characteristics, based on local flow conditions computed from the motion of the lifting line and its interaction with the 925 ambient inflow and the wake induction. Although a plethora of empirical and semiempirical corrections do exist to account for various phenomena and approximations in lower-order models, there is a lack of understanding on how to consistently correct 2D characteristics to model the full 3D turbulent flows on modern blades with arbitrarily complex shapes. Unsteady, large, angle-of-attack conditions, as those encountered at pronounced yaw misalignment angles or at standstill, pose further challenges and put the validity of many 2D models in question.

### 930 5.2 Modeling computational capabilities

Simulation technology–working in tandem with modeling–is one of the key enablers of past advancements in wind energy and is of crucial importance for any future progress.

At the knowledge-generation level, modeling and simulation are key to understanding physical phenomena. This perspective is particularly relevant in the case of wind energy because the field environment cannot be controlled at will, 935 and full-scale measurements are still technically challenging, sometimes not very accurate, or altogether impossible to





obtain, and often very costly. Some of the most severe challenges come from the fluid aspects of the problem, which imply the numerical solution of the Navier-Stokes equations by using CFD. A wide range of approaches has been developed that focus on simulation and targeted accuracy, differing not only in how the Navier-Stokes equations are approximated and solved, but also in the way the rotor is modeled. At the highest end of the fidelity spectrum are the so-called rotor-resolved

methods, wherein the shape of the wind turbine blade is accounted for in the CFD domain by means of complex, 3D wall-type boundary conditions. The flourishing of modern high-performance computing (HPC) architectures and solution methods is bringing such first-principle methods finally within reach.

However, the design and application level poses additional requirements on simulation technology. In fact, high-fidelity, high-generality methods are typically extremely computationally intensive because of the need to resolve small spatial and

temporal scales, at the same time as they represent a very large structure and even larger fluid domain over a long time, spanning many orders of magnitude in both time and space, which hinders their use for repetitive simulations. As a result, methods are being developed to reduce cost while retaining accuracy. For example, Fourier-based methods assume that both the aerodynamics and the structural dynamics fields can be expressed as a superposition of a few harmonics, such as in the non-linear harmonic method of Horcas et al. (2017) for aeroelastic simulations, or the imposed-motion harmonic

balance computations of Howison et al. (2018). Reduced basis approximations based on the proper orthogonal decomposition and similar methods (Ali et al., 2017; Fortes-Plaza et al., 2018) are also promising for achieving numerical efficiency and accuracy. However, the state of the art is still far from allowing the use of first-principle methods in repetitive industrial-level processes. Therefore, there is a need also for faster methods that can support massively iterative tasks, such as the ones necessary for design, the exploration of the solution space, optimization, and uncertainty quantification

(Dimitrov et al., 2018; Murcia et al., 2018; Bortolotti et al., 2019b). The challenge here is to develop methods that are not only fast, but also accurate enough and with the necessary range of validity. The use of excessively simplified methods can lead to sub-optimal solutions and may miss relevant couplings, in turn affecting performance, lifetime, and even safety. The high-fidelity CFD codes that rely on massively parallel, high-performance computing can model many of these complex features of the flow and both artificial intelligence and machine-learning methods can be used to extract these

features for use in lower-order, design-oriented models. This topic is discussed at length in Section 9.

### 5.3 Experimental capabilities

Experimental testing is another key enabler of past and future progress. In fact, measured data are necessary to verify any design choice, and indispensable for validating simulation models. The wind energy research community has over 30 years of experience developing and conducting advanced measurement campaigns on wind turbines, including the sharing of

open data sets and their collaborative analysis. The first experiments were performed in the late 1980s and beginning of the 1990s in the United States, Japan, Denmark, United Kingdom, and the Netherlands (Schepers et al., 1997), on stall-regulated turbines with small diameters ranging from 10 to 27 m. At that time, research was focused on understanding 3D



flow effects at the blade root that, causing an increased maximum lift, often led to excess power output and failures. The analysis conducted within IEA Wind Task 14 and 18 (Schepers et al., 2002) confirmed the hypothesized explanation of

this phenomenon: centrifugal forces cause a radial pumping of the flow that results in chordwise Coriolis forces, which have the effect of alleviating the adverse pressure gradient along the boundary layer; in turn, delaying stall. However, most of the available computational models were limited in their ability to simulate turbulent inflows, hindering the matching of numerical results with measurements.

This problem was addressed in the year 2000, when the same 10 m turbine tested in the field was installed in the National

Aeronautics and Space Administration's (NASA's) Ames wind tunnel, which features a 24.4 m-by-36.6 m cross section. A blind test with several codes showed in general a wide spread of the results, with the exception of one of the 3D CFD codes that–in a breakthrough moment for rotor CFD–obtained a close agreement with the measurements. The data set was shared within IEA Wind Task 20 (Schreck et al., 2008), and thoroughly analyzed in the period 2003–2007.

In the same period, a European consortium of 10 institutes and universities from six countries supported by European

Union funding built and instrumented a three-bladed wind turbine with a 4.5 m rotor diameter. In 2006, the rotor was tested in the Large Low-speed Facility of the German Dutch Wind Tunnel in the Netherlands, which features a 9.5-m-by-9.5-m open test section. Besides pressure measurements at several positions on the blade, this experiment also provided detailed flow-field measurements obtained with particle image velocimetry (PIV) around the turbine and in its wake. The data set led to several model validation rounds within IEA Wind Task 29 Phase 1, 2, and 3 (2007–2018) (Schepers et al., 2012;

Boorsma et al., 2018).

These two wind tunnel experiments provided excellent data for the validation of models at all fidelity levels and have contributed considerably to the improvement and further development of models. However, they also suffered from severe limitations: a small size of the models, which implies different Reynolds regimes than full-scale turbines; blade designs that were not representative of actual ones; rigid rotors, which were not modeling aeroelastic effects, and, most importantly,

steady uniform inflows.

To address these limits, from 2007 to 2010, a consortium comprising Vestas, Siemens, LM, Dong Energy, and the Technical University of Denmark (DTU) performed the DanAero experiments in Denmark (Madsen et al., 2010a, 2010b; Troldborg et al., 2013). Pressure measurements at four radial positions were obtained on the 2 MW NM80 turbine with an 80 m rotor diameter. In addition, 60 flush-mounted surface microphones were installed on the blades to measure the high-

frequency pressure fluctuations in the boundary layer, thereby providing experimental data for studying transition and aeroacoustic sources. To eliminate the effects caused by changes in the angular velocity of the rotor, the turbine was operated at constant speed. In 2018, the DanAero project partners agreed to share the experimental data, including the NM80 turbine information, with members of IEA Wind Task 29 Phase IV (Schepers et al., 2021). This IEA Wind task has





just ended but work with the same data set will continue in Task 47. Campagnolo et al. (2016) pioneered the inclusion of
closed-loop controls and aero-servo-elastic considerations in the design of scaled models, expanding the scope of wind
tunnel testing beyond aerodynamics. Nowadays, wind tunnel tests are extensively used to gain a better understanding of
wake effects, to validate simulation tools, and to help develop novel control strategies (Bottasso and Campagnolo, 2021).

Notwithstanding the accomplishments of these various efforts, high-quality data ranging from full-scale wind plants and
individual turbines to their components is today largely unavailable, especially for representative contemporary designs.
This is not only a measurement technology issue, but also a data availability problem.

In fact, measurements at full scale are still difficult to obtain, especially for the aerodynamic flow field. Although lidar
technology is making fast progress (Newsom et al., 2015), we are still far from the ability of obtaining high-resolution,
full-field measurements of flows generated by gigantic rotors in the field, or offshore when a floating turbine is rocking in
heavy seas. The range of measurements that is necessary for achieving a comprehensive understanding of all physical
effects and interactions at play in a modern wind turbine is staggering. It includes dynamic deformations of gigantic
structures due to turbulent flow fields–spanning from the millimeter to the kilometer scale–with impacts on noise and low-
frequency vibrations. Additional measurements are necessary to capture effects that go beyond the classical engineering
aspects and concern crops, climate, wildlife, and population. Clearly, such a wide range of measurements requires dedicated
specific technologies and procedures, which are still not available to cover the full spectrum of needs. Even here, size poses
unique challenges: installing sensors on extremely large structures may require special equipment and can be costly.
Additional costs are generated by the downtime that may be associated with instrumentation and data gathering operations
on production turbines. See Section 9.7 for further discussion of instrumentation needs.

Data availability can also be a challenge. In fact, data should not only be accurate, but also complete, abundant, and relevant
(i.e. representative of the state of the art). Lack of high-quality, open-source data poses a crucial conundrum, especially in
the   academic   field:   the   most   relevant   and   useful   data   are   typically   owned   by   industry   and   are
therefore–understandably–limited by intellectual property (IP) rights and confidentiality. Typical restrictions include
airfoils, details of the structural design, and implementation of the control system. If the goal is the validation (for example)
of an aeroelastic solver, the lack of data with those attributes will prevent valid conclusions. The typical palliative is to
conduct code-to-code comparisons based on representative wind turbine models; for example, as done in the INNWIND
(INNWIND.EU, 2014) and AVATAR (Schepers et al., 2018) projects. Code-to-code comparisons can be valuable when
the codes have gone through rigorous independent code verification and can serve as validation when one code is of higher
fidelity and has itself been validated. Code-to-code comparisons, wherein the codes have equivalent models and numerical
methods, can only reveal if the two methods are in agreement, but not if they are accurate representations of reality.





In conclusion, the lack of data (or access to data that are not completely relevant) hinders understanding of the physics and
validation of the design tools and ultimately slows down progress.

**5.4 Gaps**

Progress in wind turbine aeroelastic modeling accuracy and expansion of the range of applicability for larger and more
flexible structures is necessary on several fronts, and many gaps need to be filled. Although this review cannot hope to
discuss them all, certain gaps stand out.

First, the modeling and simulation arena presents some pressing needs. Today, BEM is the wind energy workhorse,
supporting all repetitive tasks necessary for design, control optimization, load analysis, system identification, uncertainty
quantification, certification, and so on. It is probably safe to say that all turbines available on the market today have been
designed and verified using some form of BEM modeling of their aerodynamics. While modern BEM implementations use
a number of correction sub-models to account for various phenomena, the theory is still based on assumptions that are
challenged by modern and future turbines. For example, highly curved and flexible blades deform in 3D space, challenging
the very existence of a rotor disk plane. Similarly, turbines can operate in yawed conditions to implement wake-steering
wind plant control (Fleming et al., 2017), challenging the assumption of axial flow. Going beyond BEM, while retaining
its desirable features and moderate computational cost, includes learning how to better use 2D airfoil sectional
characteristics (e.g. with more effective corrections for unsteady phenomena like dynamic stall), moving toward a full 3D
aerodynamic design.

At the other end of the spectrum, first-principle, blade-resolved methods are being developed. Although these methods do
not suffer from the potential limitations of the lower-fidelity approaches, they are not only enormously expensive but also
a long way from being usable for routine and comprehensive evaluation of the full range of inflow and operating conditions,
which is required for design. As a consequence, they are still confined to a few specialized research labs or to the design
of specific details (e.g. tips, winglets, aerodynamic add-ons, root connections, or other difficult regions characterized by
complex 3D flows). Further, there appears to be a slow-paced industrialization and standardization of high-fidelity
methods, probably because of cost and hardware requirements, but also because of a lack of dedicated training and
education.

Similar needs are present for the structural dynamics aspects of the problem, wherein higher-fidelity methods are needed
that can model generic geometries with minimal assumptions, exactly accounting for large deflections and the couplings
induced by the use of anisotropic composite materials.

In addition, for both aerodynamics and structures, it is desirable to have models with a full range of fidelities capable of
supporting all necessary simulation needs at the appropriate level (Figure 6). It does not seem useful to just repeat the





frequently cited aphorism that all models are wrong, but some are useful; what is necessary–and still lacking–is a complete
understanding of the limits of the models that are useful in wind energy. Clearly, this applies to the full range of model
fidelities, as even higher-fidelity models have limits.

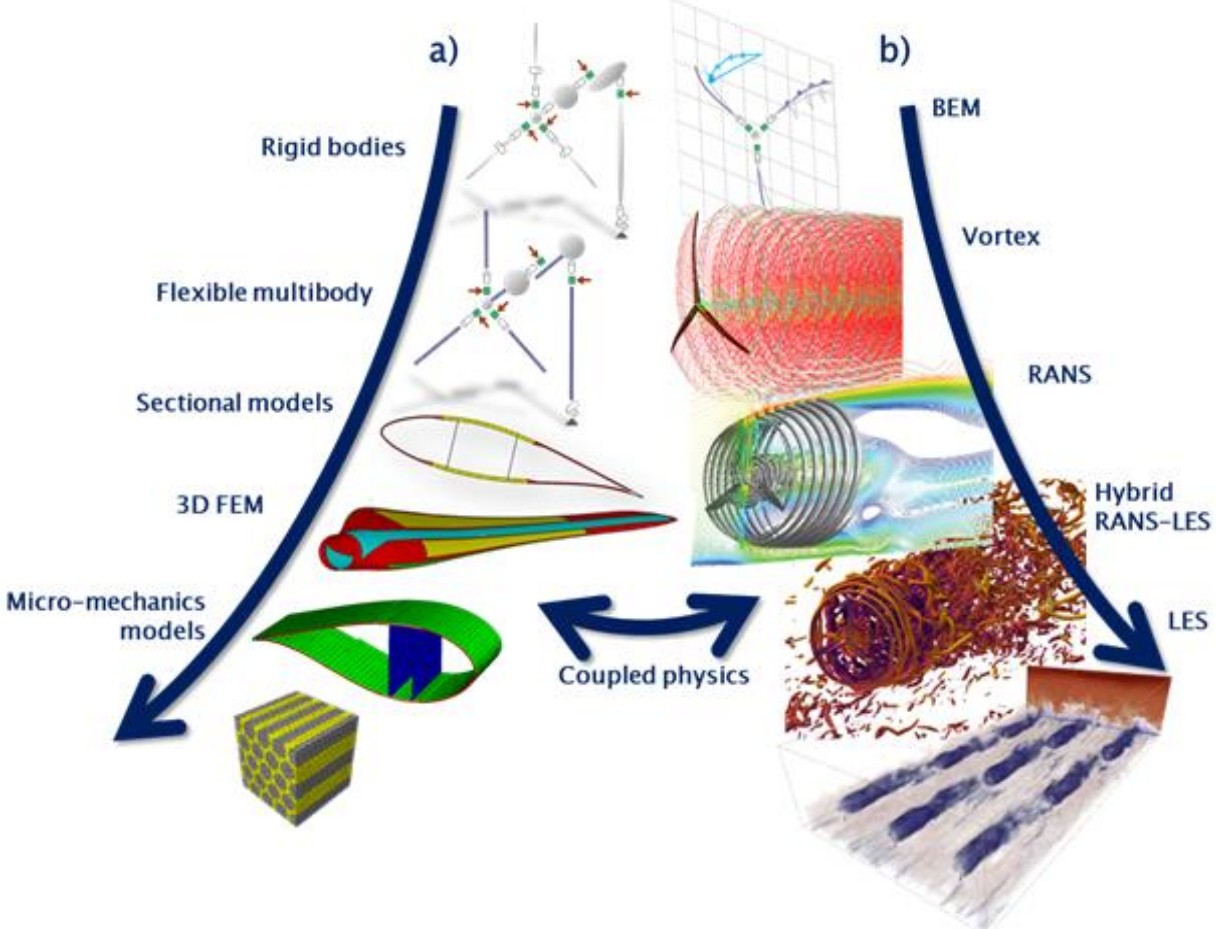

**Figure 6: The simulation of wind energy systems requires a complete and coherent hierarchy of structural (a) and aerodynamic
(b) models, covering a tremendous range of scales in space and time (from millimeters and microseconds for boundary layers
and micromechanics, all the way to thousands of kilometers and days and beyond for atmospheric mesoscale processes). Use
cases range from very high-fidelity simulations based on first principles and minimal assumptions to coupled multi-physics
transient problems to expensive repetitive tasks, such as optimization and uncertainty quantification. The validation of this wide
range of methods remains a colossal challenge, which is of crucial importance for the advancement of the field. (Figure credits
TUM Wind Energy Institute, DTU Wind Energy, NREL)**



As the physical processes involved in modern wind turbines become more complex and interrelated, the design process also increases in complexity, and calls for the integration of optimization strategies at an early stage. Numerical optimization has traditionally relied on low- to mid-fidelity models when applied to wind turbines (Bottasso et al., 2013, 2014b; Zahle et al., 2015; Bortolotti et al., 2016). However, there is a clear need to include higher-fidelity models in automated design frameworks, eventually creating multi-fidelity platforms capable of combining efficiency with accuracy.

Accuracy is necessary to ensure that all relevant couplings among the various fields and design choices are captured, whereas efficiency is necessary to explore the solution space in turnaround times compatible with industrial processes. Additionally, multi-fidelity platforms allow the design procedures to zoom in on specific details and zoom out to speed up the calculations where possible. A variable level of fidelity also helps open up the range of design variables. For example, most optimization approaches use existing airfoils, yet it would be better to perform a genuine, full 3D design of the blade.

While some initial efforts in this direction have been taken (Bottasso et al., 2014c; Dhert et al., 2017; Madsen et al., 2019b), credible, industrial-level, free-form design of flexible blades has yet to be achieved.

At the application level, SIV and VIV stand out as problems for industry, wherein achieving physical understanding of the driving phenomena, appropriate models, simulation tools, and mitigating solutions is needed. SIV is caused by the negative aerodynamic damping generated in deep stall conditions (Zou et al., 2014). This problem had already been encountered in

some early stall-regulated rotors, leading to considerable research efforts in the 1990s (Thirstrup Petersen et al., 1988). VIV involves a complex interaction between the shedding frequencies of the flow and the natural frequencies of the structure, leading to limit cycle oscillations of standstill blades (Horcas et al., 2020) and towers (Viré et al., 2020), which can result in sustained vibrations for extended periods of time. Although VIV phenomena have been studied for decades (Griffin et al., 1973; Williamson and Govardhan, 2004), more fundamental research on the phenomena in wind energy

systems is still necessary. In fact, VIV in wind turbines differs from other classical vortex-induced oscillations because of the Reynolds regime and the complex 3D flows that are involved. The urgency of addressing SIV and VIV problems is because of the design limitations that they may impose on the large and flexible rotors of the future.

At the experimental level, there is a pressing need for open data sets from highly instrumented rotors with modern (slender, highly flexible, curved, coupled) characteristics. Data should be available at the following various scales:

- At full scale in the field because it represents the actual situation where a turbine operates. The variations in the ABL now experienced by modern turbines can only be captured at full scale.
- At a reduced scale in the field, as, for example, at the Sandia National Laboratories' Scaled Wind Farm Technology facility (Berg et al., 2014) or the forthcoming complex terrain WINSENT test site (Letzgus et al., 2020). In fact, scaled rotors reduce costs, simplify measurements, and have limited or no IP issues as they do not

represent real products, but still operate in a realistic atmospheric inflow. However, reduced-scale experiments will miss some of the critical interactions with the structure of the ABL.





- At an even smaller scale in the wind tunnel, because, notwithstanding the limitations intrinsic in very small form factors, the inflow conditions are known, controllable, and repeatable, detailed measurements are possible, layouts and operating conditions are readily changeable, and costs are contained (Schreck, 2008, 2002; Bottasso et al., 2014d; Klein et al., 2018; Bottasso and Campagnolo, 2021).


In general, the use of scaled models, either in the field or in the lab, is extremely appealing for basic research because of the reasons noted earlier. Recent studies (Canet et al., 2021; Bottasso and Campagnolo, 2021) have shown that gravo-aero-servo-elastic effects can be considered by using suitable scaling laws, although various compromises must typically be accepted. Wang et al. (2021) expand this analysis by considering the effects of scaling on wake behavior; even in this case concluding that properly scaled models can produce highly realistic wakes. However, further research is needed to include hydrodynamic effects, and to better replicate the inflow at scale in various and variable atmospheric and terrain conditions.


These three approaches−full scale, scaled field, and lab testing−have complementary characteristics. Although none are perfect and each presents unique challenges, together they can provide for a comprehensive view of the problem and work in synergy to deepen knowledge, improve models, and validate simulation tools. Such an approach is also discussed in Section 9.5.


Similar needs to improve and validate models are present at the component level, from materials, airfoils, and complete blades to drivetrains and all other subsystems.

### 5.5 Future enablers

From the present discussion, there is a long list of desired steps that would help propel wind energy science and technology forward. Within the many possibilities, the authors would like to highlight two main achievements that would leave a mark: the open turbine data set and the simulation chain of variable fidelity.


### 5.5.1 The open turbine data set

The wind energy scientific and technical community needs a truly comprehensive and completely open experimental data set that represents modern configurations to enable validation of the modeling tools and scientific understanding of the underlying physics (see Section 9.5 for an in-depth discussion). In 1993, NASA launched the UH-60 Airloads Program, based on the bold vision of generating "comprehensive, accurate, documented airloads data … that will have long-term value and timely, widespread accessibility so that the rotorcraft community can increase their understanding of rotor behavior, refine and validate their analysis tools, and design improved rotorcraft" (Bousman, 2014). To this end, a Sikorsky UH-60A military helicopter was highly instrumented (Figure 7) and flown in an extensive set of conditions. Under the funding of NASA, the U.S. Army, the Federal Aviation Administration, and the Defense Advanced Research Projects







Agency, the resulting data set led to numerous industry/academia collaborations, 26 dedicated workshops between 2001 and 2014, the education of scores of students, and numerous published papers that generated hundreds of citations (Bousman, 2014) that continue to increase. It might be surprising that the data of a military helicopter was shared with the academic community, but this bold step had an enormous scientific and educational impact on the field. Wind energy also

needs comprehensive, accurate, documented data sets that will have long-term value and timely, widespread accessibility so that the community can increase their understanding, refine and validate their analysis tools, and ultimately design the wind turbines of the future. If it was done for a military helicopter, it should be doable for wind turbines and plants as well.

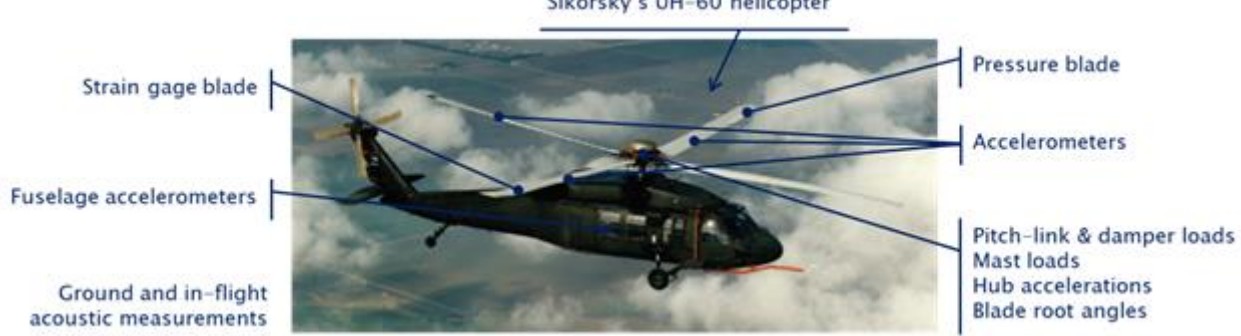

**Figure 7: A highly instrumented UH-60A flying over the San Joaquin Valley in California during the UH-60A Airloads Flight Test Program. The data set collected in these experiments led to major advances in the modeling and simulation of rotorcraft vehicles. Wind energy science needs similar high-quality, fully open-access data sets, which are indispensable for validating the models and tools that will enable the design of the wind turbines of the future. (Photo from https://rotorcraft.arc.nasa.gov/Research/Programs/uh_60_program.html)**

To mitigate the effects caused by the lack of available data sets, the technical community has recently developed joint
efforts aimed at defining open reference systems, both at the turbine and plant levels (Ning, 2019). These efforts have also spurred the definition of a common terminology and description of the key components of such systems to encourage the interfacing of codes and exchange of data.

### 5.5.2 The simulation chain of variable fidelity

The wind energy community needs a comprehensive set of validated methods of variable fidelity, spanning from first-
principle, high-fidelity models to engineering models for repetitive tasks. Such an effort has already started. For example, the National Renewable Energy Laboratory (NREL) develops and maintains a suite of tools that include, among others, OpenFAST, an aeroelastic simulation environment for wind turbines (OpenFAST); WISDEM, a system design framework (WISDEM, 2021); FLOw Redirection in Steady State Systems Engineering (FLORIS), a controls-oriented engineering



wind plant flow model (NREL, 2021); Simulator fOr Wind Farm Applications (SOWFA), a large-eddy-simulation actuator-line-model tool for simulating wakes in turbulent inflows; and ExaWind, a multi-fidelity CFD package, spanning Reynolds-averaged Navier Stokes (RANS) plus actuator disk to fully blade-resolved, multi-turbine computations, targeting exascale computational platforms (Sprague et al., 2020). At DTU, the HAWC2 code (Larsen and Hansen, 2015) has now been developed into a framework wherein the structural solver can be coupled with five different aerodynamic models of varying fidelity, ranging from BEM, hybrid BEM and lifting line, actuator line, and vortex wake to body-resolved CFD with the EllipSys3d solver (Sørensen, 1995; Michelsen, 1994, 1992). These and similar efforts should be further expanded to cover the full range of needs, including controls, optimization, system identification, model adaptation, uncertainty quantification, life cycle assessment, cost and impact models, and the many other tools that are necessary to support wind turbine analysis and design. In addition to physics-based models, data-driven methods, modern machine-learning, and data analytics tools should also be included, as appropriate.

Such a comprehensive simulation chain would benefit from similarly comprehensive and completely open data sets, accelerating knowledge, empowering the turbine designs of the future, increasing the acceptability of wind energy by reducing its impacts, and contributing to the education of the next generation of engineers and scientists. For further discussion of modeling at various fidelity levels, see Section 9.2.

## 6. Wakes

Wind plant developers lose revenue when observed power output underperforms preconstruction energy estimates that rely on inaccurate wake models (Veers, 2019b; Porte-Agel et al., 2020). Turbulence and wind speed gradients across wind farms are also known to impact turbine loads and reliability in ways that are poorly understood (Sathe et al., 2013). Most wake models are extremely simplistic and highly uncertain, suggesting that innovative wake-modeling improvements could yield increases in owner's profits. With wind plants costing typically half a billion U.S. dollars and growing, the risk associated with using unvalidated production, reliability, and loss models is too high for many projects. However, improved and well-tested wake models that can accurately estimate individual and aggregate wakes over a range of atmospheric conditions, terrain, and vegetation could revolutionize the processes for wind plant design and operation.

Improved modeling of wind turbine wakes could have a large impact on multiple aspects of wind turbine design and wind plant layout. Of all the categories that factor into the uncertainty of preconstruction energy production estimates for wind plants, wakes represent the largest contribution (Fields et al., 2021). It is well-known that wake impingement reduces power output and increases dynamic loading on wind plants, but the complex physics associated with these processes are challenging to model; therefore, accurately predicting impacts is limited. Aside from current concerns, future opportunities such as wake steering, wind plant control, and optimizing wind plant layout based on local winds and terrain are only





realizable with more sophisticated models. Questions at scales larger than the wind plant, such as wind plant blockage effects, impact of upstream wind plants on downstream wind plants, agriculture, and weather could all be considered with a more accurate wake modeling capability.

Trusted models ranging from limited-physics engineering tools to highly detailed physics-based models could enable capabilities such as optimization of wind plant layout and coupled control of wind turbines to maximize wind plant production. Wake losses in a typical wind plant are estimated to be 10%, and the uncertainty in industry wake model predictions is 20%–50% (Barthelmie et al., 2009; Clifton et al., 2016; Lee and Fields, 2021). Reducing uncertainty in estimated wake losses could reduce the risk in otherwise marginal projects to make them viable. Low profit margins of typical wind projects do not allow these marginal projects to be built when the uncertainties are too high. If wake losses can be cut in half (5%), improving capacity factor by 2.5% and lowering the uncertainty of the models to 2%, the owner of a 1 GW wind plant could expect an additional 8.75±3.0 million U.S. dollars each year. Scaling this up to the hundreds of gigawatts expected to be installed annually in the future suggests an annual impact of billions of dollars.

These immediate financial gains are only the start of what could be achieved. Trusted models could be used to examine new strategies of designing wind turbines and wind plants (see Section 4). Innovative, large downwind turbines and turbines using dynamic wake control that are currently too risky to develop and deploy could be thoroughly investigated to develop confidence in their potential. However, without understanding the detailed physics and gaining confidence in new validated models, the ability to explore such novel concepts will be limited.

### 6.1 Wake physics

The reason for the shortcomings in our understanding and modeling of wakes arises from the complexity of wake development and evolution and its coupling to the local atmospheric boundary layer and surface (Vermeer et al., 2003; Brower, 2012; Barthelmie et al., 2013; Emeis, 2014; Veers, 2019b). Figure 8 shows the processes by which turbine wakes form, interact, impact downstream turbines, and merge. The initial wake develops when the blades extract momentum from the inflow to produce the forces necessary to turn the rotor (see Sections 3.3 and 3.4). The unsteady nature of the wind inflow implies that the initial wake development will also be unsteady and highly dependent on the nature of the inflow. As the wake moves downstream, its evolution is affected by the turbulence generated by the wake itself (low momentum and rotating flow), interaction with the surface, whether it be the ground or waves, breakdown of the near-wake, and interaction with the atmospheric boundary layer. The turbulence in the atmospheric boundary layer surrounding the wake has a profound impact on this process and is affected by its stability characteristics (Hansen et al., 2012; Machefaux, 2016; Xie and Archer, 2017). For example, wakes can persist for very long distances in a stable, low-turbulence boundary layer associated with offshore wind plants. Wake losses are thus typically greater in large offshore wind plants than in their land-based counterparts (Göçmen et al., 2016; Bodini et al., 2019).



As the wake moves downstream, additional complexity is introduced including merging of individual wakes and the impact of wakes on downstream turbines. The interaction between these wakes and the wakes generated in subsequent downstream turbines further complicates the flow within the wind plant. This aggregated flow interacts with the atmospheric boundary layer above and around the wind plant and becomes a source of momentum (Calaf et al., 2010). Surrounding winds can help reenergize the flow to offset the losses incurred by turbine energy extraction. As the flow exits the wind plant, the

combined wakes travel downstream where they impact the atmosphere and surface as well as any downstream developments (e.g. towns, cities). Of particular interest is the impact of the wind plant wake on downstream wind plants (Nygaard, 2014), which is the subject of another article (Lundquist et al., forthcoming). To further add to this complexity, the slowing of wind through the wind power plant causes some of the flow ahead of the plant to divert around it, a process known as blockage. Because blockage directly impacts revenue, it is an area of active research (see, for example, Branlard

and Meyer Forsting, 2020).

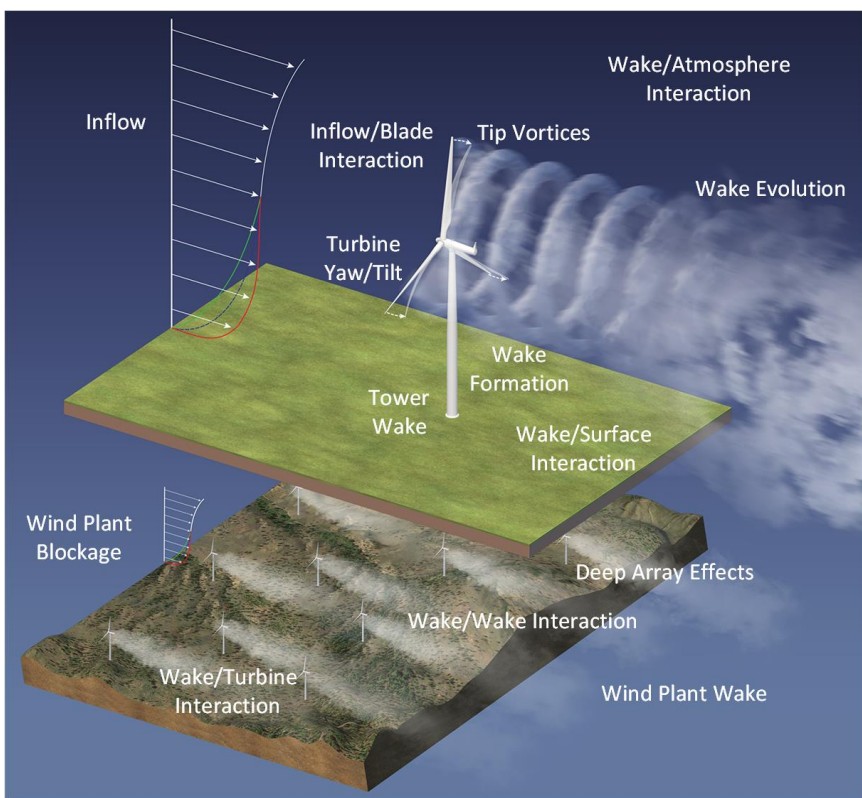

**Figure 8: The inflow interacting with the wind turbine influences the wake produced. Wakes interact, merge, and impact downstream wind turbines. (Image credit Besiki Kazaishvili, NREL)**



To better understand these complex processes, we must both measure and simulate these flows. As might be expected, field measurements and computer simulations are challenging because of the unsteady inflow and complexity of the wake flow. In the field, extensive data records are needed to produce statistics with low uncertainty. Instrumentation capable of such long-term measurements with sufficient time and spatial resolution at large scales is particularly difficult to develop and deploy. Similarly, simulations run with many different inflow conditions that characterize the wakes statistically and reduce uncertainty take substantial computational resources. See Section 9 for further discussion of these topics.

## 6.2 Limitations imposed by limited understanding of the wake

As stated earlier, the knowledge that is needed to improve models to address wake-related opportunities is going to be hard to develop. Measurements are challenging, computations are difficult, and ensuring simulation accuracy will require extensive validation. With the current understanding of wind turbine wakes and the limited capabilities of existing modeling tools, wind-plant wake models are less accurate (Lee and Fields, 2021). This uncertainty restricts the ability to exploit wakes when addressing wind plant layout and wind plant control for grid services. For example, innovative approaches to wind plant design are limited by the inability to trust the model results to commit to the designs they suggest. Similarly, increasing the use of wind plant optimization with AI approaches will be hindered if the simulation data provided to train those models are not trusted. To address these limitations, we must increase knowledge in these areas so that models with greater certainty can be realized.

## 6.3 Gaps

Gaps in our understanding and ability to model wakes at all scales range from wake generation to wind plant wake evolution—with some of these gaps common at all scales (Veers et al., 2019a). Modeling capabilities are still limited because of the physics that need to be modeled at multiple scales. In addition, there is a lack of experimental data available to validate the codes and quantify the modeling uncertainty (Moriarty et al., 2020). Validation data are missing because many of the necessary experiments are complex and expensive to conduct. The measurements needed for validation often require instrumentation that is not yet fully developed or does not exist. Improved instrumentation measurement capabilities will require significant focus, cross-industry collaboration, and resources, and is one of the main recommendations of this article. Any measurements that are made will also require rigorous investigation to determine their uncertainty so that they are useful for validation purposes. See Section 9 for further discussion of measurement and modeling challenges.

Starting at the smallest scale, the processes involved with the initial development of the wake are not entirely understood. The wake initially arises from the inflow interacting with the turbine blades. The flow over the blade produces flow-aligned and tangential forces that cause thrust on the turbine and blade rotation, respectively. These forces result in the development of the wake including both a reduction in the flow momentum and rotation in the flow. Some specific features in wake development include the rollup of vorticity shed from the blade into tip and root vortices and the interaction of the tower



and nacelle wakes with the wake from the blades. The characteristics of the near-wake depend on inflow turbulence,
turbine operating conditions (e.g. tip-speed ratio, turbine yaw), unsteady blade loading and the resulting blade movement
(aeroelastics), and wave-induced motion for floating platforms (Sørensen et al., 2014). Greater comprehension of these
different processes, as well as their influence on wake development including which of them have the largest impact under
different operating conditions, would aid in the modeling of wake development. Currently, not many experiments that
capture the inflow, turbine response, and wake behavior have been done (Hassanzadeh et al., 2020, 2022).

As the wake produced by the turbine propagates downstream, it evolves (Crespo and Hernandez, 1996). The individual
features in the near-wake break down and merge into a single wake. The wake interacts with the atmospheric flow, and
this interaction can affect wake propagation direction, size, and unsteadiness. As with the near-wake, the evolution strongly
depends on the turbine operating conditions (e.g. loading, yaw, tilt), as well as interaction with the surface, which can be
complex (e.g. variable terrain and waves). The physical processes occurring within the wake (e.g. diffusion, wake-
generated turbulence) add to the complexity. All of these processes yield wake behaviors such as meandering and varying
decay rates. Increased understanding of these processes and their interaction would lead to better wake modeling that could
allow wind plant owners to benefit from techniques such as wake steering or enhanced wake diffusion.

Depending on the local topography and atmospheric state, the trajectory of wake propagation can vary considerably
(Medici and Alfredsson, 2008; Howard et al., 2015). This phenomenon is known as wake meandering and can cause large
lateral variations in the location of turbine wakes as a function of downstream distance from the turbine. Because of the
lateral motion, the wakes from adjacent turbines can interact, mix, and generate additional turbulence. The interaction of
these mixed wake structures with the background flow is currently not well-captured in current engineering models, thereby
requiring high-fidelity, microscale LES models. Depending on the atmospheric stability state, the computational mesh
requirements for high-fidelity microscale simulations can quickly become very expensive, remaining a significant
challenge for researchers.

For a given wind plant layout and prevailing wind direction, certain turbines will end up shadowed by the turbulent wakes
generated by the upstream turbines. In waked situations, these turbines operate in a velocity deficit region and, therefore,
produce less power than their upstream counterparts. Furthermore, these turbines experience significantly higher fatigue
loads in wake conditions due to several factors. The wake breakdown process results in higher turbulence (wake-added
turbulence) (Quarton and Ainslie, 1990) than the ambient flow, resulting in larger fluctuations of the local aerodynamic
loading along the blade span. In addition to full shadowing, partial shadowing, where only part of the rotor disk operates
in the upstream wake, results in large lateral load imbalances (Scott et al., 2020; Sun et al., 2020). While there are several
models that can adequately account for the impact of wakes on power production of the downstream turbines, the impacts
on fatigue loading are not well-understood (Shaler et al., 2019). Accurate prediction of fatigue loads will require high-
fidelity, fluid-structure interaction models coupled with an atmospheric boundary layer model that can capture the turbine



structural response to nonlinear variations in the local aerodynamic forces that are in turn influenced by the blade deformations.

Wind turbines in a wind plant act collectively to create complex atmosphere interactions. These interactions are becoming
increasingly important as both plants and turbines continue to grow in scale. Blockage, an upstream impact on the incoming
wind resource created by a plantwide induction zone, is one of these phenomena. Blockage slows the incoming wind speed
by up to 3% (Bleeg et al., 2018). It can also create localized accelerations of flow around the wind plant, leading to
downstream wind turbines located on the edge of the plant producing more power than those closer to the front edge.
Another impact of interest is commonly known as the deep-array effect (Frandsen and Christensen, 1994; Barthelmie and
Jensen, 2010; Nygaard, 2014). The deep-array effect is intertwined with the growth of an internal boundary layer over the
wind plant and often appears with plants containing more than three rows. The internal boundary layer growth results in
lower wind speed above the plant and, with less resource from which to draw, the wind speed within the wind plant is also
reduced. Both blockage and the deep-array effect are closely coupled to more dynamic behavior, such as the creation of
gravity waves (Allaerts and Meyers, 2017). Like mountains, large wind plants can trigger waves over the plant that will
have variable impact on the resource above the plant and affect the power production of different rows in a process that is
currently unknown. All of these effects are highly dependent on plant layout, turbine size, operation, wind direction, speed,
and atmospheric stability or turbulence. The dependency on each of these variables on physical processes remains
unknown, necessitating further observations and research.

One challenge with modeling the large-scale impacts of wind plants is that many models that predict plant atmospheric
interactions arise from single-turbine models of induction and wakes. Because these models do not account for the
interaction of multiple machines, they must be numerically coupled, which often involves complex superposition (Göçmen
et al., 2016). Such superposition must be carefully calibrated so as not to minimize the overall impact or doubly count the
contribution from a single phenomenon. It also highlights the importance of more complex, nonlinear, coupled models for
validating simpler tools (see Section 9.2).

Questions remain about the overall environmental impact of wind plants on their local surroundings. Some observational
and simulation studies have shown that wind plants increase mixing of the ABL between the surface and rotor plane (Baidya
Roy and Traiteur 2010; Rajewski et al., 2016; Moravec et al., 2018). At night and under stable conditions, this mixing may
lead to increased temperatures at the ground. It remains to be seen if this is a negative or positive impact or if there is a
definitive influence on crop yields. More research in this area is required, particularly at larger scales, as the wind energy
industry would like to ensure that environmental impacts are minimal.



## 7. Offshore wind technology

As countries around the world look to expand renewables in their energy portfolios, the need for more widespread wind deployment will increase. The abundant wind resource on land, which has become a mainstay energy source, may not be able to achieve the scale needed for deep renewable penetration, especially near some coastal cities (as a result of land and transmission constraints). Offshore wind energy, which consists of wind turbines installed in oceans, seas, and lakes, opens the door to a significantly larger resource.

There are several advantages to placing wind turbines offshore, including stronger, more consistent wind with less intrinsic turbulence, fewer transportation constraints (allowing for larger turbines), and shorter distances between energy generation and the consumer (allowing for better grid infrastructure). However, there are also significant challenges to overcome to make offshore wind a significant contributor to the renewable energy sector, including getting the cost on par with land-based wind systems, developing the industrialization infrastructure, and addressing social and environmental issues.

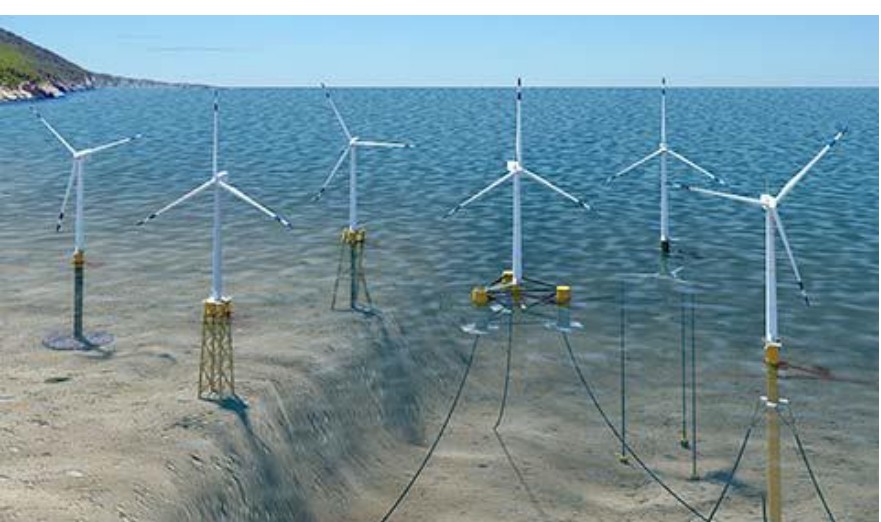

**Figure 9: At water depths up to 60 m, offshore wind turbines can be installed on fixed-bottom foundations. Deeper than approximately 60 m, floating substructures are needed. (Image credit Josh Bauer, NREL)**

There are two fundamental offshore wind design types based on whether they are placed in shallow water (approximately 0–60 m) or deep water (approximately 60–1000 m), as shown in Figure 9. In shallow-water regions, the substructure below the tower is fixed to the seafloor and thus referred to as a "fixed" design. A viable fixed commercial industry has already been operating in Europe, mainly in the North Sea, but that technology may need to be adapted for the different conditions elsewhere in the world (including for the possible occurrence of large tropical storms) and with consideration for each





region's industrial infrastructure (Diaz and Guedes Soares, 2020). In deeper water, it becomes too costly to build a structure attached to the seafloor, so floating designs may be used, which are kept in place through mooring lines anchored to the seafloor. Floating wind is still in a developmental stage globally and it offers greater challenges because of the increased motion created by the compliant substructure and increased coupling between the wind and wave excitation. But, if the challenges of placing wind turbines in the deep-water regions of the world can be overcome, the resource increases

significantly, opening up the possibility for offshore wind in regions off the coasts of California, Oregon, Maine, and Hawaii in the United States, in Asia, and in southern Europe. As an example, in the United States, approximately 60% of the offshore wind technical resource potential, estimated to be greater than 2000 GW, is in deeper waters that would require floating platforms (Musial et al., 2020).

### 7.1 What is needed to make progress

While the potential for offshore wind using both fixed and floating systems is vast, there are significant challenges that must be overcome—including technology advances to achieve continued cost reduction (Musial et al., 2020). Areas of focus that will help address economies of scale include light-weighting; advances in component design, manufacturing, and materials; achieving a greater understanding of the resource; custom design for the interaction of offshore wind plants with the ocean and the atmosphere; improved controls for optimal performance; and measures that consider offshore-

environment-specific issues surrounding durability, inspection, repair, maintainability, and long-term reliability.

To identify technologies with the most promise, greater knowledge of the physics associated with the technology's interaction with the environment is needed, as well as tools that can accurately represent these phenomena. While offshore wind might at first appear to simply be a merging of two well-established energy fields−land-based wind and offshore oil and gas−it presents its own new challenges because of its fundamentally different dynamics wherein the forcing from the

wind and waves requires equal consideration and coupling. In summary, design tools that can represent the physics with an accuracy sufficient to drive innovations are needed to achieve the cost levels for sustained commercial viability.

With the combined wind and wave physics, it is key to think of the system holistically, and not as separate components (e.g. the turbine, support structure, and controller). And, to achieve true optimization, the entire life cycle needs to be considered, including the impact of design choices on the cost of manufacturing, deployment, installation, O&M,

repowering, recovery, disposal, and dismantling. Some examples include how the stability of the system (fully or partially assembled) affects the installation process, the challenges regarding quay-side vs. on-site assembly, the availability of large-scale crane capacity, and how the state of the infrastructure might affect design choices, including supply chain, manufacturing, and storage. To keep costs low, the design process may also need to be adapted, it must be tractable, and it must use safety margins commensurate with the risks associated with wind energy, and not the more exacting regulations





employed in high-risk fields, such as oil and gas, wherein the consequences of a single structural failure can be great
        because of human habitation and potential environmental damage (Michel et al., 2011).

        Innovative technology pathways also need to be explored across the components to identify those that provide economic
        and performance benefits to the integrated system, including how the design affects construction, installation, and
        maintenance costs. While the main trend for turbine development is the continuous upscaling of the three-bladed upwind
design, it is conceivable that different turbine technologies and support structures can emerge. Downwind rotors, two-
        bladed rotors, and turbines with multiple rotors may offer higher value for offshore use. Wind turbine floating substructures,
        a.k.a. "floaters," are yet to be standardized and new mooring solutions, especially at shallow sites, may emerge, for
        example, with passive yaw alignment. Another new concept is the unmoored floating wind turbine. Such turbines would
        not be limited by depth. On the other hand, they could not transmit power back to shore via cables, and so they would need
to convert the electricity to some storable product, such as hydrogen or ammonia. Such design concepts are more far-
        reaching, as there are fundamental and practical issues to be resolved to make them a viable technology.

        To support this technology development, collaboration between industry and the research community is needed. In the
        following sections, we outline some of the critical research questions that need to be answered to advance the offshore
        wind industry. This discussion is organized into three topic areas concerning (1) the need to understand and accurately
represent the offshore metocean conditions that impact energy capture and system loading, (2) interaction of the wind and
        waves with fixed offshore wind designs and the associated structural response and (3) how these interactions change for
        floating technology and the associated novel dynamic characteristics.

### 7.2 External conditions, inflow, and environmental effect

        The offshore wind environment is different from that on land, affecting not just the amount of energy that a turbine can
produce, but also the loading it will need to withstand. The differences in the wind characteristics result from the various
        heating and cooling rates of water compared to land masses (because of the greater heat capacity of water and differences
        in heat transfer properties), and surface roughness effects that can differ based on the wind/wave conditions (i.e. smooth in
        low wind and waves, rough during high wind and waves). To accurately represent the offshore environment, a better
        understanding is needed of the marine boundary layer in different offshore regions, as well as the associated atmospheric
stability, vertical wind profile and wind shear, and turbulence characterization.  This information will be relevant to
        achieving more accurate resource predictions, and to create designs that are better optimized for unique offshore conditions.

        Extremes in both atmospheric and oceanographic (metocean) variables during normal conditions, but especially during
        hurricanes, typhoons, and northeast storms ("nor'easters"), and so on, need to be better understood through field
        measurement campaigns and simulation studies. Information is emerging about the unique conditions in tropical cyclones,



including the somewhat steeper wind speed profile, the cyclone-specific turbulence power spectra and coherence functions, and differing turbulence intensity and gust factors. These storms also exhibit non-stationary characteristics that are not systematically accounted for in traditional approaches for representing wind events. Fully coupled wind, wave, and current simulations on offshore wind turbines have been undertaken, but such studies are extremely demanding computationally and are expensive as has been shown for simulation of hurricane-related external conditions (Kim and Manuel, 2019) and

for simulation of loads (Kim and Manuel, 2021) as well as for a fully integrated, end-to-end software simulation suite that integrates mesoscale inputs with turbine performance (Kim et al., 2015). Similar advances related to simulations and measurements are needed to better clarify these conditions and how much they may vary between different storms.

At a larger scale across a wind power plant, the characteristics of turbine wakes offshore may also be affected by fundamentally different climate patterns. The wakes can have distinctive and evolving characteristics as they are convected

downwind from one turbine to the next, and this behavior offshore needs to be better understood, as well as how the dynamics of the floating system affects this behavior.

In addition to the complex atmospheric conditions and their variability, offshore wind turbines must contend with wave, current, and tidal (water level) loading as well. Understanding the water climate is imperative to designing an offshore system, but also how the wave climate correlates with the wind climate, e.g. the misalignment of the wind and waves, can

significantly influence the dynamic excitation of the system and what the support structure needs to withstand. Extreme conditions for the water climate include green water (water over the deck) and breaking waves, both of which will generate impact loading on the structure. Site-specific environmental contours based on structural reliability principles can aid in describing combinations of wind- and wave-related random variables using parametric and non-parametric models with different dependence models (Manuel et al., 2018; Li and Zhang, 2020). Understanding the probability and severity of

these design-driving events is critical to ensuring the safety of an offshore wind turbine.

A third component to consider offshore is the geotechnical properties of the seabed where the turbines will be built. These properties can vary significantly in different parts of the world, and even within the extent of one large wind plant. Some regions have a rocky seafloor, which can make driving piles difficult or impossible, whereas other regions are sandy, with little holding capacity. For fixed designs, the seabed properties determine which type of foundation is most workable; either

piles such as those used for a monopile or jacket, or a gravity-based foundation, wherein a large mass rests on the seafloor. And, for floating systems, seabed information is needed to assess the appropriate approach for anchoring the wind turbine. While each offshore wind project needs its own geotechnical survey to address these questions, general public information on these properties is needed to ensure that the design methodologies can accurately represent the varying conditions. We note here that geotechnical surveys are, of course, also needed for land-based wind turbine foundations; at offshore wind

sites, the issue of foundation design and soil property variability is especially acute because of the greater difficulty in





obtaining soil properties from logs and the greater costs associated with foundations offshore as a fraction of the total wind plant design cost.

It is important to also consider the impact that offshore wind will have on the environment, through all phases of a wind plant's life cycle, in order to maximize its value to the community. Different technology approaches should be considered
to address noise during construction, installation, and operation, and how the design will interact with marine life. Impacts on other industries (e.g. fisheries, tourism, aviation) that use the same offshore region must also be considered. Certain design approaches could significantly improve the usability of the region for other industries (e.g. floating designs that use smaller moorings to limit the footprint). Synergistic use of platforms for multiple applications such as aquaculture could also further increase the value of a wind power plant.

**7.3 Challenges for fixed designs**

Fixed offshore wind systems follow similar design trends to land-based systems, but they must endure the extra loading from the wave environment. A key aspect in the design is fatigue loading from the long exposure to wind and waves, which becomes an increasingly greater design challenge as turbines increase in size. Larger rotors spin at slower rates because maximum tip speeds are held constant as blade length increases. This effect drives down the blade-passing frequencies,
moving them closer to wind and wave excitation frequencies, making them susceptible to possible coalescence. To support the larger turbines, the support structure may need additional stiffness to avoid resonances with both the wave and blade frequencies. Aerodynamic damping from the operating rotor and soil will help lessen the vibrations; therefore, accuracy of the soil and aerodynamic models is important, in addition to accuracy in the wave load description.

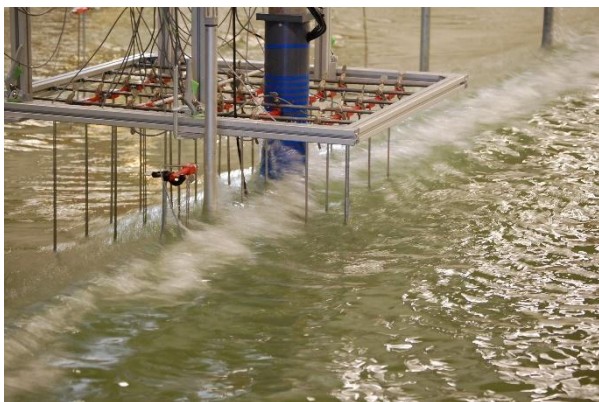

**Figure 10: Breaking wave impact at a monopile from model tests of the DeRisk project (Source: Bredmose et al., 2016)**





For severe sea states, wave nonlinearity can drive extreme loads. Natural frequencies of the structure are typically designed to be well outside the dominant frequencies of waves, but extreme conditions lead to steep or even breaking wave conditions (as shown in Figure 10) that induce high-frequency excitation into the system, known as "ringing." There is still a gap in comprehension of the hydrodynamic loading chain for efficient fatigue models and in the development of accurate

models for nonlinear 3D wave forcing and diffraction in storm waves. Other topics that have only been considered to a small degree are accurate load models for ladders, J-tubes (used to protect cables in the support structure), and other appurtenances that induce loads on themselves and the main structure. Loading from all these secondary structural elements needs further investigation.

### 7.4 Challenges for floating designs

Floating wind turbines present a further set of challenges. The floating substructure allows the entire wind turbine system to move in six different kinematic motions at any time: surge (fore-aft translation), heave (up-down translation), sway (side-side translation), pitch (forward-backward rotation), roll (sideways rotation), and yaw (rotation about a vertical axis). The extent of these motions depends on the type of floater design, the details of the mooring system, and the metocean conditions. The floating substructure options are categorized by the approach used to achieve stability: the spar (which is

a deep-draft, ballast-stabilized tube), the tension-leg platform (which is stabilized by taut mooring lines connected vertically to its anchors), and the semisubmersible (which is stabilized by buoyancy at the waterplane by virtue of its spatially separated floating members).

The rigid-body frequencies of floating wind systems are typically set to be very low to avoid wave interaction (typically 0.01–0.04 Hz) but can still be excited by non-linear hydrodynamic forces. The loading is a result of both the surrounding

wave field–which has a characteristic frequency of approximately 0.05–0.10 Hz–and the body's instantaneous position, velocity, and acceleration. These phenomena are well-understood for small-amplitude motion, wherein the approximations of linear wave theory are valid and allow separate solutions for the forces from waves and the effect of the floater's velocity and acceleration to be combined. At the next level of approximation, second-order theory offers increased accuracy, but only for the wave part. Recent research has shown that the low-frequency pitch motion can be forced by equal contributions

from the viscous drag loads and the second-order potential flow forcing (Orszaghova et al., 2021). Further, in the Orszaghova study, the structure of the low-frequency drag forcing was shown to be almost identical to that of third-order potential flow forcing. Given the complexity of the hydrodynamic loads for various floater designs, a higher-order, non-linear force model that also takes the rotor-induced motion into account is therefore recommended.

Excitation of the floating substructure results not only from the direct wave interaction, but also from the rotor loads, which

are transferred to the floater through the tower. These rotor loads, in turn, depend on the velocity and position of the system. Hence, the problem is fully coupled. Another complication for offshore wind floaters is the structural flexibility that is




inevitably associated with the larger size needed to support the larger turbines, and the desire to streamline the design to reduce cost. Although the flexible deformations may not be large, the coupling with the tower leads to a change in the global natural frequencies and coupling to the flexible floater motion. The latter can impose forcing from the waves to
global vibrations as well as damping. It is thus important to have an accurate description of these phenomena to ensure an accurate and reliable design of the full floater-turbine system.

Aerodynamics for floating wind turbines is complicated by at least two effects related to the motion of the floater. First, the floater surge and pitch motion make the blades go into and out of their own wakes, whereby the local induction is changed and presents some challenges to simpler modeling practices (as shown in Figure 11). This motion can lead to a
more rapid breakdown of the wake structure downstream and may thus reduce the wake loss. On the other hand, the platform motion may create dynamic effects in the wake that can resonantly drive the low-frequency platform modes of the downstream turbines. The wakes are further deflected upward by the permanent tilt of the rotor through the mean tower and floater tilt. These effects are yet to be properly understood and need to be incorporated into engineering models.

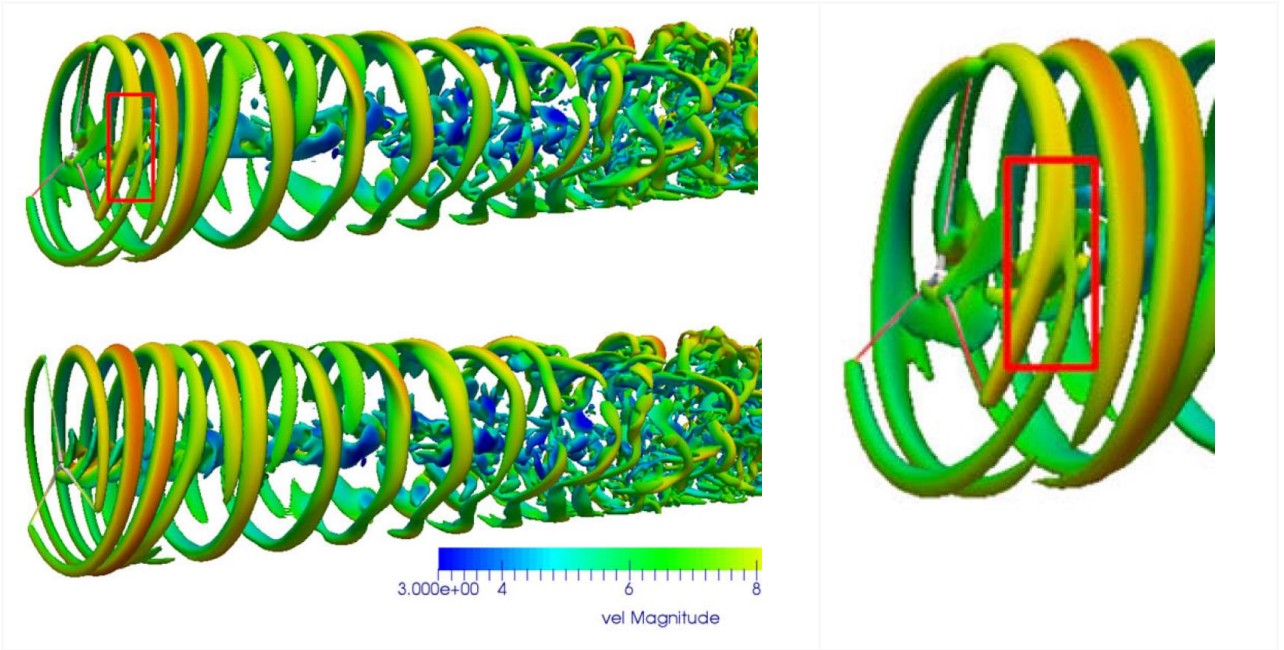

**Figure 11: Wake breakdown for a floating wind turbine. The pitching floater motion enhances the tip vortex instability through**
**the non-constant position of the blade tip between the rotor revolutions. In the top left, the rotor leans along with the wind, whereas in the bottom left, it leans into the wind. The right image is an enlarged version of the first (top left), with the red box indicating the merging of two consecutive tip vortices due to the rotor's tilt motion (Source [Ramos-García et al., forthcoming]).**



A second effect of the floater motion is the need for improved control algorithms for the turbine's generator torque and blade pitching. The rigid-body motion of the floater will cause an increase or decrease in the perceived wind speed at the turbine as it moves back and forth in the wind, thus causing the controller to have an oscillatory reaction. Unless correctly accounted for, these controller actions can induce destabilizing dynamics into the system. This ability to induce dynamic excitation, however, means that the pitch controller can also be used to damp system dynamics. Designing the control system in conjunction with the floating wind structure (a method called control co-design) can thus be used to enable lighter and more optimized floating designs. In addition, work is being done to determine if controller performance can be further improved with feedforward, wave-based laws (i.e. adjusting the turbine behavior depending on the oncoming metocean conditions) (FLOATECH, 2021).

### 7.5 The model chain: from physics to design

While gaining a physical understanding of the phenomena mentioned earlier is the first step to achieving sound and cost-efficient designs, the additional steps needed to capture the phenomena in simulation models usable for design are just as important. An offshore wind turbine system is typically designed for a lifetime of at least 20 years and is naturally exposed to a large variety of wind, wave, and operational climatic conditions. Therefore, for a given location, an assessment of historical and statistical data of the environmental conditions is needed. The statistical data, especially for extreme loading evaluation, must consider the range of joint weather variation information that might be expected. Long historical databases are required for reliable predictions of long-term loads. Stochastic simulation of the relevant variables of the coupled weather variation must be at the appropriate temporal and spatial resolutions to allow accurate load and response evaluations. Simulations are needed that consider 20-30 years of service as well as extreme (50-year) conditions for wind speed, turbulence, wave height, and so on, as well as combinations thereof, so efficient tools for stochastic simulation are imperative. Further complications that add to an already high-dimension space of random variables that define the input excitation arise from directionality of wind and waves separately as well as their relative misalignments with each other.

To bridge the gap between the physics and the need to predict 20-30 years of loads, models at different levels of fidelity are needed. High-fidelity models are often used to help better understand the fundamental physical properties and to conduct simplified parameter studies, for example, for wake propagation inside wind power plants. They are computationally expensive (requiring supercomputer resources), and the output requires substantial post-processing to extract the physical results. Medium-fidelity models run at speeds comparable to real time and are used for simulating the time series of stochastic/deterministic responses to specific inflow and wave fields. These models are used for the design verification, wherein, for example, 20,000 10-minute simulations are routinely treated, and need to be verified/calibrated against the high-fidelity models. Finally, low-fidelity models are also required for design exploration. These models are calibrated against the medium-level engineering models and often make use of frequency-domain approaches and linearization to allow for considerable computational speed up (Lim et al., 2021; Nguyen et al., 2019). An alternative





approach is low-dimension surrogates or reduced-order representations of the complex and stochastic external conditions and loads.

Although there are many epistemic uncertainty challenges associated with imperfect models for our materials, aeroelasticity, aerodynamics, and hydrodynamics, it is the external conditions and their own aleatory (inherently stochastic and therefore irreducible), as well as epistemic uncertainties in the loading environment, that pose the greatest challenges

for safe and economic design of wind turbines offshore. This includes the joint variability of the wind, waves, and currents, both locally and across the spatial scales of an offshore wind plant, as well as the soil properties, including the potential for scouring. This variability complicates the design of offshore wind turbines for either a class-based turbine certification or a project certification. Uncertainty quantification (UQ) tools can help develop efficient, low-dimension surrogate models that, through validation studies against higher-fidelity quasi-truth models, can offer full-system models that capture the

most important physics for evaluating designs. Given the need for cost efficiency in this emerging industry offshore, it is important that the development of such tools that explicitly address uncertainty are a focus.

Such methods can be combined with probabilistic, reliability-based design methods and can be used within automated design optimization. There are many characteristics of loads and material properties in the offshore environment that are not well-understood, and the ultimate consequence of a failure may be far-reaching. Techniques exist to relate multiple

factors to the probability of failure and the consequences thereof (such as First Order Reliability Method, or FORM). All available techniques have limitations, however, especially as the systems become more complex and the number of significant external design factors increases. Improved techniques will help optimize designs and reduce system costs while ensuring that the overall energy supply is adequate.

**7.6 Recommendations for Offshore Wind**

For offshore wind to become a vital component of the global renewable energy portfolio, costs, especially for floating systems, need to come down significantly. Ground-breaking technology pathways need to be explored across components to identify those that provide economic and performance benefits to the integrated system, including how the design affects the construction, installation, and maintenance costs. A recent study (Barter et al., 2020) outlines several innovations for future floating offshore wind energy installations—including novel substructure designs; novel anchoring methods;

integrated system design including the substructure that seeks to reduce overall cost and mass; alternative materials; cost uncertainty and sensitivity studies; designs that promote greater towability; consideration for port availability and trade-offs against deep-draft designs; advanced floating plant control; and consideration for more comprehensive floating turbine classes for design. Because of significant complexities associated with floating offshore wind turbines, a multi-disciplinary, multi-fidelity, systems modeling approach that also considers uncertainty quantification is required that aims to reduce the

number of viable technologies and identify optimal solutions.





To examine the feasibility of different technology approaches, it is imperative that we develop a multi-fidelity suite of tools and design methods that can accurately represent these systems. Many of these tools already exist, but improvements are needed to address the challenges outlined earlier to ensure they can model the array of offshore wind designs being proposed, address the needs of scaling to the larger designs that will lower cost, and consider the diverse metocean and geotechnical conditions that will be encountered around the world.

To ensure the accuracy of the tools, validation campaigns are needed that compare modeling capabilities to the real-world physics of these systems. The large variability in design approaches requires a series of validation campaigns addressing the unique needs of each technology. Similarly, the large variability in offshore environments necessitates validation under a variety of metocean conditions. Model testing within wave tanks and wind tunnels makes up the first step toward developing the appropriate modeling theories and design practices (Viselli et al., 2015; Borisade et al., 2018; Chen et al., 2020; Popko et al., 2021). Ultimately, though, these design capabilities need to be validated against full-scale, real-world systems that are subjected to the complex metocean and geotechnical conditions of the open ocean with the associated and realistic dynamics and uncertainties of full-scale structures. This validation will require demonstration projects that afford open (shared) knowledge of the turbine, controller, and support structure properties, and measurements of the metocean and geotechnical conditions and structural response—this, in turn, will require collaboration across multiple sectors and parties.

The Research at alpha ventus (RAVE) project in Germany is an excellent example of the type of public projects that can help progress the technology. RAVE outfitted an offshore wind power plant with instrumentation to supply to institutions for offshore wind research. Located near the FINO tower, associated measurements of the full metocean climate are also available. Limitations still exist on the openness of the data set, especially regarding the turbine properties, which are a hindrance to its usefulness, and should be addressed in future public campaigns. Public data sets like this are needed now for floating technology. A complete data set would include these full-scale measurements along with laboratory testing of the same design in controlled environments. This combination would provide knowledge about the limitations of scaled testing, as well as modeling tools, enabling an understanding of the best and most-affordable methods for offshore wind design. With this knowledge, we envision that high-fidelity tools may one day be able to replace the need for any physical testing of designs, but significant validation data are needed to reach this goal. Further discussion of these topics is covered in Section 9.

## 8. Manufacturing and Materials

A state-of-the-art wind turbine represents a unique technical and commercial challenge because of the demanding combination that results from the sheer scale of the hardware, the remoteness of many installations, maintenance



difficulties, expectations for longevity, and cost constraints. The largest land-based turbines exceed 200 m in height with rotors that are over 150 m in diameter, whereas the largest offshore wind turbines can exceed 250 m in height with rotors more than 200 m in diameter, and must be supported by fixed or floating structural systems of significant size on their own. These sizes of wind turbine systems are, as such, comparable to that of the largest civil infrastructure systems, such as

buildings and bridges but, in contrast, the latter are not expected to rotate some 100 million times during their lives. Further, this large size of today's wind turbines is comparable to aircraft; however, frequent inspection and reliability-centered maintenance is a required and expected part of the total cost of ownership of passenger aircraft. Because of the (often) remote location of most wind power plants and the difficulty with accessing nacelle and rotor components, they are designed for high reliability and infrequent scheduled maintenance. The highly competitive nature of the electric generation

industry also requires that wind turbine hardware must be manufactured at costs of less than $20/kg, whereas aviation hardware typically costs an order of magnitude more.

The operating environment of wind turbines is also challenging. A wind turbine will typically experience approximately 100 million cycles of fatigue during its lifetime. Compare this to the fuselage of a commercial jetliner, which experiences 30,000 cycles of pressurization and depressurization. The wind inflow conditions encountered by a wind turbine rotor are

highly transient, characterized by significant variations in the moving average wind speed and rapid changes in wind speed and direction because of turbulence, gusts, and other coherent inflow events. This, combined with the rotation of the rotor, results in magnitudes of fatigue that are unparalleled in other hardware of this size. In addition to the loading, wind turbines and components are also subjected to harsh conditions including significant seasonal variation in temperature; constant exposure to ultraviolet radiation; exposure to rain, lightning, ice, snow, dust, and sand; and exposure to corrosive saline

spray (in marine environments).

Subject to this harsh environment, wind turbines have historically been designed to operate for at least 20 years without replacing any major components. Operational principles have begun to evolve, and many owners and manufacturers are now planning for plant lives of 30 or 40 years, with an expectation that major components such as gears, bearings, generators, and blades may need to be refurbished or replaced one or more times during that lifetime, but not more

frequently than every 10–12 years. This model is more consistent with other modes of power generation. Historical expectations of minimal maintenance have also evolved to accommodate more frequent (e.g. annual) inspections and periodic maintenance of major components. Nevertheless, the frequency remains far less than is expected for other large hardware such as aircraft or industrial equipment. These requirements create high demands for materials and manufacturing processes and logistical models that can deliver structural integrity, long life, and high reliability at minimal cost.

Wind turbine blades are some of the largest composite structures in the world while also some of the cheapest on a per-mass basis. This size-at-low-cost combination presents enormous challenges in manufacturing tolerances and quality (see Figure 12). Yet, demand for improved manufacturing capabilities is driven by the trend of increasing blade length for a





given rated power. The direct relationship between increasing rotor diameter and turbine capacity factor can only serve to reduce LCOE if the incremental blade manufacturing costs are offset by the marginal increase of revenue from higher

energy output. Achieving this end requires blade manufacturers to improve the total yield of their operation by increasing material throughput by the simple metric of kilograms of material per unit hour of labor (kg/hr). This is accomplished by ensuring the metric of a 24-hour "cycle time" is maintained even as blade mass and surface area increases. This cycle time, defined as the interval of time from an open mold to the demold of a blade unit, is essential for level staffing so the optimal number of direct laborers are attending to various operations on the mold at given times in the cycle. It is simply not

possible to maintain minimum direct labor hours without establishing this circadian rhythm.

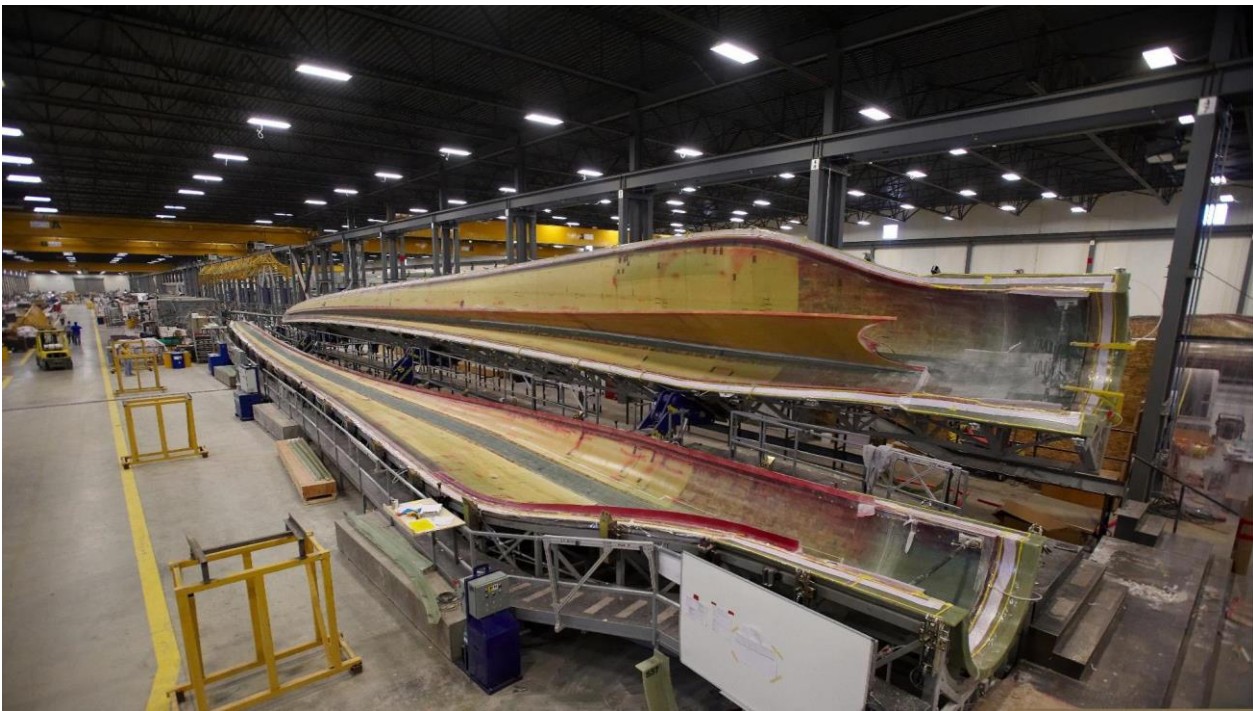

**Figure 12: A 40 m wind turbine blade in production demonstrating the size and complexity of manufacturing wind turbine blades at modern scales. The assembled blade skins and internal webs are about to be bonded together, a process which has tolerances of 1 mm in some cases, and requires both speed and attention to detail for the adhesive material to bond**

**effectively. (Photo credit TPI Composites)**

Similarly, blade extension for a given rated power demands lighter-weight design and use of higher specific stiffness materials to prevent the overloading of systems and avoid tower strike. Material cost can make up 65% or more of the blade total cost. Therefore, using more expensive carbon fiber requires both judicious and innovative approaches for





application, such as premanufactured pultrusions that are laminated together in molding processes. This approach can
shorten cycle time for components and uses the fiber in its lowest cost form. But much greater innovation is required for
this trend to continue.

Transportation of wind turbine blades, nacelles, and towers has become increasingly challenging because of length and
width constraints for both road and rail transport. Logistics engineers have continued to increase the transportation
dimension limits through specially made transportation equipment, although the pace has slowed considerably. For
example, in the United States, the length limit is approximately 80 m and the width and height limit is 4.75 m; other
jurisdictions have similar constraints (Smith and Griffin, 2019). Absent further innovation, transportation imposes limits
on the design of components. Towers and drivetrain components must compensate for smaller diameters with thicker walls
or longer and heavier and costlier components, and rotors become a trade-off between overall power capacity and capacity
factor, for instance. The most obvious solution to these problems is increasing modularity of components with on-site
assembly or even on-site fabrication. However, the joining of highly engineered components leads to added complexity,
often in adverse conditions, and potentially more maintenance and reliability concerns in operation. Additionally, in the
case of blades, joints add weight and gravitational loading.

## 8.1 Emerging opportunities and challenges

The pursuit of higher capacity factors in wind power has resulted in larger, nameplate capacity turbines with associated
large rotor blades, drivetrains, towers, and foundations. Additionally, matching rotor diameter to the wind resource for a
given site has led to the fabrication of blade families that share a common root but allow for custom-length tips. Blade
performance benefits from unitary bonded construction; however, the transportation from factory to inland erection sites
is challenging, as described earlier. Developing a truly modular blade design for on-site assembly provides a clear solution
to both the manufacturing challenge of multiple blade families, as well as enables the complex logistics of large blade
transport. In a similar fashion, families of gearboxes are being developed that share common mechanical characteristics
but can be configured relatively easily for different rotor diameters and wind resources (Nejad et al., 2021). In addition to
on-site/port-side manufacturing of components, such as blades, gearboxes, generators, and nacelles, innovations in tower
(Jay et al., 2016; GE, 2020), foundation, and installation technology that reduce logistics burdens and installation costs are
occurring for both land-based and offshore wind.

The push to constrain rotor weight growth leads to more flexible blades, which creates difficulty in maintaining blade-to-
tower clearances under high loading events. To date, the strategies to alleviate this concern have been to pre-bend the
blades away from the tower in the design, as well as use carbon-fiber composites, which have a higher specific stiffness
(stiffness/mass) than the more common glass-fiber composites. However, as blades continue to grow, the dimensions of
the required pre-bend create challenges in manufacturing and logistics because of the high curvature of the blades. Also,





carbon fiber remains very expensive compared to glass fiber. Thus, the industry and the research community have begun to look at alternatives, including processing methods to optimize the cost and performance of carbon fiber, as well as considering downwind turbine rotors to alleviate blade-tower clearance issues (Pao et al., 2021). Finally, the torque density of drivetrain components continues to increase as well, by either optimizing conventional designs with gearboxes or introducing new technologies such as superconducting generators (Nejad et al., 2021).

Wind turbine designs can incur high loading due to both harmonic resonance with aerodynamic loading and aeroelastic instabilities, increasing in concern as blades become more flexible. These loads must be mitigated either through passive, aerodynamic and structural damping, or active damping by the turbine controller. Structural damping is typically assumed to be very small, around 1%, thereby representing an opportunity to develop and integrate new materials and structural forms in blades, towers, and other components to increase structural damping and provide increased design margins.

Continuing growth in rotor diameter and power result in increasing drivetrain torque and other loads because the rotor tip speed is constrained. Typically, this increases the diameter of the drivetrain, whether it is the diameter of the support bearings, the gearbox ring gear, or the generator air-gap diameter. In each case, there are strict stress, deflection, and manufacturing tolerance requirements that become more difficult to achieve affordably as the size of the component increases.

The impacts of lightning to wind turbines continue to change as turbines get taller and carbon is increasingly used. There are important lightning effects that are not well-understood for modern turbines. The mechanism of lightning changes from predominantly cloud-to-ground to ground-to-cloud as turbine blade tip heights grow past 200 m. This is a phenomenon not seen in other structures such as towers and buildings until reaching heights of 600 m or more and relates to the blade rotational effects on electrical field buildup. Additionally, the frequency and magnitude of the strike increases with tip
height, leaving conventional protection systems possibly inadequate. Without a more complete understanding of the physics of lightning strikes on turbines and the effects that it has on the protection system and structure, turbines may face higher-than-expected rates of failure in the future.

   Erosion of the leading edge of blades is also a prominent and growing concern for blades. This is predominantly caused by rain droplet impacts, with the damage rate growing with the impact velocity, droplet size, and number. The tip speeds of
modern blades have grown from 75 m/s to 85 m/s for land-based machines over the past two decades, and over 100 m/s for offshore machines, leading to an exponential increase in observed erosion at wind plants. Mitigation solutions do exist, both in the form of specialized coatings as well as derating during rain periods and have been shown to be successful. However, increases in tip speeds are beneficial to the turbine system, especially the drivetrain, where it reduces torque and with it, drivetrain mass. Thus, erosion will continue to be a problem for turbines in the future and require better knowledge



of erosion mechanisms. Obtaining that knowledge requires defining high-strain rate material behavior and environmental aging effects, as well as conducting accelerated erosion testing.

Modern wind turbine blades are produced with thermoset resins reinforced by either glass or carbon fiber. These resins exhibit higher thermal stability with higher mechanical properties at elevated temperatures including stiffness and strength. An alternative type of resin, thermoplastics, have other desirable properties, including higher toughness with the potential

to improve long-term fatigue resistance. They may be fused by melting adjacent contacting surfaces and then cooling, which could reduce assembly time and cost. The use of thermoplastic matrix composites has generally been limited to advanced polymers with expensive processing technologies because of the high melt viscosity of the plastic. However, new liquid resins have been introduced that have good thermal stability throughout the blade operating environment, along with having the process capabilities of a thermoset with the benefits of thermoplastic materials. They are also recyclable.

Carbon fiber represents a significant opportunity for increasing the stiffness and strength of blades, improving performance, and allowing for passive load alleviation through bend-twist coupling. The material is currently present in some blade designs but remains significantly more expensive than glass-fiber composites. The cost of carbon fiber depends on the precursor and processing method used. Recent research has shown the possibility of optimizing these feedstocks and processes specifically for the design requirements of wind turbine blades, but further pathways remain for improvement.

These include sourcing textile-based (polyacrylonitrile) PAN based on super heavy tow (i.e. higher weight per unit length) as a precursor to drive down feedstock cost and increase capital utilization, use of bio-based precursors, and use of low-cost, mesophase, pitch melt spun into precursors for carbon fiber. Great challenges remain, but the industry has benefited from both this work and the scaling of unprecedented volumes of carbon-fiber capacity, with a continuous decline in the cost of these enabling materials.

Rare-earth elements, such as neodymium, praseodymium, and dysprosium, are used in many permanent-magnet generators, as they allow for power-dense designs because of their very strong magnetic field and high mechanical-to-electrical conversion efficiency at low power. However, the price of rare-earth elements and the resulting capital cost of the generator compared to the overall techno-economic benefits of the variety of other drivetrain technologies will continue to be a design consideration. As a result, the wind industry is actively developing innovative technologies to reduce, substitute for, or

entirely eliminate the need for rare-earth elements, including superconducting generators that could be positioned to reach technology and manufacturing readiness levels required for deployment in the commercial market in the next decade (Veers et al., 2020).

The relative immaturity of utility-scale wind power has, until recently, meant that the end-of-life issues related to turbine decommissioning and material reclamation have not been fully addressed. A significant portion of the turbine, including

many metallic components from the hub, drivetrain, generator, bearings, and tower, can be recovered through well-



established pathways of reuse and recycling. However, recovery of rare-earth elements in permanent-magnet generators, foundations, and rotor blades are not currently supported by similar direct pathways. While foundations can be readily reduced to rubble with the aggregate harmlessly distributed or reused, the cost of this remains significant. Progress has been made toward addressing both fabrication and end-of-life recovery such that now the spotlight has been placed on the expensive turbine blades and rare-earth permanent magnets (Alves Dias et al., 2020). Unfortunately, tens of thousands of wind blades are decommissioned and disposed of annually, typically in landfills. An alternative and less-used option is the recovery of thermal content from polymer resin, paste and core materials to fuel cement kilns, with the balance of the inorganic materials providing a portion of the cement content. Other thermal technologies that extract embodied energy from pyrolysis for power cogeneration combined with recovery of glass and carbon fiber offer new pathways. Recovered carbon fiber in the form of short discontinuous materials has ready markets in automotive, brake pads, and other applications, competing favorably to virgin fiber selling at more than $18/kg. However, recovered glass fiber, comprised mainly of silica, is challenged by virgin fiber selling less than $1.25/kg. Identifying high-volume applications for recovered glass fiber from pyrolyzed fiberglass composites is critical.

The industry is exploring new engineered polymer resins offering a means of "polymer disassembly" and use for new products. The decommissioned blade may be considered a "chemical warehouse" that can use accepted low-cost and environmentally friendly processes to extract these chemicals as feedstocks for new products. An available example, approaching industrial scale, is a novel amine that is designed to react with epoxy resin in a process identical to conventional blade processing. The result is an epoxy resin for glass- and carbon-fiber composites soluble in 20% acetic acid at modest temperatures (80º C). The recovered fiber is not damaged by the process and the epoxy is recovered by precipitation as a fully recyclable thermoplastic resin that can be used to mold new products.

### 8.2 Unknowns and their impact on the ability to make progress

Material and manufacturing defects affect the strength and durability of wind turbine components, especially blades. Commonly found defects include fiber waviness, porosity, delaminations, and disbonds. These flaws significantly lower the strength and fatigue resistance of structures, possibly beyond what is accounted for in design standards. It is critical to understand the impact that these defects have on structures, including material aging and uncertainty of operational loading. Improved manufacturing methods as well as design improvements to increase resilience are needed to ensure component reliability in increasingly larger components with longer expected design lives.

The current design life estimation process for the drivetrain only accounts for failure of the gears, rolling element bearings, and shafts by fatigue. Other failure modes such as wear, axial cracking (see Figure 13), micropitting, brinelling, and corrosion lead to enough of a disparity between the design and service life that, in the wind industry, have led to higher-than-expected operational costs. Continued research into these failure modes that are most prevalent and costly is necessary.





Innovations in material processing, coatings, lubricants, and additives also hold the possibility of addressing reliability challenges that have historically been experienced in the wind industry (Nejad et al., 2021).

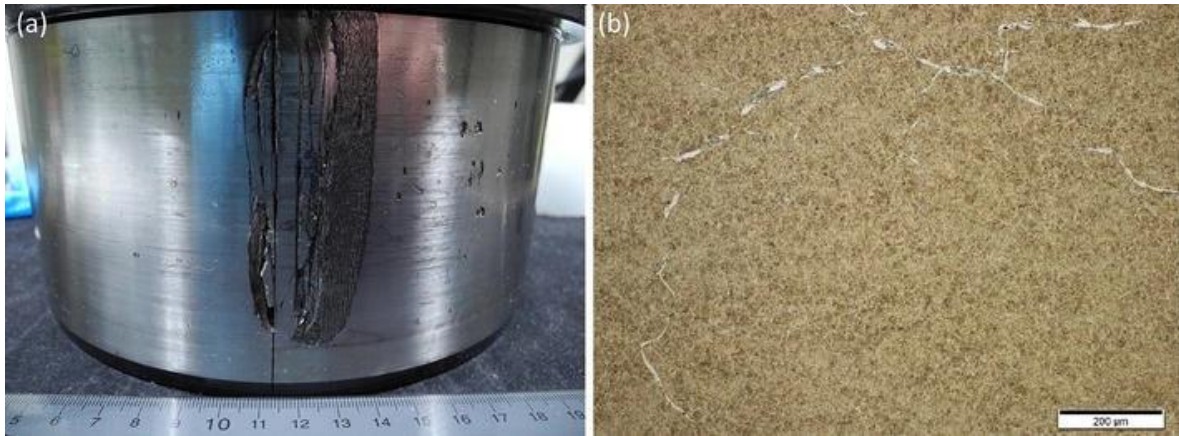

**Figure 13: (a) A failed bearing from a wind turbine gearbox and (b) a microscope cross-sectional image showing the failure mechanism of white-etch axial cracking from a failed wind turbine gearbox bearing. Such failure mechanisms are critical in understanding and predicting component lifetimes. (Photo source Gould et al., [2015])**

Moving toward damage-tolerant design will require a better understanding of how damage progresses from an initiation site to structural failure. The rate and extent of this growth can be limited by materials and structures that are specifically

designed to arrest the progression. Such designs would offer improved resilience to flaws that emanate from the manufacturing process, and damage caused by transportation, unanticipated operational loading, or environmental degradation. These design methods necessitate probabilistic modeling of damage progression throughout structures (Strasser et al., 2015). There always exists an inherent trade-off between increasing performance and increasing reliability. Unfortunately, the lack of accurate material damage models and resulting uncertainties in prediction of component lifetimes

make calculating this trade-of very difficult with the current level of knowledge.

Design verification of wind turbine blades relies on a series of four to six static (i.e. ultimate load) tests of a nearly full-scale test blade coupled with one or two accelerated fatigue tests comprising the application of 1 to 10 million cycles of oscillating load. It is widely accepted that this approach has several shortcomings. Among the most important are the challenges in accurately replicating design loads along the entire span of the blade in a single test, the failure to capture the

influence of load history, and challenges in accurately replicating anticipated fatigue damage in all components of the blade structure, particularly components constructed of different materials (e.g. carbon, glass, adhesives) with differing fatigue responses. Moreover, with the largest blades already exceeding 100 m in length, full-length testing is becoming more





challenging and time-consuming. As a result, blade test standards are evolving toward approaches that test substructures comprising partial-span and partial-chord sections of blades. These approaches will require careful validation to ensure

that the application of loads on the boundaries of these substructures accurately reflect the boundary conditions in the full-length blade. The use of such substructures will enable more accurate application of loads to structurally critical sections and components of the blade and will allow for more cost-effective performance of more testing (i.e. a higher number of structurally critical design conditions).

Design verification of gearboxes can be a combination of full-scale physical testing, advanced simulations, and prior

experience with similar designs as described in IEC 61400-4 (IEC, 2012). Testing consists of a series of four tests of different load levels and durations, in both stand-alone configurations and integrated with the entire drivetrain in a dynamometer or installed turbine. Similar to blade design verification, it is recognized that this approach has its own shortcomings. As turbines continue to increase in size, the dynamometer facilities need to grow as well, from the existing 5 MW to 16 MW range in operation to planned facilities that are 25 MW or larger. In addition to the driving torque and

speed, non-torque loads are required, as well as test cases involving interactions with the electric grid that require a grid simulator. Because of the expense of these facilities and test programs lasting hundreds of hours, verification by testing individual drivetrain parts or components is of increasing interest just like it is for blades. Regardless, the critical failure modes with the highest risk in operation are often not the same ones that are discovered or addressed during common overload testing.

Several promising non-destructive inspection technologies have been analyzed and developed for inspecting wind turbine components. Wind turbine blades, which are difficult to inspect because of varying construction and high signal attenuation, can be inspected using phased array ultrasonics, thermography, and other technologies. Ultrasonics have been shown to give detailed results for solid laminates and bond lines in manufacturing plant settings. Thermography has been used to give qualitative indications for field inspections. Improving the ability of these methods to provide cost-effective,

qualitative results in the field is critical to improving the reliability of components and allowing for full implementation of durability and damage-tolerant design.

Further growth in large turbine components will need to move toward more on-site assembly. Blades will need to have joints that can be reliably assembled in the field and be verified to last through high numbers of fatigue loads. On-site fabrication of conventional steel towers will need to grow in maturity, and other hybrid solutions and alternative concepts

may need to be explored, especially if wind power generation is to be deployed economically in low-wind-resource environments.



**9. High-fidelity modeling, high-performance computing, and validation**

The need for HPC and HFM in wind turbine and plant innovation is a common underlying theme across all technology development areas. Validated computational models running the gamut from truly predictive and computationally intensive
high-fidelity simulation for scientific enquiry (Sprague et al., 2020), to computationally parsimonious resource models used in design and optimization workflows are now essential in assessing the complex inflow and structural interactions with modern turbine architectures (Veers, 2019a). Although wind turbine technology and the understanding of the underlying physical phenomena driving turbine performance and loading has matured considerably (Haupt et al., 2019), coupled physics considerations across a wide range of spatial and temporal scales and combined external physical drivers
(e.g. from wind, waves, and currents in offshore systems) make design and optimization at the turbine- and wind-plant levels challenging.

Luckily, the last two decades have seen a tremendous surge in the computational power available to the wind research community. The top supercomputers have peak computational performance exceeding 200 petaflops and the first exascale (1,000 petaflop) systems are scheduled to come online in 2022; their names are "Frontier" (Schneider, 2022) and "Aurora"
(Mann, 2020). These leadership-class supercomputers are adopting specialized accelerators for computation-intensive operations, such as general-purpose GPUs and tensor processing units (TPUs) to minimize energy requirements and accelerate scientific computing speed and scale. With technology that has evolved from smartphones and edge computing devices, a single GPU processor now packs more data processing capability than was available on Apollo 13. The rise in cloud computing over the last decade has further made access to large-scale computing resources, including advanced GPU
processing capability, which is ubiquitous and readily available to the wind industry.

The availability of exascale computing (Kothe et al., 2019) gives wind researchers an opportunity to perform high-fidelity simulations of wind plant phenomena at unprecedented resolution that account for all relevant spatial and temporal scales. Federal investments and partnerships within the U.S. Department of Energy, especially the National Nuclear Security Administration, have led to the development of computational resources and tools that are now being applied to massively
parallel wind plant simulation at exascale. The Office of Science Exascale Computing Project, ExaWind, is developing the next generation of high-fidelity modeling capabilities that will allow full simulation of a 100-turbine wind plant with complete atmospheric physics and fully resolved 3D blade and rotor flows (Alexander et al., 2020). Simulations capable of modeling wind plants containing hundreds of turbines in a computational domain spanning several tens of kilometers with realistic turbulent ABL inflow will require efficiently utilizing the first exascale system capabilities of supercomputers
"Frontier" and "Aurora.". Such simulations are considered "grand challenges" (Sprague et al., 2017) that will demonstrate the potential of Exascale computing.  Simulation fidelity will allow future researchers to use first-principles-based analyses to examine the effect of even complex inflow physics (e.g. wake generation, evolution, mixing, and interaction with the



background atmospheric flow) and provide suites of deterministic loads and dynamic coupled response simulations of integrated turbine and plant systems.

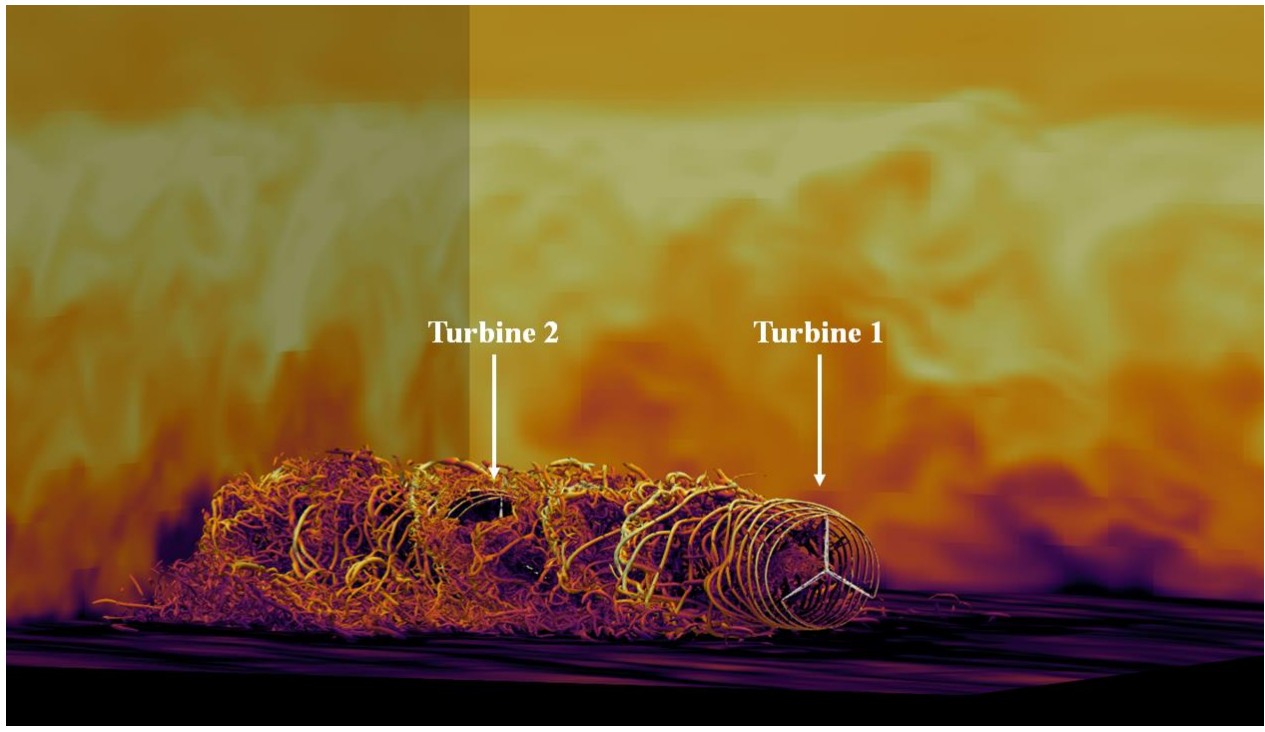


**Figure 14: Simulation of two NREL 5 MW wind turbines in a neutrally stable atmospheric boundary layer. The colors represent velocity magnitude, and the vorticity isosurfaces highlight the vortices coming off the rotors. This simulation was performed using the ExaWind hybrid solver, which connects, via overset meshes, the Nalu-Wind near-blade-flow models to the AMR-Wind background flow solver. (Image credit Ashesh Sharma, NREL)**

**9.1 Challenges in first-principle modeling of physical phenomena**

Even with the tremendous advances being made in HPC, and the advanced physics being included in the evolving generation of HFM simulation capabilities, not all underlying physical drivers can be fully resolved even with exascale computing. Analysis and modeling approaches incorporated into HPC models are still needed in conjunction with, and to address the following science challenges and barriers to innovation:

• Direct numerical CFD simulation is only possible for low-Reynolds-number flows even with the most advanced HPC and HFM simulation capabilities. Multi-megawatt wind turbine and plant physics at high Reynolds numbers must still rely on empirical models for turbulence embedded within LES and RANS, necessary computational simplifications of



the underlying fluid physics. These models add empiricism even at the highest levels of fidelity and their impact remains an area of ongoing research, especially at high Reynolds numbers.

- Turbulence at different spatial scales within the atmosphere are driven by very different physical processes. Flow phenomena at macro (kilometer) scales governing weather-driven and other atmospheric events are very different from the flows that determine turbine performance at the micro (millimeter) scales within the boundary layer of a rotating turbine blade. Modeling the flow and turbulence characteristics across eight orders of magnitude in physical scale requires different modeling physics and coupling strategies that transition flows from the global level (weather) where the energy is produced, to the ABL, where the energy is delivered, to blade-resolved simulations where the energy is extracted and performance efficiency established.

- Multi-phase flow modeling is a new requirement with offshore deployment in shallow and deep water. Resolving the hydrodynamic and load interactions on both fixed and floating turbine structures requires additional physics that account for ocean waves, current flows, and the associated interactive radiated interactions produced within the plant. Resolving the structural impact, especially from nonlinear waves, requires critical modeling elements like hydrodynamic damping coefficients to ascertain structural loads. Expensive tank testing of complex physical models to ascertain these variables represents a significant cost impact to design innovation, development, and deployment of new offshore technology.

- Novel numerical algorithms that capture the nonlinear coupling between physical regimes remain a significant challenge. These include the ABL inflow resource (in complex terrain and the marine boundary layer), air/sea turbine interactions offshore, and the dynamic structural system and component coupling that occurs when large and dynamically soft turbine architectures are driven with broadband air and sea excitations. ABL coupling examples that impact turbine performance include inflow/plant two-way coupling, gravitational waves induced by the influence of large turbines on the ABL, and wake flows between downwind turbines and wind plants. Air/sea interactions have well-established effects on the wave field characteristics, influence of local current flows and bathymetry, and the physical modification to the inflow wind resource field (Sullivan et al., 2014). Structural dynamic response and loading are driven by flexible structural components (e.g. blades, towers, floating platform). Notably, critical resonant coupling events, such as SIV and VIV cannot be predicted without higher fidelity CFD modeling. Each topic is complex with unique physical modeling attributes that require detailed examination and thorough discussion beyond the current scope as to the causal relationships, interactions, and modeling needs.

- The need to model and assess integrated wind plant performance at scale requires a continued investment in advanced computational capabilities. The limitations that impact existing HPC simulation are directly linked to the fundamental limits imposed by system hardware and software. Expanded computational capacity is ultimately determined by processor clock speed, internode communication performance, and computational node availability. Strong and weak scaling, at the device and system levels, determine computational throughput. Desired HPC performance improvements are driven by the insatiable desire for better resolution, computational speed, and the expansion of





complex physical systems modeling needed to address national strategic interests by the science community. As a result, scientific computing evolution is expected to evolve beyond the exascale, with advanced capabilities to address multiscale and multiphysics problems that will benefit the wind energy community. Similarly, software is progressing 1895 at unprecedented speeds to take advantage of hardware advances. The need for improved linear solvers and parallel programming strategies to efficiently utilize all available computational resources, as well as programming libraries that reduce programming complexity on advanced GPU architectures, are also priorities that will impact wind energy research.

- The advances in computing hardware and transitions in scientific computing to GPU architectures will place an ever-
increasing demand on a limited pool of programming expertise. Most software development is done by wind subject matter experts with programming skills honed for CPU architectures and using a single or a limited number of cores. Taking advantage of the scale and computational capability GPUs offer when simulating wind-plant-scale environments requires a fundamentally different programming approach and skill set. The ExaWind HFM development shows the evolution of exascale computing methods in wind energy. Massively parallel scientific
computing requires advanced software libraries, data storage, and input/output as foundational considerations driving development. Parallelism and application scalability are also amongst the principal software design drivers. In the near-term, multi-skilled software development teams will be needed to develop large-system modeling and simulation capabilities. Subject matter expertise will define the needed physics, relevant partial differential equation numerical descriptions, and system interaction and computing requirements. Computational scientists will translate the desired
computational domain into the structured solver architectures that best utilize the GPU resources available. As GPU computation capabilities become more accessible through the cloud, the scientific programming expertise will be in very high demand as industry adopts the emerging tools and adds the necessary technical programming staff to meet their strategic modeling and development requirements.

### 9.2 HFM simulation hierarchies and associated design challenges

The traditional approaches for turbine technology development using a combination of low- to mid-fidelity modeling tools, scaled-model validation testing, and risk mitigation through prototype field testing is the current state-of-the-art used by industry. Multiple numerical simulations using independent realizations of stochastic inputs in various combinations are used to achieve a statistically significant result, and when coupled with best practice safety factors, establish the basis for design. In contrast, fewer high-fidelity simulations are possible due to the cost, both computationally and in design iteration
turnaround time, and lack statistical significance given the broad range of independent operating conditions. Resolving the concerns of not capturing the appropriate physics for complex systems vs. insufficiency in capturing all relevant design drivers requires additional research and application experience to fully realize the value of HFM methods.



The coupled dynamic response of very large systems and unique operating conditions, especially floating offshore wind plants in deep water, requires a much higher level of fidelity for deterministic estimates of loads and system response early in the design process. High-fidelity modeling on HPC leadership-class machines is not a viable alternative to conventional methods. The large computational resource and tremendous amount of time required for each HPC simulation will not accommodate the numerous simulations needed for design and certification of new architectures. Future innovations must, therefore, rely on a hierarchy of validated models for different aspects of design and development that optimize design workflows.

Development and adoption of a hierarchical chain of modeling fidelities has been challenging. At the highest fidelities, simulations must account for complex nonlinear interactions (e.g. fluid-structure interaction, air/sea interface, mesoscale-microscale coupling). These simulations are computationally expensive and require leadership-class supercomputers not readily available to industry practitioners. Advanced HPC methods require a team of scientists with both wind energy expertise and a strong computational science and applied mathematics background to deal with a multitude of computer-science issues (e.g. strong- and weak-scaling in parallel environments, input/output and bandwidth challenges). Exploiting emerging computing architectures (GPUs) requires a significant investment as existing HFM codes are simply not portable and literally need to be completely reformulated.

Code development has traditionally been modular and decoupled with each discipline (e.g. aerodynamics, structures, materials, mesoscale atmospheric modeling) developing analysis methods that are discipline-centric. Multi-physics simulation is achieved through a concatenation of multiple models. However, coupling physics to account for all relevant nonlinear interactions at appropriate fidelities requires developing novel coupling algorithms and solver methodologies. Scalability requires numerical algorithms capable of massively parallel execution and the implementation of modern software design and development practices not common within the wind research community.

Mid- and lower-fidelity models involve approximations or simplifications to capture important physics without explicitly modeling all the higher-fidelity nuance. However, simulation codes are often used beyond the intended scope and applicability range embedded in the simplifying assumptions. Without proper validation, however, these results may lead to inaccurate load estimates and/or wrong design decisions. The development and application of empirical design models relies on engineering intuition, experience, and limited experimental or computational data sets. These formulation approaches should always incorporate a continuous improvement process that incorporates higher-fidelity modeling and rigorous experimental validation.

Finally, the analyst must be able to seamlessly slide across the modeling-fidelity hierarchy during the design and optimization process and use appropriate models at different design stages. These capabilities require well-designed software frameworks with well-defined application programming interfaces (APIs) facilitating code coupling and



integration. A common programming language is needed to define various aspects of wind plant design and allow for the
different technical disciplines to communicate seamlessly within a multi-disciplinary design and optimization workflow.

## 9.3 AI and machine learning—the next generation of high-fidelity-modeling design tools

The continued improvement in advanced computational capability and the more recent access to these resources through
the cloud has led to a renewed interest in machine-learning technology. Over the last 5 years, deep neural networks have
been used by researchers with tremendous success in areas of computer vision and natural language processing (LeCun et
al., 2015; Halevy et al., 2009). These self-learning, deep neural networks have a promising future in multiple applications
within the wind community (Elyasichamazkoti and Khajehpoor, 2021; Stevens et al., 2020), including probabilistic wind
resource forecasting, advanced wind-power-plant-level production control, seamless grid integration, and a new generation
of mid-fidelity design tools facilitating superior physics and non-linear dynamics in design processes.

Machine-learning (ML) models can be thought of as high-dimensional, functional approximators that can learn the non-
linear transformations between a set of relevant inputs and outputs. Networks are built on simple but deep layers of linear
transformations trained on data sets obtained through observations or high-fidelity simulations that capture the dominant
physical processes of interest. Network training is now an automated process, with continuous updating and incorporating
new data under diverse operational conditions as it becomes available. Hence, these models continuously evolve and
improve. The underlying computational operations comprise basic linear algebra that can be implemented quite efficiently
on GPUs and TPUs. The simplicity of this approach, combined with developing novel automatic-differentiation and
optimization techniques, has resulted in the availability of high-quality, open-source, ML frameworks (e.g. scikit-learn,
TensorFlow, pyTorch), leading to what ML practitioners have termed the "democratization" of AI.

The emergence of ML-based surrogate modeling presents wind researchers with an alternative to traditional reduced-order
models (ROMs) to developing computationally parsimonious empirical design models. ML surrogates can use training
data from experiments and high-fidelity simulations, as well as continuously streaming data from operational wind plants.
In fact, the wind community is already adopting these techniques to develop reduced-order representations of turbulence
(Stengel et al., 2020), unsteady aerodynamics (Ananthan et al., 2020: Vijayakumar et al., 2021), loads (Dimitrov, 2019)
and plant-level interactional effects for siting optimization (Dimitrov and Natarajan, 2022). ML-based models can also be
used to augment or improve legacy models (which often encapsulate an enormous amount of knowledge and experience).
ML can work in synergy with legacy ROMs, rather than replacing the traditional models by data-driven black boxes. ML
approaches therefore provide a unified approach to translating scientific insights gleaned from high-fidelity simulations
into ROMs suitable for design and optimization workflows.

## 9.4 Future high-performance-computing and high-fidelity-modeling enablers



Given the highly uncertain operating environments and the need to design wind turbines and plants for a 20-year operating
life, the lack of reliable data that can be used to validate the computational models remains the biggest challenge to
effectively harnessing the computational power currently available to industry. Data needed for validating existing models
and developing future design and operational capabilities for both deterministic and probabilistic simulations can be broken
down by:

- Data sets that allow for model validation at the unit, subsystem, and full-system levels that incorporate the applicable
driving physics across the entire operational range of length and time scales. Examples include high-resolution data
sets for aerodynamic performance of 2D airfoils and 3D blades across a wide range of Reynolds numbers (1 to 20
million) and turbulence intensities, detailed atmospheric inflow characteristics (e.g. veer, shear, turbulence,
stratification) in complex terrain, and the unique combination of air/sea interaction (e.g. marine boundary layer, wave
and current) in the metocean environment.

- Data sets that allow for the verification and validation of models capturing interactional dynamic effects including
subsystem resonant coupling. Examples include detailed loads measurements under relevant
aerodynamic/hydrodynamic environments to understand fluid-structure interactions and the resulting response of
turbine structure (e.g. bend/twist coupling, blade pitching moment and dynamic stall response, air/sea interaction,
floating platform response, rotor dynamics, and drivetrain loading response).

- Data sets targeting the future development of AI- and ML-based design tools. These data must be comprehensive and
diverse, covering the range of operating conditions and at a fidelity sufficient to ensure that the non-linear response of
physical drivers and dynamic system response are adequately captured.

- Data sets that allow for the investigation of long-term climate effects, component degradation, and sensor drift effects.

Today's big-computing opportunity cannot be fully realized without a sustained investment in multi-fidelity design tools
developed specifically for modern computational architectures. Capturing the higher-fidelity physics and non-linear
dynamics of modern turbines in a single, unified modeling code with "selectable" fidelity options is the preferred industry
approach that facilitates both the detail and the iterative capability for design and certification. The future development of
HFM, AI, and ML design tools are critically dependent on open-source data availability for systematic code validation.
Detailed validation data are critical in evolving the next generation of simulation capabilities for modern flexible
architectures and mitigating the development risk of future innovative concepts.

### 9.5 Verification and validation

To fully leverage emerging and existing simulation tools, a user must know the reliability of the results for the application
being considered. For example, the confidence of the designer of an optimized wind plant layout is dependent on the
capabilities of the wake models employed. The verification and validation (V&V) process is not performed to endorse or



discourage the use of a particular simulation capability, but to assess the uncertainty of the simulation predictions so a user
        can judge its suitability for a specific application.

        As the name suggests, there are two components to the V&V process. Verification is the process of determining that a
        model implementation accurately reflects the conceptual model. Validation is the process of determining the level to which
        the results from the implemented model reflect the actual physical case for a specific application. Although validation is
the primary focus here, this should not be interpreted as verification being less important.

        Validation compares the results from a particular simulation to a trusted solution, which can be quality experimental data
        or results from other models. Validation is not just performed on the entire system (here a wind turbine), because it is
        difficult to identify the cause of any differences between the simulation results and trusted solutions in complex systems.
        In addition to comparison with results from the full system, comparisons for cases that isolate specific phenomena are also
useful. For example, validating a model's capability to capture fundamental unsteady blade aerodynamics builds
        confidence in that model's ability to accurately predict wake development. In a best-case scenario, a validation campaign
        would measure the level to which the model captures physics of isolated subsystems and the full system.

        Because V&V was introduced in the computational community in the 1990s, it is fair to question why its application to
        wind energy is not straightforward. Among the communities that first adopted the need for V&V were the nuclear and
aerospace fields (AIAA, 1998; Oberkampf and Roy, 2010). In the aerospace field, numerous experimental campaigns have
        been conducted to validate simulation capabilities. Fortunately, validation of wind turbine aerodynamic simulations uses
        data from these campaigns. However, wind energy has phenomena, discussed throughout this article, that are unique to the
        field including high-Reynolds-number blade flows, highly flexible blades that respond to an atmospheric inflow that varies
        along the blade and from blade to blade, the impact of motion of turbines mounted on floating platforms on the flow, wakes
from upstream turbines that impact downstream turbines, and the merging of individual wakes into a wind plant wake. As
        a result, entirely new data sets will be required for assessing wind energy simulation capabilities. A recent set of reports
        considers in detail the application of the V&V process to wind energy (Hills et al., 2015; Maniaci and Naughton, 2019)

        To address wind energy validation needs, a distinct set of validation campaigns are needed to investigate the breadth of
        phenomena in wind energy simulations. To meet this need, experiments will be conducted in wind tunnels and in the field
with test objects ranging from simple geometries to complete wind plants. Some of the experiments will isolate some of
        the phenomena particular to wind energy. Such experiments often use simplified geometries tested in facilities under
        controlled conditions that can leverage existing instrumentation and provide high-quality data with quantified uncertainty
        at a limited cost of time and resources. However, other tests at full scale that include many or all of the phenomena expected
        in a wind plant will also be necessary. Such tests typically take multiple years to plan and execute, require many personnel
to perform, and involve special instrumentation, and still yield data that can have high uncertainty because of limited





duration campaigns in uncontrolled test conditions. Nonetheless, the combination of campaigns of varying levels of complexity is required to provide the range of data required to fully validate wind energy simulation capabilities. Although costly, the return on investment in validating the simulation capabilities available today and the even more powerful tools under development will be high when wind turbine and wind plant developers can better know the degree to which they

can rely on the tools they are using.

At this point, the question is not whether these validation campaigns should be carried out, but whether such an ambitious undertaking is feasible. As already argued in Sect. 5.5.1 (see also Figure 7), the rotorcraft community provides a good example of such a campaign, where unique extensive datasets collected on a military grade helicopter were made available to the scientific community, in turn enabling major advancements of the field. We recommend conducting a similar

coordinated validation program for the wind energy community. A comprehensive validation program will require significant investment but is well-justified based on the wind energy installation goals discussed earlier. Considering the breadth of validation required, multiple government agencies, universities, and industrial partners worldwide are needed for this collaborative effort. A range of experiments from fairly simple but important experiments (e.g. high-Reynolds number airfoil tests) to highly complex tests (e.g. tests in an offshore floating wind plant) will be needed. Designing and

carrying out this validation suite will be an enormous undertaking that will require substantial investment in time, test facilities, and personnel. A key component to this effort will be open data sets that include, to the fullest extent possible, the details of configuration necessary for simulation. These data should be archived in a form that is easily accessed and easy to use. The time to start this effort is now so that the data can have the maximal impact on future wind plants through model improvements that reduce uncertainty, enable more capable turbines and wind plants, yield better power production

predictions, decrease cost and increase penetration, and develop capabilities that facilitate integration with the electrical grid.

### 9.6 Uncertainty quantification

To make the validation process meaningful, quantification of the uncertainty in both experimental data and in computational results is necessary. Although uncertainty analysis is almost routine practice for validation-focused

experiments, uncertainty quantification of simulated results is a relatively more recent development. In the context of simulations, UQ seeks to assess the impact of the uncertainties in inputs and models on the desired quantities of interest (QoIs) (Xiu and Karniadakis, 2002; Najm, 2009). The uncertainty sources (stochastic inputs/parameters) may arise from inherent (irreducible) aleatory variability of what is being modeled, or they may result from lack of knowledge of what the input should be (i.e. epistemic uncertainty). Rational accounting for the uncertainty in the QoI from both the experiment

and simulation allows for a meaningful comparison during the validation process. When previously validated simulation capabilities are used for prediction, uncertainty quantification of the QoI can provide useful confidence bounds or intervals within which model results can be evaluated for a specific application. Challenges remain that are associated with the



efficient implementation of uncertainty quantification for simulations; thus, continued development of more efficient methodologies is required.

**9.7 Instrumentation**

To support the validation-focused effort discussed earlier, instrumentation capable of measuring the quantities of interest is necessary. Instrumentation for wind tunnel testing is fairly mature, and the wind energy community can leverage the facilities and instrumentation developed by the aerospace community. In contrast, instrumentation for field tests of wind turbines and plants is far less mature. There are considerable challenges to performing measurements on turbines ranging
in size from 2 to 20 MW, which are likely to be performed in the future.

A sustained instrumentation development effort will be needed if the previously mentioned validation efforts are to be successful. A list of example efforts is presented here but should not be thought of as comprehensive. As discussed in Section 5, understanding the unsteady loads and resulting response of long flexible blades is critical as turbines grow larger. Measurement systems capable of measuring quantities of interest relevant to such blades will have to operate for long
periods in harsh conditions. For turbines on floating platforms, such measurements will be important to evaluate the effects of rotor plane motion. Examples of measurement systems that could be deployed on turbines to investigate these effects are surface pressure, wall shear stress, blade deformation, and tower and blade accelerations. The importance of understanding and accurately modeling the behavior of wakes and groups of wakes was discussed in Section 6. Velocity measurements upstream, within, and downstream of wind plants are needed to develop this understanding. Depending on
the focus of the measurements, domains and resolutions ranging from 100 mm/0.1 mm (near blade) to 1000 m/100 m (wind plant wake) will be needed. Measurement systems, such as lidar and X-Band radar, have made great strides, but continued development and refinement are needed. Measurements of the atmosphere relevant to wind energy continue to be essential, as the understanding of the atmosphere at larger distances from the ground where modern turbines operate is limited. Continued development of instruments and platforms that can carry them to larger heights, such as unmanned air systems
and tethered balloons, is warranted. As offshore wind energy grows in importance, the challenges of moving many of these measurements to the offshore environment becomes a development issue. In addition, measurements specific to the offshore environment, such as measuring wave properties and the motion of floating platforms, will be required. A detailed discussion of instrumentation for wind energy applications is in Herges et al. (forthcoming).

As is evident, there is much instrumentation development work to do if we are to achieve the measurements necessary to
allow for model validation. A good outcome of this development will be honing the expertise within the wind community to make these instruments and to process them to determine the quantities of interest required for validation.



## 10. Recommendations

Modern wind turbine architectures are engineering marvels, achieving performance on a par with complex aerospace systems from massive, rotating machines producing multi-megawatts of power and designed with the endurance, longevity,

and safety requirements more common to civil engineered structures. Wind turbine energy extraction efficiencies are near theoretical limits, and the total energy production costs meet or beat the LCOE of modern fossil-fuel power plants. Achieving this cost parity has required optimized integrated systems performance, lighter weight, and highly efficient component design while maintaining stringent requirements on safety, performance, and reliability in a market environment of accelerated development and deployment. Advances in materials, system and component design optimization, and in

rotor performance achieved through advanced modeling and simulation, coupled with modern manufacturing methods, have provided the foundation for today's engineering and market successes.

What about the wind energy technology of tomorrow? With global electricity demand already comprising over 5% wind generation, wind energy is one of the few carbon-free sources able to quickly replace fossil-fuel-based generation and supply up to half of all electricity by mid-century. Wind plants will therefore be responsible for delivering not only bulk

electricity, but grid reliability, stability, and resilience. All this will have a major impact on turbine and plant design and control and therefore on what is demanded of modeling approaches. Fundamental changes are needed for continued cost and performance improvements that may lie beyond the existing knowledge of critical physics, applicability of design assumptions, and modeling and simulation capabilities. The size of modern turbines has pushed the limits of the physical understanding of site-specific wind resource characteristics and weather-driven phenomena; the atmospheric sciences will

play a predominant role in future development, deployment, system integration, and dispatchability. Increasingly "dynamically soft" turbine structures are necessary for continued mass and stiffness reductions (light-weighting), leading to capital-cost reduction. Continued innovation in turbine design, not just value engineering of the current product lines (although that will also be a near-term necessity), will require that the whole design tool chain be validated far outside the current assumptions for system configuration. Major opportunities for system improvement will not move from the drawing

board to the field without simulation capabilities that model every aspect of the novel system.

A recurring research gap noted in each review section is the need to further understand and model the physical nonlinear processes and dependencies. Future design, development, and certification processes will rely on a new generation of high-fidelity modeling and analysis methods that incorporate physics and analytic methods needed for deterministic and probabilistic design optimization of dynamically soft systems. Turbine rotors that operate at scales equivalent to the

physical dimensions of the atmospheric surface layer are subject to complex fluid/structure interactions driven by shear, veer, and turbulence. Plant operating environments are further complicated by turbine wake interaction in all wind plants and air/sea interactions when offshore. Each of these key load drivers is a stochastic, highly non-linear physical process, varies both temporally and spatially, is made up of a range of scales, and must be included in any integrated system design



analysis—thus making high-fidelity turbine and integrated-plant systems modeling a computational grand challenge.
Existing tools and design assumptions are simply incapable of capturing the phenomena necessary to yield well-trusted
designs. The complex interactions of coupled physical drivers directly impact both design and operational performance.
Tomorrow's wind technology will require a new generation of high-fidelity modeling and simulation supporting lower-
order tools for design and optimization of complex plant and turbine systems. Continued cost and performance
improvement is also dependent on evolving high-bandwidth control paradigms for active system monitoring and
anticipatory load mitigation and resonance control needed to further reduce design constraints without compromising
safety, resiliency, or operational longevity.

Specific subject area recommendations are listed in the following.

**1. Revise the design process.** Standard inflow definitions and design load cases have been key to high reliability and
manufacturability but are unable to fully capture the complexity of inflow across the rotor as turbines grow to exceed the
height of the atmospheric surface layer. Offshore turbines incorporate both atmospheric inflows above and wave and
current physics below, expanding the dimensionality of the design space and making full-system evaluation increasingly
unwieldy. Manufacturing processes that can create components with higher quality and greater strength or durability need
to be rewarded with reduced (trimmer) safety factors. Novel approaches to the design process, investigating probabilistic
methods, machine learning, the use of reduced-order surrogate models, and other techniques to address the stochastic nature
of the environment and incorporate inherent uncertainty in the manufactured structures need to be investigated and brought
into common practice. A new design process that is better able to capture the physics of inflows, scales, uncertainty, high
dimensionality, and to leverage advanced simulation capabilities is critical to building turbines and plants suited to the
requirements and limitations of each site.

**2. Create new measurement techniques and equipment for turbines, inflow, and wakes.** Existing instrumentation used
in experimental campaigns to validate design capabilities and to reveal the underlying physics of turbine operation often
does not provide the level of resolution or the range of scales needed for wind energy applications. Full-field inflow and
structural loads measurements across entire rotor planes and even multiple wind farms with high spatial and temporal
resolution are needed to accurately represent turbulent winds and the resulting turbine response. Concurrent wind, wave,
aerodynamic, and structural response measurements are also needed. However, distributed aerodynamic load sensors and
structural response measurements are difficult to deploy because of turbine size and harsh environments. Continued
advances in remote-sensing device technologies will be key to measuring surrounding flow fields as turbine sizes continue
to expand and traditional meteorological towers become cost-prohibitive or impractical. Such devices will need to capture
not only the behavior of winds in the immediate inflow but also the wakes both from individual turbines and full-sized
wind plants and their interaction with the surrounding ABL, surface, and neighboring plants. New instrumentation may
include ground-based systems such as advanced lidar, radar or sodar, as well as turbine-mounted instruments or even





instruments installed on mobile airborne platforms. More easily and rapidly deployed measurement systems would make it possible to conduct validation campaigns in remote and difficult-to-access locations. Lastly, future instrumentation systems should include embedded advanced simulation tools (i.e. edge computing) that can augment the capabilities of the hardware and provide more detailed physical insights in as close to real time as possible. Some of the measurement

capability developed may make its way into sensing systems for plant operations, including performance, loads, and health monitoring, as well as for turbine control.

**3. Develop new design and installation methods for offshore systems.** To increase the penetration of offshore wind in the global energy market, more cost-effective designs and installation methods are needed. These methods will require new approaches that consider overall lifetime costs, including manufacturing, installation, O&M, and decommissioning. The

complexity of offshore wind energy designs, with their coupled wind-wave excitations, requires a thorough understanding of the stochastic metocean climate and how it impacts structural loading on these systems. Storm systems with large spatial coverage in the ocean environment, such as 1000 km diameter and larger tropical cyclones, require refined and validated models of complex and coupled constituent wind, wave, and current fields for the design of support structures, control systems, and related power needs. A multi-fidelity toolset is needed to accurately represent the physics, as well as a design

process that integrates these capabilities in a way that is tractable. Extensive validation campaigns are required to ensure that these design practices are accurate across the diverse design space and metocean environments including full-scale campaigns with open information of the measured loads/performance and design properties, as well as the turbine and control system.

**4. Establish a deep-water offshore research facility.** Floating offshore wind energy systems present an even greater

challenge for validation of design tools and evaluation of innovative solutions. The remote and often harsh deep-water offshore environments where such systems are to be sited push the limits for deploying instrumented systems capable of executing meaningful and informative campaigns. Therefore, dedicated, pre-wired facilities with strong data and power connectivity to shore and with a base at a local port are needed for deep-water experimental campaigns. Such a facility could then be leveraged to develop the full-scale data sets critically needed to validate floating wind modeling tools.

Building from the needs for aeroelastic model validation, additional requirements for floating wind systems include time-based measurements of the wave and current environment in coordination with the atmospheric conditions, and the directionality of both. Global motion measurements and loading at different locations in the structure are also critical for understanding the coupling between the environmental loading and the system response. Compared to fixed systems, the large motion of floating offshore installations can significantly change the aerodynamics, hydrodynamics, and elastic

response.

**5. Conduct full-scale wake research.** Wind turbine and plant wakes coupled with other wind plant atmosphere interactions remain the one source of losses that are most difficult to quantify and control in the entire system, resulting in uncertainty



that puts plant revenue projections at risk, which in turn raises the cost of financing. Wake models of individual turbines and plants need to be customized for the full range of turbine operation (including consideration for tip-speed ratio, yaw,

tilt, loading, etc.), for the full suite of atmospheric conditions to which they are exposed, and for which control systems are designed. Turbines in large plants operate in the wakes of other turbines over their entire lifetime, and the wakes need to be incorporated into their design conditions. Bankable, validated, and accurate wake models are required to foster the low-cost financing of future large-scale deployment. Major campaigns are already in the works (Moriarty et al., 2020), but more of these need to be conducted that span the full range of potential atmospheric conditions and characteristic site

topographies, including offshore.

**6. Expand manufacturing research for critical components.** Adoption of advanced materials with superior performance is often inhibited in component manufacturing because of uncertainty in the as-manufactured properties at scale and in the realized cost benefits over the 20 or more years of a turbine's service life. New materials and processes are also slow to be adopted into blade, bearing, and drivetrain manufacturing applications for purposes of achieving longer lifetimes, lower

weight, reduced variability, and recyclability. The research needed to reduce risks of related sub-optimal performance can be prohibitively expensive if they must be conducted at full scale and in significant quantities. Materials and manufacturing research needs to bridge this gap between the lab and the manufactured component to enable innovation to flourish in all the critical components.

**7. Investigate advanced control systems.** Historically, controls research and implementation played a post-design and

development role in addressing turbine-level system integration and performance issues arising during test and deployment. This paradigm has rapidly shifted with the advent of adaptive turbine control to increase turbine energy capture, and plant-level system's wake steering and consensus control demonstrating enhanced plant-level energy capture and market potential. Design of controls has perhaps no longer been an afterthought for more than a decade, and many advances have been made with model-based control strategies that manage loads and optimize power performance in operational and

extreme cases. Analysis has highlighted the benefits of active control in mitigating structural loads that can thereby reduce component mass in pursuit of lighter-weight designs. Future wind technology development requires active control capabilities to be merged into the design and development process and become an integral element of achieving capable, lighter-weight structures. Full system optimization of the physical wind turbine and control design is computationally intensive and prohibitive. Further, existing software tools do not accommodate simultaneous optimization of the many aerodynamic, structural, and control parameters at the level of fidelity typically expected in wind turbine design. Active

monitoring and system-level plant control need to evolve as seamless grid integration and hybrid system dispatchability will be required. It is necessary to develop computationally efficient and accurate software modeling tools so that effective control algorithms can be designed that achieve the desired performance in turbine, plant, and renewable grid systems. There is also a need for development of new and advanced model-based control algorithms that are capable of detecting





and sensing varying operating conditions and adapting and responding appropriately to the situation at hand, at both the
wind turbine and wind farm levels. The challenges of developing probabilistic load control methods, fault-tolerant control
methods, and cyber-physically secure closed-loop controllers are also of growing importance.

**8. Implement high-performance computing and reduced-order modeling.** High-fidelity analysis and modeling
approaches that accurately capture complex flow physics will be needed to fully investigate phenomena uncovered through
experimentation, observation, and subject matter expertise. Ultimately, high-fidelity models operating on advanced HPC
systems will address a multitude of science challenges and barriers to innovation and will provide deterministic and
probabilistic simulations with high accuracy and low uncertainty. While HPC capabilities currently exist and are available,
their application to design and optimization are not yet practical given the numerous simulations needed in the development
of new architectures. Capturing increasingly higher fidelity in the physics and the non-linear dynamics of modern wind
turbines in a suite of design tools, or a single unified modeling code with "selectable" fidelity options, is the preferred
approach to facilitate both the detailed and the iterative design process. The development of the next generation of design
capabilities based on AI and ML has the potential to produce the fidelity and accuracy of HFM at computational speeds
needed for system design and optimization. Further, the continued evolution of advanced modeling and simulation requires
subject matter experts to design, develop, and implement models on advanced computational hardware that is not readily
available outside of academia or the national laboratories. Developing the skills to effectively program within the emerging
parallel computing environment is itself a critical challenge. All of the suggested effort is challenging in itself, but it is
important to note that this activity will take place on computing resources that will continually evolve.

**9. Develop comprehensive open validation data sets.** Limited access to open-source, high-resolution (temporal and
spatial) experimental data, obtained through full-scale field observation, is a critical gap in the development and validation
of next-generation modeling and simulation capabilities, and must be overcome in order to mitigate the development risk
of future innovative concepts. Full-scale, open-source validation campaigns for wind turbine aeroelasticity and control at
multi-megawatt scales requires openly available information on the geometry, structure, and control system, and advanced
instrumentation to measure inflow, aerodynamic forces, structural response, and the resulting wake. As noted previously,
additional requirements for floating wind systems require time-based measurements to fully resolve the wave and current
environment synchronized with detailed atmospheric inflow conditions and turbine/platform response at different structural
locations. Similarly, experimental campaigns to assess detailed component performance, such as the aerodynamic and
acoustic properties at Reynolds numbers relevant to tomorrow's turbines that cannot be tested with existing facilities, are
also needed for development and risk mitigation. Although some of these data sets may exist in the private sector,
intellectual property considerations and private investment in their collection have prevented or delayed their public release.
Future joint international collaboratives between public and private entities are needed to evolve the next generation of
fully validated modeling and simulation capabilities that will drive innovation and continued progress.



**Author Contribution**

All authors were involved in the original draft preparation, review and editing. PV directed the work and was responsible for much of the introductory, recommendations and summary material. LM led Sect. 3 with JR and JP. LP led Sect. 4. CB led Sect. 5 with AB, AB, SGH and HAM. JN led Sect. 6 with PM and SA. AR led Sect. 7 with HB, LM, and JM. JP led Sect. 8 with JK and SN. MR led Sect. 9 with SA and JN. Much material was shared or moved between sections and editing responsibilities were comprehensive, so Sect. authorship is never exclusive.

**Competing interests**

Some authors are members of the editorial board of Wind Energy Science. The peer-review process was guided by an independent editor. The authors have also no other competing interests to declare except what is implied by their affiliations.

**Acknowledgements**

The authors appreciate the substantial time and effort taken by Takis Chaviaropoulos, Simon Watson, Dominic von Terzi, Rogier Blom, Walt Musial, and Michael Sprague to comment on an early draft. Their insights resulted in significant revision and expansion of this article, which undoubtedly improved its message and corrected both errors and omissions. Support from a Palmer endowed chair at the University of Colorado Boulder is gratefully acknowledged. This work was authored in part by the National Renewable Energy Laboratory, operated by Alliance for Sustainable Energy, LLC, for the U.S. Department of Energy (DOE) under Contract No. DE-AC36-08GO28308. This work was also authored in part by Sandia National Laboratories, a multimission laboratory managed and operated by National Technology & Engineering Solutions of Sandia, LLC, a wholly owned subsidiary of Honeywell International Inc., for the U.S. Department of Energy's National Nuclear Security Administration under contract DE-NA0003525. The views expressed in the article do not necessarily represent the views of the DOE or the U.S. Government. The U.S. Government retains and the publisher, by accepting the article for publication, acknowledges that the U.S. Government retains a nonexclusive, paid-up, irrevocable, worldwide license to publish or reproduce the published form of this work, or allow others to do so, for U.S. Government purposes

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
