# Peer review of "Grand Challenges in the Design, Manufacture, and Operation of Future Wind Turbine Systems"

_Wind Energy Science, 2022_

## Referee Comment (RC2)

**Review of "Grand Challenges in the Design, Manufacture, and Operation of Future Wind Turbine Systems"**

**Author: Paul Veers *et al.***

The manuscript is one of the most comprehensive reviews of wind turbine systems that the reviewer has come across. It provides detailed discussions about the range of topics from inflow wind; aerodynamics and aeroelasticity of wind turbines; turbine and wind farm controls; experimental capabilities, modelling, field measurements; offshore wind turbines and so on. The manuscript is well written and is easy to read. The reviewer however suggests the authors to reduce the repetition of the discussions as much as possible. Some discussions in the earlier sections are repeated in the later sections (probably unintentionally). Some minor comments which are listed below. Other than that, the reviewer strongly recommends publication of this manuscript.

**Specific comments:**

1. Pg 11, line 282: "The accuracy . . . "
   This sentence is not clear

2. Pg. 20, line 519: "However, the size of modern-day turbines . . . "
   Are there studies which have investigated how existing inflow models may not be suitable for modern wind turbine size? If so please add them as references over here.

3. Having one sub-sub section 3.5.1 in sub-section 3.5 does not make sense.

4. Pg 31, line 855 to 863
   This paragraph is not very clear. How will the measurement during the development stage pose challenge?

5. Pg. 33, line 907: ". . . **and damage of the turbine and its components**."
   I do not think damage of the turbine and its components themselves are uncertainties, but they are the consequence of uncertainties.

6. Figure 6: From this figure, one may interpret that Hybrid RANS-LES is a lower fidelity model compared to LES. This may not be true. Hybrid RANS-LES is more like one of the LES methods. So you can either put the Hybrid RANS-LES figure at the same level as LES or remove it.

7. Pg 58, line 1599: "The highly competitive . . . manufactured at costs less that \$20/kg . . . "
   Reference is needed.

8. Pg 58, line 1608: Remove **and**

9. Pg 62, lin 1710 to 1719: Do you have reference(s) for this paragraph?

---

## Referee Comment (RC3)

Paper ID:        wes-2022-32

Paper Title:    Grand Challenges in the Design, Manufacture, and Operation of Future Wind Turbine Systems

Authors:        Paul Veers, Carlo L. Bottasso, Lance Manuel, Jonathan Naughton, Lucy Pao, Joshua Paquette, Amy Robertson, Michael Robinson, Shreyas Ananthan, Athanasios Barlas, Alessandro Bianchini, Henrik Bredmose, Sergio González Horcas, Jonathan Keller, Helge Aagaard Madsen, James Manwell, Patrick Moriarty, Stephen Nolet, Jennifer Rinker

Reviewer:      Panagiotis (Takis) Chaviaropoulos

**REVIEWER's COMMENTS**

This article is a joint effort of highly reputed researchers in the field of wind energy to identify critical unknowns in the design, manufacturing, and operability of future wind turbine and wind power plant systems and articulate and recommend relevant research actions.

The paper is complete and well and clearly written. It is not the first time that the reviewer meets the text, as it becomes clear from the Acknowledgements section. Having read and commend on earlier versions of the article, which the authors took under consideration, there is not much new to suggest for further improving the quality of the publication. Some minor comments are:

1.  There is an extended list of authors cosigning the publication. It would be nice to know, already from the introduction how these authors came to work together and under which initiative.
2.  Section 1.3 Comment on the scope. The reviewer believes that the scope should be further refined. The paper focuses on "Mechanical Engineering" aspects and does not elaborate on other disciplines that maybe of equal importance for wind energy research. This is not necessary a problem if the boundaries are clearly stated in the scope.
3.  Some additional references may better support some of the statements in the text. Examples:
    a.  lines 153-154, statement on non-Gaussian turbulence
    b.  lines 337-338, machine learning training of low-order models
    c.  lines 958-960, high-fidelity CFD and ML training low-order models
    d.  lines 1090-1091, support statement with reference
    e.  lines 1677-1679, new materials with increased structural damping (see for instance the DAMPBLADE EU project)
    f.  lines 1715-1719, new materials
    g.  lines 1748-1750, new epoxy resin
    h.  lines 1800-1801, non-destructive inspection technologies

Given the above, the reviewer's final recommendation for this article is "Publication with minor corrections".

---

## Author Comment (AC1)

**REVIEWER's COMMENTS and Authors' responses**

**Referee #1 Pietro Bortolotti**

This article continues the exciting series of publications about the grand challenges faced by wind energy technology. The list of authors is the right one for such publication and I support the publication of this long article, which can become the backbone of many research proposals moving forward.

I do not have the expertise to add anything to some of the sections, but I do have a list of suggestions.

Overall structure:

1) I would have liked to read a paragraph about what's new in this article compared to previous publications focusing on the grand challenges of wind energy, for example https://www.science.org/doi/10.1126/science.aau2027, https://www.nrel.gov/docs/fy19osti/72437.pdf, or https://doi.org/10.5194/wes-1-1-2016

> **Authors' Response:** Excellent suggestion - the Introduction has been expanded to include these references and an explanation of what is new.

2) Some topics are grouped together into the same section, and sometimes I miss the rationale. Why inflow and the design process together, but aerodynamics and aeroelasticity in a separate section? Why HFM and validation? And uncertainty quantification is described in a single paragraph in 9.6, although there is a large body of literature addressing it.

> **Authors' Response:** The authors appreciate this comment - the organization into a minimal set of sections was a point of major discussion during the planning of this paper. In covering this expansive treatment of all the effects that impact turbine design, some condensation and compromise is required, and the chosen structure may not be unique. Inflow determines the design criteria and is therefore fundamental to the design process. However, the design process permeates the full article. We agree that perhaps a better title would therefore be "Inflow and design criteria" and have moved the discussion of "Design" to the Introduction. Aero and Aeroelasticity are a natural pairing. The main message is that all issues are interconnected. We have made a greater effort to explain in the beginning of section 2. The section on uncertainty has also been expanded.

3) Some sections are built on references (sections #3, #4, #5, #6), which I think is a must for this type of articles. Other sections use very few (#7, #8, #9). I'd suggest consistency, possibly leaning toward the first approach. References are not present in section #2, and that might be ok, but a note pointing the reader to the following sections would help.

> **Authors' Response:** Significantly more references have been added throughout, especially in #7 and #8.. Much of the manufacturing section (#8) is based on the direct experience of the authors who are engaged in industry manufacturing of blades. The HFM Section (#9) was not intended as a comprehensive review of the current methods applied given the extensive application as indicated in multiple topic areas. Rather, an indication of the path forward gleaned from all topic areas noting use of HFM methods.

The references included provide an extensive summary of multiple applications with numerous references to specific HFM methodologies. A note was made to this effect in the section 9 opening.

Abstract

Line 15: what about wind farm blockage?

**Authors' Response:** A reference to wind plant interference has been added to the abstract.

Line 21: I see the lack of validation as a key obstacle for a wider adoption of HFM. BEM, with all its limitations, has shown to work and several publications show what matches and what not. HFM not yet, or at least far less.

**Authors' Response:** Good point. The idea that HFM is not yet fully validated has been added to the abstract.

Intro

Figure 2: I wouldn't have the Emirates logo on a scientific paper that will receive thousands of downloads and citations. Please edit.

**Authors' Response:** The figure has been changed, and now shows the A380 with the manufacturer's logo, as for the turbine picture.

Section 2

Line 125: LCOE has clearly shown its limitations (https://doi.org/10.1016/j.tej.2021.106931, among many others). Such limitations will only grow with a higher share of wind energy in the mix and the growth of hybrid systems and storage. Should this be a challenge?

**Authors' Response:** The fact that other value functions beyond LCOE have and are being considered is indeed mentioned and discussed in some detail in the paper (see Sect. 3.1 of the original manuscript, now in Sect. 1.2 of the revised version, but also at the beginning of Sect. 3 etc.). To further strengthen this aspect, we have added a new reference (Canet et al., 2022) that discusses the use of various economic and environmental value metrics in wind turbine design.

Line 136: This sentence might sound a little unfair for the wind energy pioneers. Unknowns have always been out there. We've resolved some, some others are still there, and some new ones have come up

**Authors' Response:** This is a good point and was intended to be clear, but needs more specificity. Wording has been added to be specific that "we have gradually moved outside the design basis and are acting as though we were not. High safety factors have kept us safe, but continuously removing cost and weight pushes safety margins down. As we try to customize designs for the specific conditions of each application, unknowns expand and the science gaps evolve."

Line 257: It won't be possible to increase tip speeds without innovations on the leading edge erosion front. I'd suggest adding a note

> **Authors' Response:** This discussion is about acoustic constraints. The erosion issue is a blade material problem, solvable in other ways and not necessarily a constraint. It is discussed extensively in chapter 8. However, to be clear, we have added "assuming blade leading-edge erosion can be managed."

Section 2.6: I miss a paragraph about the cost and complications of testing prototypes, which I understand to be a very real obstacle to innovation. What about allowing subcomponent testing? Or moving toward numerical prototyping in 2.7?

> **Authors' Response:** We have added a sentence about verification of parts (rather than the whole component) or verification by simulation, which was already described in 8.2. We agree this is an important enough point to have added to 2.6. I wouldn't describe these sorts of simulations as "in" the HFM realm (at least not yet), even though they are FEM-based. Section 2.7 has also been updated to reflect the opportunity for numerical prototyping and its inherent demand for validated and predictive models.

Line 341: I agree that validation is key for the HFM to succeed. But validation is easier said than done even when data are present. We lack rigorous data assimilation techniques merging multiple instruments and the inherent errors in their measurements into a single validation process. This is listed in section 9, but I'd suggest adding a note here as well.

> **Authors' Response:** A comment on data assimilation difficulties and the need to improve has been added.

Section 2.8: I found 2.8 somewhat blurry to me. I'd suggest to reformulate how the authors propose to move forward, which in my head is system design based on validated and therefore trustworthy numerical models

> **Authors' Response:** Agreed. We have restructured the paragraph.

Section 3

As mentioned before, I am not sure whether I understand why inflow and design process come into the same section.

> **Authors' Response:** Agreed. We have changed the heading to "Inflow and design criteria" and elevated the discussion of the goals of design to the Introduction.

Line 418: I feel that a reader could get confused here as LCOE and value are used almost interchangeably. The references are very relevant and maybe this section could be expanded a little to better explain these trends without the need to accessing the references. Also, I would add a note about the arrival of storage solutions, which might change trends very rapidly.

> **Authors' Response:** In this case we disagree. We feel the discussion clearly distinguishes cost from value and discusses the complications derived from such a nuanced market and application. No change has been made.

Line 428: I personally like the words "numerical prototyping" to describe this innovation

> **Authors' Response:** The reference to numerical prototyping has been inserted.

Section 4

Line 780: I found this paragraph a little confusing. In my mind the biggest obstacle is the combination of a very complex (nonlinear) solution space generated by aero-servo-hydro-elastic models combined with high computational costs. This is probably what authors are saying, but I think it could be written more explicitly.

> **Authors' Response:** Thank you for this comment. We have revised this paragraph to make this point clearer.

Line 812: This is possibly misleading. The 25% gain in LCOE was mostly generated by switching from 3 to 2 blades, not by the actual codesign optimization. That change was done upfront and it wasn't driven by co-design approaches, but rather by a change in turbine configuration.

> **Authors' Response:** Thank you for the opportunity to clarify this common conclusion that the 25% reduction in LCOE was primarily a result of reducing from 3 to 2 blades. We have added several sentences to more clearly explain that control co-design was in fact necessary to achieve the 25% reduction in LCOE: "While reducing from three to two blades led to a 25% decrease in *rotor* mass and hence an approximately 25% decrease in *rotor* cost, it only yielded about a 7% decrease in LCOE for the entire wind turbine (Pao et al., 2021). This is because the rotor is only a portion of the wind turbine, which also includes the tower and nacelle. Further, the wind turbine only accounts for approximately one third of the overall capital expenditures for fixed-bottom offshore wind systems and less than one quarter of the overall capital expenditures for floating offshore wind systems (Stehly and Duffy, 2022). Hence, control co-design was instrumental in achieving 25% lower LCOE in the novel two-bladed wind turbine design relative to conventional three-bladed turbines (Pao et al., 2021)."

Section 5

Line 1050: I would add the lack of validation to the list of obstacles slowing HFM

> **Authors' Response:** This list is about aerodynamic details, not a comprehensive list of obstacles. No change.

Line 1080: you could add https://doi.org/10.1088/1742-6596/2265/4/042052 to the list of free form studies

> **Authors' Response:** This reference has been added.

Line 1100: I'd suggest to add references about successful scaled blade testing, for example https://doi.org/10.5194/wes-5-1411-2020, https://arc.aiaa.org/doi/abs/10.2514/6.2021-1718, and https://doi.org/10.1016/j.renene.2020.03.157

> **Authors' Response:** These references have been added.

Line 1137: I see one big obstacle here. No funding agency / national lab / research institute has designed a multi-MW wind turbine, at least not in the last 20 years. How should these organizations convince an OEM to publish data? I'd add a note about this key obstacle

> **Authors' Response:** Yes this is the critical issue, which was alluded to but not stated explicitly. A note to this effect has been added..

Line 1147: please reference https://doi.org/10.2172/1868328

> **Authors' Response:** Reference added.

Section 7

The start of section 7 gives the impression that offshore wind is something of the future, whereas it is already reality. Also, the paragraph seems to imply that offshore wind is only present in the North Sea, whereas Asia also has offshore wind farms (maybe not yet fully commissioned?). The same consideration is valid for floating, which is certainly in a development stage (line 1342), but there are already the first commercial projects operating out there (Hywind in Scotland?)

> **Authors' Response:** We agree with the impression that offshore wind is something of the future, and have tried to rewrite the first paragraph to address the comment. China has also been added as a significant contributor to the fixed offshore wind industry. For floating, commercial scale is considered as significantly larger than what is existing today. At 30 MW, we consider Hywind Scotland a "pre-commercial" farm.

Line 1467: an image would help describing DOFs and platform configurations

> **Authors' Response:** A figure has been added.

Section 8

I've struggled with this section, which I think needs some attention.

To start, showing a 40m long blade (figure 12) in a paper about the future of wind seems a little out of place

> **Authors' Response:** We agree that a larger and more recent blade manufacturing photo would be helpful. Another photo showing the manufacture of a 97m Siemens Gamesa Renewable Energy blade from 2020 has been added. The existing photo makes an excellent visualization of the difficulty in bonding while the additional photo shows the challenge of fiber layup.

Also, the section alternates considerations about blades, tower, drivetrain. I found the resulting text very confusing and I would suggest splitting the section per component.

> **Authors' Response:** We understand this viewpoint and don't necessarily disagree, but believe reorganizing is more trouble than it's worth at this point. We added a bit more text to balance drivetrains with blades, and also added a few more recent drivetrain references.

The section also adopts very few references, which does not help. Focusing on the blades alone, some relevant recent studies in this broad field were led by Ennis, for example https://www.osti.gov/servlets/purl/1592956, or Murray, for example https://doi.org/10.1016/j.jclepro.2018.10.286

> **Authors' Response:** Many additional references have been added including Ennis and Cousins, Murray, et al., etc.

Going into the details:

Line 1672: I don't think you should talk about downwind as related to manufacturing challenges. Also, if you do, there is a vast body of literature about downwind technology.

> **Authors' Response:** Agree, deleted.

Line 1678 should specify whether this damping value is critical or logarithmic. A reference supporting this value would help, maybe https://www.osti.gov/pages/biblio/1726023 or http://www.measnet.com/wp-content/uploads/2020/04/16bt01-Report-IECRE.pdf.

> **Authors' Response:** Modified text to state 1% or less of critical damping and added recommended reference.

Line 1696: I don't think we're above 100 m/s yet due to aeroacoustic and erosion issues

> **Authors' Response:** Perhaps the past fleet is below 100 m/s but a future above 100 m/s is sure to come. At least the Vestas 15 MW has design tip speed over 100m/s. We have added this reference:
>
> de Vries, Eize: Exclusive: How Vestas beat rivals to launch first 15MW offshore turbine, Windpower Monthly, https://www.windpowermonthly.com/article/1706924/exclusive-vestas-beat-rivals-launch-first-15mw-offshore-turbine, 10 February 2021. (last access 26 Dec. 2022)

Line 1709: I understood that the whole point of thermoplastics is about being recyclable, whereas the text seems to imply that recyclability is only a side advantage

> **Authors' Response:** - We agree and have modified this paragraph to more accurately explain the role of thermoplastics including both manufacturability and property advantages in addition to the primary motivation of recyclability.

Line 1710: CFRP is not necessarily linked to BTC, which can be realized with GFRP as well.

> **Authors' Response:** - We agree. This paragraph is modified to more accurately explain that BTC is facilitated by carbon fibers, but not dependent on their use.

Line 1713: "Recent research" calls for references

> **Authors' Response:** Added References
>
> [1] Ennis, B. et al (2019), "Optimized Carbon Fiber Composites in Wind Turbine Blade Design", Albuquerque, NM: Sandia National Laboratories, SAND2019-14173.
>
> [2] Bonds, T., "Low Cost, High Volume, Carbon Fiber Precursor for Plasma Oxidation", Knoxville, TN: Institute for Advanced Composite Manufacturing Innovation, IACMI/R006-2020/6.13.
>
> [3] Shin, H.K. et al, "An overview of new oxidation methods for polyacrylonitrile-based carbon fibers", Carbon Letters, Vol. 16, No. 1, p. 11-18 (2015).
>
> [4] Ogale, A., et al, "Recent advances in carbon fibers derived from biobased precursors", Journal of Applied Polymer Science, May 2016, DOI: 10.1002/APP.43794.

[5] Weisenberger, M., Boyer, C., "Preparation of Mesophase Pitch Feedstock for Carbon Fiber ", Knoxville, TN: Institute for Advanced Composite Manufacturing Innovation, IACMI/R005-2019/6.18.

[6] Sloan, J. "Carbon Fiber Suppliers Gear-up for Next Gen Growth", Composites World, March 2020, p. 12-13.

Line 1719: I'm not so convinced that the price of CFRP has been steadily declining, especially recently. References or numerical values are needed.

> **Authors' Response:** See references above. Also, we purposely avoided quoting prices as much as possible because prices can fluctuate rapidly and that can cause the article to quickly go out of date.

Line 1728: I wouldn't link the EOL issues of blades with the immaturity of wind, which could be called mature since 2010 (at least). Isn't it more like a lack of a good alternative? Research on thermoplastics have been going on for decades, see the body of literature from TU Delft for example with Prof Beukers and Prof Bersee

> **Authors' Response:** The point is that very few turbines of the total installed wind energy capacity have been decommissioned at this point because they have a 20+ year lifespan and annual installations have been growing rapidly. Therefore the EOL issues have not been critical until now. The wording is changed to better reflect the intent.

Section 9

The intro of section 9 is heavy on the progress funded by DOE, but there is more than that, see for example the work happening at DTU or at Uni Stuttgart (Prof Lutz)

> **Authors' Response:** References to the activities of the mentioned institutions have been added. The introduction of this section has been rephrased, so that it does not strictly revolve around US research centers. More could be included, but the main idea is to show where to go, rather than give a comprehensive review of what has been done.

Line 1859: I don't understand the use of the word especially. I understood that was the actual issue.

> **Authors' Response:** The point is that wind turbines operate at very high Reynolds numbers for which wind tunnel data is especially difficult to obtain. The wording is changed to better reflect the intent.

I would have had section 9.5 at the top of the paper, since this affects every field, not just HFM. Just a suggestion, the authors can leave things as they are.

> **Authors' Response 1:** The recommended move to the front is tempting. It is also possible that the discussion would not have sufficient background at that point in the paper. Perhaps reconfiguring the paper to move the V&V discussion up front at this time may do more damage than it is worth. We will keep the discussion where it is for now. Also, "What drives validation efforts?" is why this section appeared here. Although we agree that it applies to all facets of wind energy simulation, it is really HFM that drives it.

We added a sentence indicating the importance to all modeling efforts to address this comment.

Similarly, section 9.6 looks weird. It's only a tiny paragraph hidden inside a much wider discussion about HFM and validation, with no references.

**Authors' Response:** The section has been expanded and several references were added.

Section 10

Line 2148: I think that a key improvement would be to move from single turbine design approaches to designs that account for wind farm effects.

**Authors' Response:** The authors have had similar thoughts but found this level of design change to have lukewarm support from the broader community. The issue of designing to turbine wake impacts was discussed in the body of the paper, so a mention of the need to design for wind farm effects has been added.

---

## Author Comment (AC2)

**Referee #2 Anonymous**

**REVIEWER's COMMENTS and Authors' responses**

Review of "Grand Challenges in the Design, Manufacture, and Operation of Future Wind Turbine Systems"

Author: Paul Veers et al.

The manuscript is one of the most comprehensive reviews of wind turbine systems that the reviewer has come across. It provides detailed discussions about the range of topics from inflow wind; aerodynamics and aeroelasticity of wind turbines; turbine and wind farm controls; experimental capabilities, modelling, field measurements; offshore wind turbines and so on. The manuscript is well written and is easy to read. The reviewer however suggests the authors to reduce the repetition of the discussions as much as possible. Some discussions in the earlier sections are repeated in the later sections (probably unintentionally). Some minor comments which are listed below. Other than that, the reviewer strongly recommends publication of this manuscript.

*Authors' response - Thank you for the recognition of the comprehensive nature of the effort. Some repetition is necessary as each section may be read independently; people may not read the whole document but may seek out topics of particular interest. However, we have tried to remove any unnecessary repetition.*

Specific comments:

1. Pg 11, line 282: "The accuracy . . . "
   This sentence is not clear

*Authors' response - The sentence has been modified to say, "A new class of high-fidelity computational capabilities based on first principles and limited modeling assumptions offers a valuable means by which these tools can be made more accurate and validated for situations where measurements are impossible."*

2. Pg. 20, line 519: "However, the size of modern-day turbines . . . "
   Are there studies which have investigated how existing inflow models may not be suitable for modern wind turbine size? If so please add them as references over here.

*Authors' response - This is more of a commonly noted issue that is more or less obvious than a research outcome. There are references that touch on this topic, including:* Hannesdottir, Á., Kelly, M., and Dimitrov, N.: Extreme wind fluctuations: joint statistics, extreme turbulence, and impact on wind turbine loads, Wind Energ. Sci., 4, 325–342, https://doi.org/10.5194/wes-4-325-2019, 2019.

3. Having one sub-sub section 3.5.1 in sub-section 3.5 does not make sense.

*Authors' response - Agreed. 3.5.1 has been changed to 3.6.*

4. Pg 31, line 855 to 863 This paragraph is not very clear. How will the measurement during the development stage pose challenge?

*Authors' response -* *We have reworked the paragraph to clarify things as best we can.*

5. Pg. 33, line 907: ". . . and damage of the turbine and its components."
   I do not think damage of the turbine and its components themselves are uncertainties, but they are the consequence of uncertainties.

*Authors' response -* *Agreed. The sentence has been changed to remove that inconsistency.*

6. Figure 6: From this figure, one may interpret that Hybrid RANS-LES is a lower fidelity model compared to LES. This may not be true. Hybrid RANS-LES is more like one of the LES methods. So you can either put the Hybrid RANS-LES figure at the same level as LES or remove it.

*Authors' response -* *The figure has been modified, replacing "Hybrid RANS-LES" with "Hybrid methods". Details about the various possible CFD approaches used in wind energy are given in the text.*

7. Pg 58, line 1599: "The highly competitive . . . manufactured at costs less that $20/kg . . .
   " Reference is needed.

*Authors' response -* *The discussion has been expanded to indicate that this is an approximate target cost for wind turbine systems (which is well known in the turbine manufacturing world) and an order of magnitude lower that typical aerospace costs.*

8. Pg 58, line 1608: Remove and

*Authors' response -* *Done*

9. Pg 62, lin 1710 to 1719: Do you have reference(s) for this paragraph?

*Authors' response -* This *paragraph has been significantly modified to recast the current challenges with carbon fiber as one of producing carbon fiber in a more cost-effective way specifically adapted to wind turbine blade current needs and potential future design applications.*

---

## Author Comment (AC3)

**Reviewer #3** Panagiotis (Takis) Chaviaropoulos

**REVIEWER's COMMENTS and Authors' responses**

This article is a joint effort of highly reputed researchers in the field of wind energy to identify critical unknowns in the design, manufacturing, and operability of future wind turbine and wind power plant systems and articulate and recommend relevant research actions.

The paper is complete and well and clearly written. It is not the first time that the reviewer meets the text, as it becomes clear from the Acknowledgements section. Having read and commend on earlier versions of the article, which the authors took under consideration, there is not much new to suggest for further improving the quality of the publication. Some minor comments are:

1.    There is an extended list of authors cosigning the publication. It would be nice to know, already from the introduction how these authors came to work together and under which initiative.

> **Authors' response -** The following text has been added to the Introduction.
>
> "This article provides the details of research on the turbine that could not be included in the highly condensed article in *Science* which defined three areas of Grand Challenge: the atmosphere, the turbine, and the plant/grid (Veers, et al., 2019a). It also updates the work of the European Academy of Wind Energy (EAWE) in their foundational assessment of long-term research challenges given the emerging understanding of the issues over the last half-dozen years (van Kuik, et al., 2016). This new work is encouraged by the EAWE Publications Committee and is intended to assist the International Energy Agency (IEA) Wind Technology Collaboration Platform (TCP) in generating a revised roadmap for critical wind energy research.

2.    Section 1.3 Comment on the scope. The reviewer believes that the scope should be further refined. The paper focuses on "Mechanical Engineering" aspects and does not elaborate on other disciplines that maybe of equal importance for wind energy research. This is not necessary a problem if the boundaries are clearly stated in the scope.

> **Authors' response** - This is an excellent point. The following paragraph has been added to the "Scope" section:
>
> "As noted above, design is an exercise that necessarily encompasses a wide range of topics, involving engineering, environmental, and social considerations. However, to set some boundaries on this article and keep the discussion to a manageable length, the topic of social science and the intersection of design with environmental considerations are left to other articles in this series. This article will focus on the mechanical aspects of the turbine, its interaction with the atmosphere and sometimes the ocean, and the critical issues that arise from that perspective."

3.    Some additional references may better support some of the statements in the text. Examples:

> a.    lines 153-154, statement on non-Gaussian turbulence
>
> **Authors' response -** The following references have been added:
>
> Moeng CH, Rotunno R. Vertical-velocity skewness in the buoyancy-driven boundary layer. Journal of the Atmospheric Sciences 1990; 47(9): 1149–1162

Berg, J., Natarajan, A., Mann, J., and Patton, E. G. (2016) Gaussian vs non-Gaussian turbulence: impact on wind turbine loads. Wind Energ., 19: 1975– 1989. doi: 10.1002/we.1963.

H. Gontier, A. P. Schaffarczyk, D. Kleinhans, and R. Friedrich, A comparison of fatigue loads of wind turbine resulting from a non-Gaussian turbulence model vs. standard ones, J. Phys. Conf. Ser. 75, 012070 (2007).

b.      lines 337-338, machine learning training of low-order models

**Authors' response -** *We have added additional references to the AI/ML sections 9.3 & 9.4 addressing these two introductory paragraphs.*

c.      lines 958-960, high-fidelity CFD and ML training low-order models

**Authors' response -** *We have added additional references to the AI/ML sections 9.3 & 9.4 addressing these two introductory paragraphs.*

d.      lines 1090-1091, support statement with reference

**Authors' response -** *We have added a sentence pointing to two relevant references, of which one is new in the revised manuscript (Yin et al. 2022).*

e.      lines 1677-1679, new materials with increased structural damping (see for instance the DAMPBLADE EU project)

**Authors' response -** *Excellent suggestion. Reference to the DAMPBLADE findings has been added.*

f.      lines 1715-1719, new materials

**Authors' response -** *References to the new materials have been added.*

g.      lines 1748-1750, new epoxy resin

**Authors' response -** *References to the new epoxy resins have been added.*

h.      lines 1800-1801, non-destructive inspection technologies

**Authors' response -** *References have been added on the topic of non-destructive inspection methods for wind blades.*

Given the above, the reviewer's final recommendation for this article is "Publication with minor corrections".

**Authors' response -** *Thank you very much for your review and recommendation!*